

**Interactive effects of seawater carbonate chemistry, light intensity and nutrient**
**availability on physiology and calcification of the coccolithophore *Emiliania***
***huxleyi***
**Yong Zhang[1], Feixue Fu[2], David A. Hutchins[2], and Kunshan Gao[1]**
[1]State Key Laboratory of Marine Environmental Science, Xiamen University, Xiamen,
China
[2]Department of Biological Sciences, University of Southern California, Los Angeles,
California
Running head: *carbonate chemistry,* Emiliania huxleyi, *light, nutrients*
Correspondence to: Kunshan Gao (ksgao@xmu.edu.cn)
Keywords: calcification; $CO_2$; coccolithophore; growth; light; nutrient;
photosynthesis





**Abstract.** Rising atmospheric carbonate dioxide ($CO_2$) levels lead to increasing $CO_2$
concentration and declining pH in seawater, as well as ocean warming. This enhances
stratification and shoals the upper mixed layer (UML), hindering the transport of
nutrients from deeper waters and exposing phytoplankton to increased light intensities.
In the present study, we investigated combined impacts of $CO_2$ levels (410 µatm (LC)
and 925 µatm (HC)), light intensities (80–480 µmol photons $m^{-2}$ $s^{-1}$) and nutrient
concentrations [101 µmol $L^{-1}$ dissolved inorganic nitrogen (DIN) and 10.5 µmol $L^{-1}$
dissolved inorganic phosphate (DIP) (HNHP); 8.8 µmol $L^{-1}$ DIN and 10.5 µmol $L^{-1}$
DIP (LN); 101 µmol $L^{-1}$ DIN and 0.4 µmol $L^{-1}$ DIP (LP)] on growth, photosynthesis
and calcification of the coccolithophore *Emiliania huxleyi*. HC and LN synergistically
decreased growth rates of *E. huxleyi* at all light intensities. High light intensities
compensated for inhibition of LP on growth rates at LC, but exacerbated inhibition of
LP at HC. These results indicate that the ability of *E. huxleyi* to compete for nitrate
and phosphate may be reduced in future oceans with high $CO_2$ and high light
intensities. Low nutrient concentrations increased particulate inorganic carbon quotas
and the sensitivity of maximum electron transport rates to light intensity. Light-use
efficiencies for carbon fixation and calcification rates were significantly larger than
that of growth. Our results suggest that interactive effects of multiple environmental
factors on coccolithophores need to be considered when predicting their contributions
to the biological carbon pump and feedbacks to climate change.



## 1 Introduction

Anthropogenic emission of $CO_2$ is taken up by the oceans, decreasing pH of seawater and resulting in ocean acidification (OA) (Caldeira and Wickett, 2003). On the other hand, rising atmospheric $CO_2$ also leads to global and ocean warming, which enhances water column stratification and shoals the upper mixed layer (UML) (Wang et al., 2015). This exposes phytoplankton dwelling in the UML to higher light intensities (Gao et al., 2012; Hutchins and Fu, 2017). In addition, enhanced stratification reduces the transport of nutrients from deep oceans to the UML (Behrenfeld et al., 2006), which reduces the nutrient concentrations in the UML.

Coccolithophores take up $CO_2$ and/or $HCO_3^-$ from media for carboxylation, and use $HCO_3^-$ for calcification which produces coccoliths. Calcification processes generate $CO_2$ due to production of protons, which counteracts with photosynthetic $CO_2$ fixation, and therefore influencing $CO_2$ influx into the oceans (Rost and Riebesell, 2004). Growth rate, particulate organic (POC) and inorganic carbon (PIC) production rates of *Emiliania huxleyi*, the most abundant calcifying coccolithophore species, usually display optimum responses to a broad range of $CO_2$ concentration, with growth, POC and PIC production rates increased, decreased or unaffected by rising $CO_2$ treatments (Langer et al., 2009; Richier et al., 2011; Bach et al., 2015; Jin et al., 2017). Increased light levels could counteract the negative effects of rising $CO_2$ on calcification in *E. huxleyi* when grown under natural fluctuating sunlight (Jin et al., 2017). Differences in sampling locations, experimental setups, and deviations in the





measuring methods can generally be responsible for the differential responses of
growth, POC and PIC productions to rising $CO_2$ in *E. huxleyi* (Meyer and Riebesell,

70  2015).

POC production as well as growth rates usually increase with elevated light levels,
level off at saturated light levels and decline at inhibited high light levels in
cocolithophores (Zhang et al., 2015; Jin et al., 2017). Reduction in pigment content
and effective photochemical quantum yield ( $F_v^{'}/F_m^{'}$ ) are characteristics of
photo-acclimation (Geider et al., 1997; Gao et al., 2012). At low light intensity, the
ratio of light-harvesting protein to photosystem II (PSII) reaction center proteins is
large, which facilitates *E. huxleyi* to absorb more energy. At high light intensity, the
ratio of photo-protection proteins to PSII reaction center proteins is large, which could
protect *E. huxleyi* against damage caused by high light intensities (Mckew et al.,

80  2013).

Nitrogen is required for the biosynthesis of proteins and other macromolecules,
including chlorophyll (Riegman et al., 2000). Phosphorus is required for the synthesis
of nucleic acids, ATP, and phospholipids in cell membranes (Shemi et al., 2016). Due
to source limitation, decreased nutrient concentrations usually reduce growth and
photosynthetic carbon fixation rates (Cloern et al., 1999; Kim et al., 2007; Harrison et
al., 2008). Nevertheless, low nutrient concentrations often enhance the PIC quotas of
*E. huxleyi*. This is due to the fact that low nutrient concentrations hold the cells in the
G1 cell cycle phase where calcification occurs (Müller et al., 2008). A recent
proteome study on *E. huxleyi* also shows that nutrient limitation arrests cell cycling




(McKew et al., 2015). At molecular levels, nitrate or phosphate limitations
down-regulate expression of genes involved in cell cycling, RNA and protein
synthesis in *E. huxleyi* (Rokitta et al., 2014, 2016).
Recently, several studies investigated interactive effects of rising $CO_2$ and light
intensity on physiological rates of coccolithophores (Feng et al., 2008; Jin et al.,
2017). Zhang et al. (2015) reported that at 50–800 µmol photons $m^{-2}$ $s^{-1}$, rising $CO_2$
levels decreased the maximum growth rate, POC production rate and PIC production
rate of *Gephyrocapsa oceanica*. At low light levels, coccolithophores increase $CO_2$
uptake to compensate for inhibition of $HCO_3^-$ uptake on photosynthesis, while at
high light intensity they don't increase $CO_2$ uptake (Kottmeier et al., 2016). Under
natural solar radiation, Jin et al. (2017) reported that rising $CO_2$ levels increased the
growth and POC production rates of *E. huxleyi* at high sunlight levels. Interaction of
rising $CO_2$ with light appears to affect differentially coccolithophores when grown
under different experimental setups.
Some previous studies have examined the effects of rising $CO_2$ and nutrient
concentrations on the physiology of *E. huxleyi* (Sciandra et al., 2003; Borchard et al.,
2011; Engel et al., 2014; Müller et al., 2017). Low nitrate or low phosphate
concentrations increased POC and PIC quotas in *E. huxleyi*, and these increases were
much less at high $CO_2$ than at low $CO_2$ levels (Matthiessen et al., 2012; Rouco et al.,
2013). In addition, rising $CO_2$ levels decreased growth rates at high phosphate
concentration, though it did not affect growth rates at low phosphate concentration
(Matthiessen et al., 2012). These studies indicate that fitness-relevant traits of *E.*





*huxleyi* may be altered in future high-$CO_2$ and low-nutrient oceans.
Recently, researchers have paid increasing attentions to combined effects of
multiple stressor on marine phytoplankton (Brennan and Collins, 2015; Boyd et al.,
2016; Hutchins and Fu, 2017), considering the fact that phytoplankton cells are
simultaneously exposed to physical and chemical factors. In addition, physiological
responses of phytoplankton to one environmental factor may be synergistically,
antagonistically or neutrally affected by others (Tong et al., 2016; Müller et al., 2017).
Even across a broad range of $CO_2$ concentrations, optimal $CO_2$ levels and maximal
values for growth rate, photosynthetic carbon fixation rate and calcification rate are
modulated by temperature and light intensity (Sett et al., 2014; Zhang et al., 2015).
Under chemostat cultures, rising $CO_2$ levels were found to increase the POC quotas
of a non-calcifying strain of *E. huxleyi* (PML 92A) and a calcifying strain of *E.*
*huxleyi* (PML B92/11) at low nutrient concentration and high light intensity
(Leonardos and Geider, 2005; Borchard et al., 2011). However, relatively few studies
have observed the interactive effects of multiple environmental factors on
physiological rates of coccolithophores. To investigate responses of the calcifying *E.*
*huxleyi* strain PMLB92/11 to multiple environmental factors, we employed dilute
batch cultures, investigated its growth, POC and PIC quotas, maximum ($F_v / F_m$) and
effective photochemical quantum yield ($F_v^{'} / F_m^{'}$) and electron transport rate (*ETR*) at
different levels of $CO_2$, light, dissolved inorganic nitrogen (DIN) and phosphate
concentrations (DIP).





## 2 Materials and methods

### 2.1 Experimental design

*Emiliania huxleyi* strain PML B92/11, one of the most commonly used strain in studies of *E. huxleyi*, was obtained from the culture collection at Plymouth. *E. huxleyi* was grown in diluted batch cultures (final cell concentrations were 20,000 to 130,000 cells mL$^{-1}$) at 20 $^{o}$C in a GXZ light chamber (Dongnan Instrument Company) under a 12 : 12 h light : dark cycle (light period: 8:00 a.m. to 8:00 p.m.). The synthetic seawater medium Aquil was prepared according to Sunda et al. (2005), added by 2200 μmol L$^{-1}$ bicarbonate (as opposed to 2380 μmol L$^{-1}$ in the original recipe), in order to reflect the alkalinity in the South and East China Seas of about 2200 μmol L$^{-1}$ (Chou et al., 2005; Qu et al., 2017). Initial dissolved inorganic nitrogen (DIN) and phosphate (DIP) concentrations in Aquil were 100 μmol L$^{-1}$ and 10 μmol L$^{-1}$, respectively (HNHP). For Aquil medium with low DIN concentration (LN), the synthetic seawater contained 8 μmol L$^{-1}$ $NO_3^-$ and 10 μmol L$^{-1}$ $PO_4^{3-}$, respectively. For low DIP treatment (LP), it had 100 μmol L$^{-1}$ $NO_3^-$ and 0.4 μmol L$^{-1}$ $PO_4^{3-}$.

Under each nutrient level, the Aquil media were aerated for 24 h at 20 $^{o}$C (PVDF 0.22 μm pore size, simplepure, Haining) with air containing 400 μatm or 1000 μatm $p$CO$_2$. The dry air/CO$_2$ mixture was humidified with double distilled water prior to the aeration to minimize evaporation. Then, the Aquil was sterilized by filtration (0.22 μm pore size, Polycap 75 AS, Whatman) and carefully pumped into autoclaved 500 mL polycarbonate bottles (Nalgene). The bottles were filled with Aquil leaving about 10



ml headspace to minimize gas exchange. Carbonate chemistry parameters (total
alkalinity (TA) and pH) were measured at the beginning and end of the experiment.
20 bottles at each $p$CO$_2$ level were incubated at light intensities of 80, 120, 200,
320, and 480 μmol photons m$^{-2}$ s$^{-1}$ of photosynthetically active radiation (PAR) (4
replicates each) measured using a PAR Detector (PMA 2132, Solar Light Company,
Glenside). A flow chart for the experimental treatments is presented in Fig. S1. For
the dilute batch cultures, initial cell concentration was 200 cells mL$^{-1}$ and cells were
acclimated to the experimental treatments for at least 14 generations before starting
the experiment (6 days at 80 μmol photons m$^{-2}$ s$^{-1}$, 5 days at 120 μmol photons m$^{-2}$
s$^{-1}$, and 4 days at 200–480 μmol photons m$^{-2}$ s$^{-1}$ at all nutrient conditions). Bottles
were rotated two times per day at 10:00 a.m. and 6:00 p.m. to make the cells can
obtain light homogeniously. To minimize changes in carbonate chemistry, final cell
concentrations were lower than 130,000 cells mL$^{-1}$, and changes in dissolved
inorganic carbon (DIC) concentrations were less than 10% (0.5%–9.1%).

**2.2 Nutrient concentrations, total alkalinity and pH$_T$ measurements**
Sampling started at 10:30 a.m. and finished at 12:00 a.m.. 50 mL samples for
determination of inorganic nitrogen and phosphate concentrations were
syringe-filtered (0.22 μm pore size, Haining) and measured using a scanning
spectrophotometer (Du 800, Beckman Coulter) according to Hansen and Koroleff

176 (1999).

Carbonate chemistry parameters were calculated from total alkalinity (TA) and pH$_T$





(total scale), phosphate, temperature, and salinity using the $CO_2$ System (Pierrot et al.,
2006). In the final days of incubation, 25 mL samples for TA measurements were
filtered (0.22 μm pore size, Syringe Filter) by gentle pressure with 200 mbar and
stored at 4 $^o$C for a maximum of 7 days. TA was measured at 20 $^o$C by potentiometric
titration (AS-ALK1+, Apollo SciTech) according to Dickson et al. 2003. Samples for
$pH_T$ measurements were syringe-filtered (0.22 μm pore size), and the bottles were
filled with overflow and closed immediately. The $pH_T$ was measured at 20 $^o$C with a
pH meter (Benchtop pH, Orion 8102BN) calibrated with an equimolal pH buffer (Tris
•HCl, Hanna) for sea water media (Dickson, 1993). Carbonic acid constants $K_1$ and
$K_2$ were calculated according to Roy et al. (1993).

**2.3 Measurements of photochemical parameters**
The effective photochemical quantum yield ($F_v^{'}/F_m^{'}$) and maximum photochemical
quantum yield ($F_v / F_m$) of photosystem II (PSII) were assessed using a XE-PAM
(Walz, Germany) at 1:00 p.m.. 3 ml samples were taken from the incubation bottles,
and $F_v^{'}/F_m^{'}$ values were measured immediately at active light intensities similar to
the incubation light levels. 3 mL samples were kept darkly for 15 min at 20 $^o$C, and $F_v$
$/ F_m$ values were determined at a measuring light intensity of 0.3 μmol photons m$^{-2}$ s$^{-1}$
and a saturation pulse of 0.8 s at light intensity of 5000 μmol photons m$^{-2}$ s$^{-1}$.
For electron transport rate (*ETR*) measurements, PAR levels were set between 1
μmol photons m$^{-2}$ s$^{-1}$ and 1600 μmol photons m$^{-2}$ s$^{-1}$ with 9 steps of 45 s each. The
*ETR* (mol e$^-$ g$^{-1}$ Chl *a* h$^{-1}$) was calculated according Dimier et al. (2009), *ETR* =



$(F_v^{'} / F_m^{'}) \times PAR \times 0.5 \times A$, where $A$ represent the cellular absorption value normalized
to Chl $a$, 0.5 implicits that 50% quanta of the absorbed PAR are distributed to PSII
(Dimier et al., 2009). Original $A$ value was about $2.47 \times 10^{-7}$ µmol e$^-$ cell$^{-1}$ s$^{-1}$ and
normalized $A$ value was about $8.40 \times 10^{-3}$ mol e$^-$ g$^{-1}$ Chl $a$ h$^{-1}$. Photosynthetic
response to irradiance (P-I curves) were analyzed according to Jasby and Platt (1976):
$ETR = ETR_{max} \times \tanh (alpha \times PAR / ETR_{max})$, where $ETR_{max}$ represents
light-saturated $ETR$, and $alpha$ is the slop of the P-I curve at limiting irradiance, $I_k$
calculated from the expression $ETR_{max} / alpha$ and represents the onset of light
saturation.

**2.4 Cell density measurements**

At the end of the incubation, about 25 ml samples were taken from the incubation
bottles at about 2:30 p.m.. Cell densities were measured by using a Particle Counter
(Beckman). Growth rate (μ) was calculated according to the equation: $\mu = (\ln N_1 - \ln$
$N_0) / d$, where $N_0$ is 200 cells mL$^{-1}$ and $N_1$ is the cell concentration in the final days of
experiment, and $d$ is the growth time span in days.

**2.5 Particulate organic (POC) and inorganic carbon (PIC) measurements**

GF/F filters, pre-combusted at 450 $^o$C for 8 h, were used to filter the samples of total
particulate carbon (TPC) and particulate organic carbon (POC). TPC and POC
samples were stored darkly at –20$^o$C. For POC measurements, samples were fumed
with HCl for 12 h to remove inorganic carbon, and samples for TPC measurements



were not treated with HCl. All samples were dried at 60 $^{o}$C for 12 h, and analyzed
using a Perkin Elmer Series II CHNS/O Analyzer 2400 instrument (Perkin Elmer
Waltham, MA). Particulate inorganic carbon (PIC) quota was calculated as the
variance between TPC quota and POC quota. POC and PIC production rates were
calculated by multiplying their contents with $\mu$ (d$^{-1}$), respectively.

**2.6 Data analysis**
Responses of growth rates, POC and PIC quotas, PIC:POC ratio, POC and PIC
production rates to incubation light intensities were fitted using the model provided by
Eilers and Peeters (1988): $y = \dfrac{PAR}{a \times PAR^2 + b \times PAR + c}$, where the parameters $a$, $b$ and $c$
are fitted in a least square manner. The apparent light use efficiency, the slope ($\alpha$), for
each light response curve was estimated as $\alpha = 1/c$.
A three-way ANOVA was used to determine the main effect of dissolved inorganic
nitrate (or phosphate), $p$CO$_2$, light intensity and their interactions for these variables.
A three-way ANOVA was performed to compare the fitted $\alpha$ between growth, POC
and PIC production rates at low and high CO$_2$ levels under different nutrient
conditions. When necessary, a Tukey Post hoc test was used to identify the differences
between two CO$_2$, nitrate (or phosphate) or light levels. A Shapiro-Wilk's test was
conducted to test residual normality and a Levene test was used to test for variance
homogeneity of significant data. Statistical analysis was conducted by using R and
significant level was set at $p < 0.05$.





**3 Results**

**3.1 Dissolved inorganic nitrogen and phosphate concentrations, and carbonate**
**chemistry paremeters**
At the HNHP condition, dissolved inorganic nitrogen (DIN) and phosphate (DIP)
concentrations were $101 \pm 1.1$ µmol $L^{-1}$ and $10.5 \pm 0.2$ µmol $L^{-1}$, respectively, at the
beginning of the experiments, and were $92.8 \pm 1.6$ µmol $L^{-1}$ and $9.7 \pm 0.2$ µmol $L^{-1}$ in
the final days of the experiment (Table S1). At the LN condition, DIN concentrations
were $8.8 \pm 0.1$ µmol $L^{-1}$ at the beginning of the experiment and $1.0 \pm 0.4$ µmol $L^{-1}$ at
the end of the experiment. In the LP treatment, DIP concentrations were $0.4 \pm 0.1$
µmol $L^{-1}$ at the beginning of the experiment, and below the detection limit ($< 0.04$
µmol $L^{-1}$) at the end of the experiment.
The carbonate system parameters (mean values for the beginning and end of
incubations) are shown in Table 1. For low $CO_2$ (LC) condition, the $pCO_2$ levels of the
media were about 435 µatm at HNHP, 410 µatm at LN and 370 µatm under LP
conditions, and the $pH_T$ values (reported on the total scale) were about 8.10 at HNHP,
8.11 at LN and 8.16 at LP. For high $CO_2$ (HC) condition, the $pCO_2$ levels of the media
were about 970 µatm at HNHP, 935 µatm at LN and 850 µatm at LP, and the $pH_T$
values were about 7.80 at HNHP, 7.80 at LN, and 7.85 at LP conditions. Average
$pCO_2$ levels for all LC conditions were 410 µatm, and for all HC conditions were 925
µatm.



### 3.2 Growth rate

Under each nutrient condition, at both LC and HC, growth rates of *E. huxleyi* increased with elevated light intensity up to 200 μmol photons m$^{-2}$ s$^{-1}$ and significantly declined thereafter (Three-way ANOVA; Tukey Post hoc, all df = 2, all *p* < 0.001) (Fig. 1; Table 2). Compared with LC, growth rates at HC were 2%–7% lower at HNHP (*p* < 0.05), 5%–9% lower at LN (*p* < 0.01) and 3%–24% lower at LP (*p* < 0.01), respectively (Table 3). Under LP treatment, HC-induced reduction of growth rate was larger at higher light levels (Fig. 1c).

At LC, growth rate at LN was similar with that at HNHP under limited light intensity with 80 μmol photons m$^{-2}$ s$^{-1}$ (df = 1, *p* = 0.82), and significantly lower than at HNHP under optimal and supra-optimal light intensities (both df = 1, *p* < 0.01 for 200 treatment; *p* = 0.005 for 480 treatment). At HC, growth rates at LN were significantly lower than those at HNHP under limited, optimal and supra-optimal light intensities (all df = 1, *p* < 0.01 for 80, 200, 480 treatments).

At LC and at 80 μmol photons m$^{-2}$ s$^{-1}$, growth rate at LP was lower than at HNHP (df = 1, *p* < 0.001); while at 120–480 μmol photons m$^{-2}$ s$^{-1}$, growth rates were no significant differences between LP and HNHP (all df = 1, all *p* > 0.1) (Fig. 1; Table 3). At HC and at 80, 120 and 480 μmol photons m$^{-2}$ s$^{-1}$, growth rates were significantly lower at LP than at HNHP; at 200 and 320 μmol photons m$^{-2}$ s$^{-1}$, growth rates were not significantly different between LP and HNHP (both df = 1, both *p* > 0.05).

### 3.3 POC quota





Under HNHP or LP conditions, at LC, POC quotas were not significantly different
among 80, 120 and 200 μmol photons $m^{-2}$ $s^{-1}$ and increased with increased light
intensity from 200 to 480 μmol photons $m^{-2}$ $s^{-1}$ (Three-way ANOVA; Tukey Post hoc,
both df = 1, both $p < 0.01$); while at HC, POC quotas increased with elevated light
intensity up to 480 μmol photons $m^{-2}$ $s^{-1}$ (Fig. 2a,c; Tables 2; 3). At LN, at both LC
and HC, POC quotas at 320 μmol photons $m^{-2}$ $s^{-1}$ were significantly larger than at
other light intensities (Fig. 2b).
At HNHP or at LN, POC quotas did not show significant differences between HC
and LC (Fig. 2a,b). At LP, at 80 μmol photons $m^{-2}$ $s^{-1}$, POC quotas were significantly
larger at LC than at HC (df = 1, $p = 0.003$), while at 480 μmol photons $m^{-2}$ $s^{-1}$, they
were lower (df = 1, $p = 0.001$).
At both LC and HC, POC quotas were not significantly different between LN and
HNHP at 80–320 μmol photons $m^{-2}$ $s^{-1}$, while they were lower at LN than at HNHP at
480 μmol photons $m^{-2}$ $s^{-1}$ ($p < 0.01$). At both LC and HC, POC quotas were not
significantly different between LP and HNHP at 80–480 μmol photons $m^{-2}$ $s^{-1}$ (all df
= 1, all $p > 0.05$).

**3.4 PIC quota**
At HNHP or at LN, under either LC or HC, PIC quotas increased with increasing light
intensity until 320 μmol photons $m^{-2}$ $s^{-1}$ (Three-way ANOVA; Tukey Post hoc, all df
= 1, all $p < 0.001$) and then leveled off with further increasing light intensity (Fig.
2d,e; Tables 2; 3). At LP under LC conditions, PIC quotas increased significantly





when light intensity increased from 80 to 200 μmol photons m$^{-2}$ s$^{-1}$ and significantly
declined thereafter (both df = 1, both $p < 0.001$) (Fig. 2f), while at LP and HC, there
were no significant differences among the light levels (all $p > 0.05$).

At HNHP or at LN, PIC quotas were larger at LC than at HC (all df = 1, all $p >$

0.05 at 80, 120, 200 treatments; both $p < 0.01$ at 320 and 480 treatments) (Fig. 2d,e).
Under LP conditions at 200 and 320 μmol photons m$^{-2}$ s$^{-1}$, PIC quotas were larger at
LC than at HC (both df = 1, both $p < 0.05$) (Fig. 2f).

At both LC and HC, PIC quotas were larger at LN than at HNHP (all df = 1, all $p >$

0.05 at 80 treatment; $p < 0.05$ at 120–480 treatments) (Fig. 2d,e). For both LC and HC
conditions at 80–200 μmol photons m$^{-2}$ s$^{-1}$, PIC quotas were larger at LP than at
HNHP (all df = 1, all $p < 0.05$), while at 320 and 480 μmol photons m$^{-2}$ s$^{-1}$, they were
not significantly different between LP and HNHP (Fig. 2f).

**3.5 PIC:POC ratio**

At HNHP under LC, PIC:POC ratio increased with elevated light intensity until 320
μmol photons m$^{-2}$ s$^{-1}$ and significantly declined thereafter (Three way ANOVA,
Tukey Post hoc, df = 1, $p < 0.05$) (Fig. 2g; Tables 2; 3), while at HC, they were not
significantly different between light treatments (all $p > 0.05$). At LN in both LC and
HC treatments, PIC:POC ratio increased when light intensity increased from 80 to
200 μmol photons m$^{-2}$ s$^{-1}$ and were not significantly different between 200, 320 and
480 μmol photons m$^{-2}$ s$^{-1}$ (Fig. 2h). At LP under LC conditions, PIC:POC ratio
increased with increasing light intensity until 200 μmol photons m$^{-2}$ s$^{-1}$, and declined



with further increasing light intensity (both df = 1, both $p < 0.05$) (Fig. 2i), while at
HC, they were not significantly different between light treatments (df = 4, $p > 0.05$).
At either HNHP or at LP, at light levels of 80–480 µmol photons m$^{-2}$ s$^{-1}$, PIC:POC
ratio were not significantly different between LC and HC (all df = 1, all $p > 0.05$) (Fig.
2g,i). At LN under 320 and 480 µmol photons m$^{-2}$ s$^{-1}$, PIC:POC ratios were larger at
LC than at HC (both df = 1, both $p < 0.05$) (Fig. 2h).
At both LC and HC, under 80–480 µmol photons m$^{-2}$ s$^{-1}$ PIC:POC ratios were
larger at LN than at HNHP (all df = 1, $p > 0.05$ at the 80 treatment; $p < 0.05$ at the 120
to 480 treatments) (Fig. 2g,h). In both LC and HC conditions, at 80–200 µmol
photons m$^{-2}$ s$^{-1}$ PIC:POC ratios were larger at LP than at HNHP (all df = 1, all $p <$
0.05) (Fig 2g,i), while at 320 and 480 µmol photons m$^{-2}$ s$^{-1}$, they were not
significantly different between LP and HNHP.

**3.6 $F_v/F_m$ and $F_v^{'}/F_m^{'}$**
$F_v/F_m$ and $F_v^{'}/F_m^{'}$ showed the same patterns (Fig. 3). At each nutrient condition, at
both LC and at HC, $F_v/F_m$ and $F_v^{'}/F_m^{'}$ decreased with elevated light intensity until
480 µmol photons m$^{-2}$ s$^{-1}$ (Three way ANOVA; Tukey Post hoc, all df = 1, all $p <$
0.01) (Fig. 3a–f; Tables 2; 3).
At either HNHP or LP, only at 480 µmol photons m$^{-2}$ s$^{-1}$ $F_v/F_m$ values were
significantly larger at LC than at HC (both df = 1, both $p < 0.01$) (Fig. 3a,c). At LN in
the light range of 80–480 µmol photons m$^{-2}$ s$^{-1}$, $F_v/F_m$ values were not significantly
different between LC and HC (all df = 1, all $p > 0.05$) (Fig. 3b).
At both LC and HC, from 80 to 480 µmol photons m$^{-2}$ s$^{-1}$ $F_v/F_m$ did not show



significant differences between LN and HNHP (all df = 1, all $p > 0.05$), and at 480
µmol photons m$^{-2}$ s$^{-1}$, they were lower at LP than at HNHP (both df = 1, both $p < 0.05$)
(Fig. 3a,c).

At HNHP from 80 to 480 µmol photons m$^{-2}$ s$^{-1}$, $F_v'/F_m'$ values were similar

between LC and HC (all df = 1, all $p > 0.05$) (Fig. 3d). At LN under 200 µmol
photons m$^{-2}$ s$^{-1}$, and at LP under 480 µmol photons m$^{-2}$ s$^{-1}$, $F_v'/F_m'$ values were
larger at LC than at HC (both df = 1, both $p < 0.01$) (Fig. 3e,f).

At LC under 200 µmol photons m$^{-2}$ s$^{-1}$, $F_v'/F_m'$ values were significantly larger at

LN than at HNHP, as well as at LP compared to HNHP (both df = 1, both $p < 0.05$)
(Fig. 3d,e,f). At HC under 480 µmol photons m$^{-2}$ s$^{-1}$ $F_v'/F_m'$ values were
significantly lower at LP than at HNHP (df = 1, $p < 0.01$) (Fig. 3d,f).

**3.7 $ETR_{max}$**
At HNHP and at LC, $ETR_{max}$ increased significantly with increasing light intensities
until 200 µmol photons m$^{-2}$ s$^{-1}$ (df = 1, $p < 0.01$), and leveled off with further
increasing light intensities (Fig. 3g; Tables 2; 3). At HNHP and at HC, with light
intensities increasing from 80 to 120 µmol photons m$^{-2}$ s$^{-1}$, $ETR_{max}$ increased
remarkably (df = 1, $p < 0.01$), and declined significantly when light intensities further
increased to 480 µmol photons m$^{-2}$ s$^{-1}$ (df = 1, $p < 0.05$). At LN or at LP, under both
LC and HC, $ETR_{max}$ increased with increasing light intensities until 200 µmol photons
m$^{-2}$ s$^{-1}$ and declined thereafter (all df = 1, all $p < 0.01$) (Fig. 3h,i).

At HNHP and only at 480 µmol photons m$^{-2}$ s$^{-1}$, $ETR_{max}$ was lower at HC than at





LC (df =1, $p < 0.01$) (Fig. 3g; Table 3). At LN across the light range of 80–480 μmol
photons m$^{-2}$ s$^{-1}$, $ETR_{max}$ values were similar between HC and LC (Fig. 3h). At LP
under 320 μmol photons m$^{-2}$ s$^{-1}$, $ETR_{max}$ was larger at HC than at LC; while at 480
μmol photons m$^{-2}$ s$^{-1}$, they were lower (both df =1, both $p < 0.05$) (Fig. 3i).
At both LC and HC from 80–480 μmol photons m$^{-2}$ s$^{-1}$, $ETR_{max}$ values were larger
at LN than at HNHP (Tukey Post hoc, all df = 1, $p < 0.01$ for the 120, 200 and 320
treatments at LC; $p > 0.05$ for the 80 and 480 treatments at LC; $p < 0.01$ for the 80,
200, 320 and 480 treatments at HC; $p > 0.05$ for the 120 treatment at HC) (Fig. 3g,h).
At LC under 80 μmol photons m$^{-2}$ s$^{-1}$, $ETR_{max}$ was lower at LP than at HNHP (df = 1,
$p > 0.1$); while at 120–480 μmol photons m$^{-2}$ s$^{-1}$, $ETR_{max}$ values were larger (Tukey
Post hoc, all df = 1, $p > 0.05$ for 120, 320 and 480 treatments; $p < 0.01$ for 200
treatment) (Fig. 3g,h). At HC under 80 and 120 μmol photons m$^{-2}$ s$^{-1}$, $ETR_{max}$ values
were lower at LP than at HNHP (Tukey Post hoc, both df = 1, $p > 0.1$ for the 80
treatment; $p < 0.01$ for the 120 treatment), while at 200–480 μmol photons m$^{-2}$ s$^{-1}$,
they were larger (Tukey Post hoc, all df = 1, $p < 0.01$ for 200 and 320 treatments; $p >$
0.1 for 480 treatment).

**3.8 Apparent light use efficiency ($\alpha$) for growth, POC and PIC production rates**
At each nutrient condition, $\alpha$ values of fitted curves of growth, POC and PIC
production rates were not significantly different between LC and HC, with the
exception of $\alpha$ of PIC production rate at LP (df = 1, $p < 0.05$) (Fig. 4).
At HNHP under both LC and HC, $\alpha$ values of fitted curves for POC and PIC

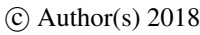



production rates were not significantly different, and they were significantly larger
than those for growth rates (both df = 1, both $p < 0.01$) (Fig. 4a). At LN under both
LC and HC, and at LP under LC, $\alpha$ values for PIC production rates were larger than
those for POC production rates, which were larger than those for growth rates (all df =
1, all $p < 0.01$) (Fig. 4b,c). At LP and HC, $\alpha$ values for POC and PIC production rates
did not show significant differences and they were larger than that for growth rates
(Fig. 4c).

At both LC and at HC, $\alpha$ values of fitted curves of growth rates or POC production

rates were not significantly different between LN and HNHP, and between LP and
HNHP (Fig. 4). At LC, $\alpha$ values for PIC production rates were lower at HNHP than at
LN or at LP (both df = 1, both $p < 0.01$); at HC, they were not significantly different
between HNHP and LP (Fig. 4).

**4 Discussion**

In this study, we investigated synergistic negative effects of low nutrient
concentrations and rising $p\mathrm{CO_2}$ on growth rates, especially at limiting low and
inhibiting high light intensities. Notably, high light intensities compensated for
inhibition of LP on growth rates at LC. LN reduced POC quota and its sensitivity to
light intensity. Both LN and LP increased PIC quotas, PIC:POC ratio, and *ETR*
efficiency.



### 4.1 Low nutrient concentrations and high $p\mathrm{CO_2}$ level synergistically reduce growth rate.

Langer et al. (2013) detected that cell numbers on the fourth to sixth days during cultures were in the exponential growth phase even at 3 μmol L$^{-1}$ NO$_3^-$ or at 0.29 μmol L$^{-1}$ PO$_4^{3-}$ with the same *E. huxleyi* strain. In addition, other *E. huxleyi* strains were in the exponential phase of growth on the fourth to the seventh days in the cultures with 2.5–8 μmol L$^{-1}$ NO$_3^-$ or at 0.4–0.55 μmol L$^{-1}$ PO$_4^{3-}$ (Perrin et al., 2016; Rokitta et al., 2016). All parameters were measured on the fourth to the sixth days, and it is most likely that cells at all treatments were sampled in the exponential growth phase in this study.

Less energy availability limited growth rates of *E. huxleyi* at lower light intensities, while reduction in growth rates at high light intensities could be related to photooxidative damage or photoinhibition (Fig. 1), because high light intensity can constantly damage the reaction centers of photosystem II (PSII) of *E. huxleyi* (Fig. 3a–f) (Ragni et al., 2008). Nevertheless, photoinhibition was not observed in electron transport rate (*ETR*) of the cells grown at 480 μmol photons m$^{-2}$ s$^{-1}$ even exposed to light intensity of 1600 μmol photons m$^{-2}$ s$^{-1}$ (Figs. 1 and S3). This implies that the photochemical performance during a short time exposure can hardly reflect the growth response. At HC, the negative effect of high [H$^+$] on growth rate was larger than positive effects of increased CO$_2$ and HCO$_3^-$ concentrations, which could be attributed to lower growth rates at HC than at LC (Fig. 1) (Bach et al., 2011).

Based on measured PON quota and cell concentration in this study (Figs. 1 and S6), PON concentrations at the end of incubations were estimated to be 7.8–9.3 μmol L$^{-1}$



at different nutrient conditions (Table S1). These were closely correlated with molar
drawdown of dissolved inorganic nitrate (DIN) in the cultures. *E. huxleyi* appeared to
be a poor competitor for inorganic nitrate under low levels of nitrate availability (Fig.
1). Reduced levels of gene expressions and nitrate reductase (NRase) activity in *E.*
*huxleyi* cells grown under low nitrate could be responsible (Bruhn et al., 2010; Rouco
et al., 2013), thus resulting in reduced nitrate assimilation. In addition, LN
concentration was shown to down-regulate transcripts of genes related to synthesis of
amino acids, RNA polymerases and nitrogen metabolism in *E. huxleyi* (Rokitta et al.,
2014), which led to lower overall biosynthetic activity and decreased the growth rates
(Fig. 1).
Synergistic effects of LN and HC on growth rates indicate that these conditions
may inhibit cellular metabolic activity simultaneously (Fig. 1) (Sciandra et al., 2003).
In fact, intracellular $[H^+]$ have been reported to be higher in HC-grown than in
LC-grown *E. huxleyi* cells (Suffrian et al., 2011). To transport extra $H^+$ out of cells, *E.*
*huxleyi* at HC need more transporters and energy, but LN is likely to limit the
synthesis of these transporters and energy supply, therefore, it exacerbated the
negative effects of high $[H^+]$ on growth of *E. huxleyi* (Fig. S6) (Bruhn et al., 2010).
*E. huxleyi* possesses an exceptional phosphorus acquisition capacity, which could
allow it to dominate in phosphate-limiting environments (Dyhrman and Palenik,
2003). In this study, at low levels of light intensity, uptake of phosphate could be
energy limited, thus their growth was more inhibited at LP (Fig. 1c). Under light
saturation condition, relationship of growth rates of *E. huxleyi* with phosphate



concentrations indicated a very high affinity for dissolved inorganic phosphate (DIP)
with 0.04 μmol L$^{-1}$ half-saturation constant for DIP (Fig. 5). Since LP was reported to
enhance expression of gene with a role in phosphorus assimilation or metabolism and
synthesis of inorganic $PO_4^{3-}$ transporters (Dyhrman et al., 2006; McKew et al., 2015;
Rokitta et al., 2016), which allowed *E. huxleyi* to take up $PO_4^{3-}$ efficiently enough, so
that LP did not result in reduced growth rate at LC in this study (Fig. 1). Rokitta et al.
(2016) showed that even $PO_4^{3-}$ concentration in the culture media declined to zero
(undetectable), cell number sustained to increase for 4 days, indicating that *E. huxleyi*
cells could store $PO_4^{3-}$ and use them later. Consequently, high affinity, efficient uptake
and storage capacity for $PO_4^{3-}$ in *E. huxleyi* could account for no significant
differences in growth rates between LP and HNHP under LC and saturating and
supra-optimal light intensities. In fact, as reported previously, higher growth rates of *E.*
*huxleyi* at LP in comparison to HP were found during exponential growth phase in
batch cultures (Rokitta et al., 2016). In natural waters, *E. huxleyi* usually starts to
bloom following diatom blooms (Tyrrell and Merico, 2004). Therefore, our results
also indicate that high growth rate of *E. huxleyi* at low nutrients concentrations may
drive the succession of diatom to *E. huxleyi*.
Rising $CO_2$ was found to lead to higher phosphorous requirements for growth,
carbon fixation and nitrogen uptake, and to decrease alkaline phosphate (APase)
activity in *E. huxleyi* (Matthiessen et al., 2012; Rouce et al., 2013). At HC, higher
phosphorous requirements may lead to lower growth rates at LP in comparison to
HNHP (Fig. 1a,c). In addition, elevated $CO_2$ concentrations can down-regulate the





uptake capacity of the cells for $CO_2$ and/or $HCO_3^-$ ($CO_2$ concentration mechanisms),
which could lead to less energy cost for maintaining active uptake mechanisms (Gao
et al., 2012), and the save energy in the HC-grown cells, consequently, might have
exacerbates photo-inhibition, leading to higher inhibition of the growth under LP and
high light intensities (Fig. 1c).

**494 4.2 Low dissolved inorganic nitrogen concentration and high $p$CO$_2$ level**

**495 synergistically reduce POC quota**

At LC, *E. huxleyi* mainly uses external $HCO_3^-$ as an inorganic carbon source for
photosynthesis and calcification, and increasing light intensities are able to increase
$HCO_3^-$ uptake rates (Kottmeier et al., 2016). This may explain why POC and PIC
quotas and production rates increased with increasing light intensity (Figs. 2 and S5).
HC down-regulates gene expression related to the $HCO_3^-$ transporter (Rokitta et al.,
2012) and decreases the $HCO_3^-$ uptake rate in *E. huxleyi* (Kottmeier et al. 2016),
leading to lower PIC quotas at HC than at LC (Fig. 2). Meanwhile, cells at HC can
increase $CO_2$ uptake to compensate for low-$HCO_3^-$-uptake for photosynthetic C
fixation (Kottmeier et al., 2016), explaining the similar POC quotas between HC and
LC (Fig. 2a–c).
LN down regulates expression of the *rbc*L gene coding for the large subunit of the
ribulose-1,5-biphosphate carboxylase/oxygenase (RUBISCO) (Bruhn et al., 2010;
Rokitta et al., 2014). To conserve nitrogen, cells at LN prefer to shut down the
synthesis of RUBISCO and then reduce carbon fixation (Falkowski et al., 1989) (Fig.





2b). At HC, lower cell division rates resulted in lower POC and PIC production rates
than at LC (Fig. S5).

**4.3 Low nutrient concentrations facilitate calcification and maximum electron**
**transport rates ($ETR_{max}$)**
Müller et al. (2008) found that calcification (PIC production) occurred only in the G1
cell cycle phase, and that LN or LP held cells in the G1 phase longer, which led to
larger PIC quotas and calcification rates at LN or at LP than at HNHP (Figs. 2 and S5).
LC and LP treatment decreased cell division rates, elongated cell cycle, and increased
coccolith production of *E. huxleyi* in the darkness (Paasche and Brubak, 1994). In the
present work, however, we found slightly faster cell division (growth) and identical
calcification rates at LP and high light intensities (Figs. 1c, 2f and S5). LP has been
shown to up-regulate the genes involved in calcium binding proteins such as the
glutamic acid related to synthesize of coccolith, calcium homeostasis and
transcription factor (*cmyb*) (Wahlund et al., 2004; Dyhrman et al., 2006), and
facilitates the formation of cytoplasmic membrane bodies (Shemi et al., 2016). These
are related to the pathways associated with production of coccoliths (Young and
Henriksen, 2003) and may also be responsible for larger PIC quotas at LP.
Calcification of coccolithophores makes an important contribution to marine
carbonate counter pumps in the pelagic ocean (Rost and Riebesell, 2004). Enhanced
calcification of *E. huxleyi* at low nutrient concentrations implies that blooms of
calcifying *E. huxleyi* diminish the potential of the oceanic $CO_2$ uptake compared to
non-calcifying phytoplankton blooms. On the other hand, larger PIC:POC ratios

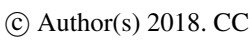



imply faster sinking rate of *E. huxleyi* cells, facilitating the export of carbon into
deeper waters (Hoffmann et al., 2015).

At low light intensities, the $ETR_{max}$ values were severely limited by low energy

input. Supraoptimal light intensities have been found to significantly reduce the
abundance of several proteins involved in repair and assembly of PSII, such as repair
of photodamaged Psb D1 proteins in the reaction center of PSII of *E. huxleyi* (McKew
et al., 2013). These suggest that high light intensity is likely to do great damage to the
PSII structure and then reduce the $ETR_{max}$. Especially at HC, supraoptimal light
intensity and saved energy from down-regulation of CCM activity synergistically
decreased $ETR_{max}$ (Fig. 3).

A previous study found that calcification can be an additional sink for electrons in

*E. huxleyi* (Xu and Gao 2012). Compared with HNHP, larger $ETR_{max}$ at LN or at LP
and at saturating light intensities likely resulted from larger calcification rates (Figs. 2
and 3). On the other hand, growth, photosynthetic carbon fixation and nitrogen uptake
need energy originating from electron transport (Zhang et al., 2015). At LP and at
limiting levels of light intensity, lower growth, photosynthetic carbon and nitrate
assimilation rates coincided with lower $ETR_{max}$ (Figs. 1–3), implying correlations of
these physiological processes.

To provide organic carbon fixed by photosynthesis to support growth and other

metabolic processes, cells need to maintain larger light-use efficiency ($\alpha$) for POC
production rates (Fig. 4). Calcification is an energy-dependent process (Riebesell and
Tortell, 2011), and increased calcification rates at low nutrient concentrations could be



aided by higher light-use efficiencies (Fig. 4). In addition, besides taking up inorganic
carbon sources and $Ca^{2+}$ from the seawater to calcify, cells need extra energy to expel
$H^+$ generated during calcification from the cells (Jin et al., 2017), these may also
account for higher light-use efficiencies for PIC production rates.

Nutrient availability, $CO_2$ level and light intensity significantly interacted to affect

growth rate, POC and PIC quotas, $F_v / F_m$, $F_v^{'} / F_m^{'}$ and $ETR_{max}$ (Table 2). Obviously,
the question how growth, carbon fixation and calcification rates of *E. huxleyi* would
respond to ocean global changes needs to be examined under multiple stressors and
under natural environmental variations (Feng et al., 2008, 2017). In comparison to the
current ocean environment, under HC and HL conditions as expected in future oceans,
effects of LN and LP on carbon fixation of *E. huxleyi* may partly negate each other
(Fig.2, Table 3). Although both HC and HL reduced calcification rates of *E. huxleyi*,
low nutrient concentrations showed dominant positive effects on PIC quota or
calcification (Fig. 2d−f), suggesting that calcification of *E. huxleyi* may increase in the
future pelagic oceans. Our study demonstrates that complex effects of multiple
environmental drivers on phytoplankton require us to investigate the underlying
mechanisms of these interactions, in order to comprehend how ecological and
biogeochemical functions of key phytoplankton groups may respond to ocean global
changes.







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

was supported by MEL's visiting scientists program.












**Figure Legends**

**Figure 1.** Growth rate of *Emiliania huxleyi* as a function of light intensities at low $pCO_2$ (LC) and high $pCO_2$ levels (HC) at high dissolved inorganic nitrogen (DIN) and phosphate (DIP) concentrations (HNHP)(**a**), low DIN and high DIP concentrations (LN) (**b**), or high DIN and low DIP concentrations (LP) (**c**). The solid lines in each panel were fitted using the model provided by Eilers and Peeters (1988). The values represent the mean ± standard deviation for four replicates.

**Figure 2.** At both LC and HC, POC quotas of *E. huxleyi* as a function of light intensities at HNHP (**a**), LN (**b**) and LP (**c**) conditions. At both LC and HC, light responses of PIC quotas of *E. huxleyi* at HNHP (**d**), LN (**e**) and LP (**f**) conditions. At both LC and HC, light responses of PIC:POC ratios of *E. huxleyi* at HNHP (**g**), LN (**h**) and LP (**i**) conditions. The solid lines in each panel were fitted using the model provided by Eilers and Peeters (1988). The values represent the mean ± standard deviation for four replicates.

**Figure 3.** At both LC and HC, maximum photochemical quantum yields ($F_v/F_m$) of *E. huxleyi* as a function of light intensities at HNHP (**a**), LN (**b**) and LP (**c**) conditions. At both LC and HC, light responses of effective photochemical quantum yields ($F_v^{'}/F_m^{'}$) of *E. huxleyi* at HNHP (**d**), LN (**e**) and LP (**f**) conditions. At both LC and HC, light responses of fitted maximum electron transport rate ($ETR_{max}$) of *E. huxleyi* at HNHP (**g**), LN (**h**) and LP (**i**) conditions. The values represent the mean ± standard





deviation for four replicates.

**Figure 4.** At both LC and HC, apparent light-use efficiency ($\alpha$) for growth, POC and
PIC production rates of *E. huxleyi* at HNHP (**a**), LN (**b**) and LP (**c**) conditions. $\alpha$ was
the slope of fitted lines for growth, POC and PIC production rates. $\mu$ represents
growth rate, POCpro represents POC production rate and PICpro represents PIC
production rate. Different letters showed statistically difference. The values represent
the mean $\pm$ standard deviation for four replicates.

**Figure 5.** Growth rate of *E. huxleyi* as a function of dissolved inorganic phosphate
(DIP) concentrations at LC under 200 $\mu$mol photons $m^{-2}$ $s^{-1}$. DIN concentration was
100 $\mu$mol $L^{-1}$ in all culture media, and DIP concentrations were set up to 0 $\mu$mol $L^{-1}$,
0.25 $\mu$mol $L^{-1}$, 0.5 $\mu$mol $L^{-1}$, 1.5 $\mu$mol $L^{-1}$, 3 $\mu$mol $L^{-1}$ and 10 $\mu$mol $L^{-1}$ in the culture
media. All samples were incubated at 200 $\mu$mol photons $m^{-2}$ $s^{-1}$ and at LC for 4 days.
Solid line was fitted using the Michaelis-Menten equation. The values represent the
mean $\pm$ standard deviation for four replicates.










**Table 1.** Carbonate chemistry parameters (mean values for the beginning and end of incubations) of the media at different nutrient conditions. TA and pH samples were collected and measured before and in the final days of the experiment.

| | $p\mathrm{CO_2}$ ($\mu$atm) | pH (total scale) | TA ($\mu$mol $\mathrm{L^{-1}}$) | DIC ($\mu$mol $\mathrm{L^{-1}}$) | $\mathrm{HCO_3^-}$ ($\mu$mol $\mathrm{L^{-1}}$) | $\mathrm{CO_3^{2-}}$ ($\mu$mol $\mathrm{L^{-1}}$) | $\mathrm{CO_2}$ ($\mu$mol $\mathrm{L^{-1}}$) | $\Omega$ calcite |
|---|---|---|---|---|---|---|---|---|
| HNHP | 435±56[a] | 8.10±0.05[a] | 2225±22[a] | 1970±26[a] | 1778±37[a] | 178±17[a] | 14±2[a] | 4.3±0.4[a] |
| | 970±157[b] | 7.80±0.06[b] | 2223±22[a] | 2100±24[b] | 1970±29[b] | 99±14[b] | 31±5[b] | 2.4±0.3[b] |
| LN | 410±52[a] | 8.11±0.04[a] | 2139±47[a] | 1888±60[a] | 1700±65[a] | 172±10[a] | 13±2[a] | 4.1±0.2[a] |
| | 936±143[b] | 7.80±0.05[b] | 2154±41[a] | 2034±55[b] | 1908±58[b] | 96±10[b] | 30±5[b] | 2.3±0.2[b] |
| LP | 372±26[a] | 8.16±0.02[a] | 2225±25[a] | 1950±27[a] | 1740±30[a] | 198±8[a] | 12±1[a] | 4.7±0.2[a] |
| | 852±158[b] | 7.85±0.06[b] | 2226±21[a] | 2092±28[b] | 1954±34[b] | 110±15[b] | 28±5[b] | 2.7±0.4[b] |

HNHP, 101 $\mu$mol $\mathrm{L^{-1}}$ dissolved inorganic nitrogen (DIN) and 10.5 $\mu$mol $\mathrm{L^{-1}}$ dissolved inorganic phosphate (DIP); LN, 8.8 $\mu$mol $\mathrm{L^{-1}}$ DIN; LP, 0.4 $\mu$mol $\mathrm{L^{-1}}$ DIP. Different letters indicate statistical difference between two $p\mathrm{CO_2}$ treatments (Tukey Post hoc, $p < 0.01$). The values are expressed as mean values ± SD calculated from measurements before and in the final days of incubations.





**Table 2.** Results of three-way ANOVAs of the impacts of dissolved inorganic nitrate
(DIN) or phosphate (DIP) concentrations, $p\text{CO}_2$, light intensity and their interaction
on growth rate, POC and PIC quotas, PIC:POC ratio, $F_v/F_m$, $F_v^{'}/F_m^{'}$ and $ETR_{max}$.

| | Factor | $F$ value | $p$ value | Factor | $F$ value | $p$ value |
|---|---|---|---|---|---|---|
| Growth rate (d$^{-1}$) | N | 215.9 | <0.001 | P | 1015.5 | <0.001 |
| | C | 547.8 | <0.001 | C | 213.3 | <0.001 |
| | L | 1330.4 | <0.001 | L | 1863.8 | <0.001 |
| | N×C | 9.1 | =0.004 | P×C | 147.6 | <0.001 |
| | N×L | 11.8 | <0.001 | P×L | 274.4 | <0.001 |
| | C×L | 18.3 | <0.001 | C×L | 11.1 | <0.001 |
| | N×C×L | 4.1 | =0.006 | P×C×L | 19.7 | <0.001 |
| POC quota (pg C cell$^{-1}$) | N | 27.1 | <0.001 | P | 13.7 | <0.001 |
| | C | 0.6 | =0.435 | C | 0.1 | =0.731 |
| | L | 34.7 | <0.001 | L | 103.2 | <0.001 |
| | N×C | 13.2 | <0.001 | P×C | 14.5 | <0.001 |
| | N×L | 17.9 | <0.001 | P×L | 0.4 | =0.780 |
| | C×L | 1.0 | =0.432 | C×L | 21.6 | <0.001 |
| | N×C×L | 1.9 | =0.125 | P×C×L | 7.3 | <0.001 |
| PIC quota (pg C cell$^{-1}$) | N | 544.0 | <0.001 | P | 619.1 | <0.001 |
| | C | 70.5 | <0.001 | C | 105.8 | <0.001 |
| | L | 71.2 | <0.001 | L | 55.3 | <0.001 |
| | N×C | 2.8 | =0.098 | P×C | 6.3 | =0.015 |
| | N×L | 7.0 | <0.001 | P×L | 9.7 | <0.001 |
| | C×L | 11.4 | <0.001 | C×L | 2.2 | =0.078 |
| | N×C×L | 0.6 | =0.639 | P×C×L | 7.0 | <0.001 |
| PIC:POC ratio | N | 934.6 | <0.001 | P | 395.0 | <0.001 |
| | C | 81.8 | <0.001 | C | 9.1 | =0.004 |
| | L | 30.9 | <0.001 | L | 47.6 | <0.001 |
| | N×C | 6.6 | =0.013 | P×C | 13.4 | <0.001 |
| | N×L | 9.8 | <0.001 | P×L | 14.4 | <0.001 |
| | C×L | 6.8 | <0.001 | C×L | 1.5 | =0.202 |
| | N×C×L | 0.7 | =0.567 | P×C×L | 4.7 | =0.002 |
| $F_v/F_m$ | N | 335.8 | <0.001 | P | 171.2 | <0.001 |
| | C | 1.5 | =0.229 | C | 189.6 | <0.001 |
| | L | 246.7 | <0.001 | L | 153.9 | <0.001 |
| | N×C | 16.1 | <0.001 | P×C | 34.8 | <0.001 |
| | N×L | 4.8 | =0.002 | P×L | 13.8 | <0.001 |
| | C×L | 12.6 | <0.001 | C×L | 10.7 | <0.001 |
| | N×C×L | 4.6 | =0.003 | P×C×L | 2.6 | =0.048 |
| $F_v^{'}/F_m^{'}$ | N | 10.1 | =0.002 | P | 675.4 | <0.001 |
| | C | 33.6 | <0.001 | C | 134.0 | <0.001 |



| | | | | | | |
|---|---|---|---|---|---|---|
| | L | 670.5 | <0.001 | L | 1007.7 | <0.001 |
| | N×C | 11.7 | =0.001 | P×C | 195.5 | <0.001 |
| | N×L | 3.4 | =0.014 | P×L | 22.8 | <0.001 |
| | C×L | 14.6 | <0.001 | C×L | 8.2 | <0.001 |
| | N×C×L | 12.6 | <0.001 | P×C×L | 3.5 | =0.012 |
| $ETR_{max}$ | N | 811.2 | <0.001 | P | 335.2 | <0.001 |
| (mol e$^-$ g$^{-1}$ Chl $a$ h$^{-1}$) | C | 67.9 | <0.001 | C | 71.3 | <0.001 |
| | L | 176.6 | <0.001 | L | 625.4 | <0.001 |
| | N×C | 11.2 | =0.001 | P×C | 20.2 | <0.001 |
| | N×L | 15.3 | <0.001 | P×L | 151.0 | <0.001 |
| | C×L | 4.8 | =0.002 | C×L | 35.1 | <0.001 |
| | N×C×L | 12.7 | <0.001 | P×C×L | 9.4 | <0.001 |

N, dissolved inorganic nitrogen (DIN, μmol L$^{-1}$); P, dissolved inorganic phosphate
(DIP, μmol L$^{-1}$); C, $p$CO$_2$ (μatm); L, light intensity (μmol photons m$^{-2}$ s$^{-1}$); POC
quota, particulate organic carbon content; PIC quota, particulate inorganic carbon
content; $F_v/F_m$, maximum photochemical quantum yield;  $F_v'/F_m'$, effective
photochemical quantum yield; $ETR_{max}$, maximum electron transport rate.















**Table 3.** Experimental treatments, growth rate, carbon quotas, photosynthesis
parameter in dilute bath cultures.

| Initial N/P | $pCO_2$ | L | Growth rate | POC quota | PIC quota | PIC: POC | $F_v/F_m$ | $F_v^{'}/F_m^{'}$ | $ETR_{max}$ |
|---|---|---|---|---|---|---|---|---|---|
| 101/ 10.5 | 435 | 80 | 1.11(0.02) | 8.8(0.5) | 1.6(0.4) | 0.19(0.05) | 0.59(0.01) | 0.58(0.03) | 1.25(0.07) |
| | | 120 | 1.21(0.03) | 9.1(0.3) | 2.3(0.7) | 0.25(0.08) | 0.55(0.00) | 0.54(0.01) | 1.52(0.12) |
| | | 200 | 1.37(0.02) | 8.5(0.6) | 2.8(0.7) | 0.33(0.08) | 0.55(0.01) | 0.48(0.01) | 1.65(0.02) |
| | | 320 | 1.29(0.03) | 9.7(1.0) | 5.0(1.3) | 0.52(0.16) | 0.47(0.03) | 0.37(0.03) | 1.58(0.09) |
| | | 480 | 1.17(0.03) | 12.3(0.7) | 3.5(0.4) | 0.28(0.04) | 0.45(0.06) | 0.31(0.02) | 1.63(0.06) |
| | 970 | 80 | 1.06(0.01) | 7.7(0.4) | 0.9(0.1) | 0.12(0.02) | 0.58(0.01) | 0.57(0.02) | 1.16(0.01) |
| | | 120 | 1.19(0.03) | 8.9(0.2) | 2.2(0.4) | 0.25(0.04) | 0.54(0.01) | 0.52(0.01) | 1.69(0.16) |
| | | 200 | 1.32(0.01) | 8.2(0.7) | 2.3(0.4) | 0.28(0.06) | 0.53(0.01) | 0.47(0.01) | 1.61(0.01) |
| | | 320 | 1.21(0.02) | 9.9(0.8) | 2.9(0.7) | 0.30(0.09) | 0.49(0.03) | 0.37(0.02) | 1.60(0.09) |
| | | 480 | 1.16(0.01) | 11.7(1.2) | 1.7(0.4) | 0.14(0.02) | 0.33(0.03) | 0.28(0.02) | 1.24(0.1) |
| 8.8/ 10.5 | 410 | 80 | 1.08(0.01) | 7.3(0.4) | 2.9(0.6) | 0.39(0.09) | 0.59(0.01) | 0.58(0.01) | 1.44(0.04) |
| | | 120 | 1.21(0.01) | 8.4(0.4) | 4.7(0.9) | 0.57(0.12) | 0.57(0.00) | 0.55(0.01) | 2.03(0.11) |
| | | 200 | 1.31(0.01) | 8.1(0.3) | 5.9(0.8) | 0.74(0.08) | 0.59(0.01) | 0.53(0.01) | 2.50(0.15) |
| | | 320 | 1.29(0.01) | 9.9(0.4) | 8.7(0.7) | 0.87(0.07) | 0.45(0.04) | 0.37(0.04) | 2.10(0.07) |
| | | 480 | 1.12(0.02) | 7.9(0.8) | 6.8(0.8) | 0.87(0.17) | 0.41(0.03) | 0.35(0.04) | 1.69(0.14) |
| | 936 | 80 | 1.00(0.01) | 7.8(0.3) | 2.4(0.7) | 0.31(0.11) | 0.59(0.01) | 0.57(0.01) | 1.66(0.04) |
| | | 120 | 1.11(0.01) | 8.9(0.5) | 4.3(0.3) | 0.48(0.04) | 0.55(0.01) | 0.54(0.02) | 1.86(0.06) |
| | | 200 | 1.25(0.01) | 8.3(0.5) | 5.6(0.8) | 0.68(0.09) | 0.54(0.01) | 0.44(0.01) | 2.35(0.16) |
| | | 320 | 1.21(0.01) | 9.7(0.2) | 5.4(0.4) | 0.56(0.05) | 0.50(0.01) | 0.41(0.03) | 2.00(0.08) |
| | | 480 | 1.06(0.06) | 7.2(1.1) | 4.2(0.6) | 0.54(0.06) | 0.37(0.02) | 0.33(0.04) | 1.76(0.15) |
| 101/ 0.4 | 372 | 80 | 1.00(0.02) | 8.7(0.3) | 3.2(0.5) | 0.36(0.06) | 0.59(0.01) | 0.55(0.01) | 1.01(0.05) |
| | | 120 | 1.24(0.01) | 8.3(0.2) | 4.2(0.4) | 0.51(0.05) | 0.59(0.01) | 0.55(0.01) | 1.58(0.04) |
| | | 200 | 1.39(0.01) | 8.1(0.3) | 5.3(0.5) | 0.66(0.09) | 0.56(0.01) | 054(0.02) | 2.10(0.06) |
| | | 320 | 1.31(0.02) | 9.6(0.5) | 4.1(0.6) | 0.43(0.08) | 0.47(0.02) | 0.38(0.01) | 1.85(0.06) |
| | | 480 | 1.18(0.05) | 10.8(0.6) | 2.7(0.5) | 0.25(0.03) | 0.38(0.08) | 0.29(0.04) | 1.61(0.18) |
| | 852 | 80 | 0.97(0.02) | 6.9(0.5) | 2.6(0.4) | 0.38(0.04) | 0.58(0.01) | 0.54(0.02) | 0.91(0.03) |
| | | 120 | 1.08(0.01) | 9.0(0.1) | 3.7(0.7) | 0.41(0.07) | 0.55(0.01) | 0.49(0.01) | 1.29(0.02) |
| | | 200 | 1.27(0.01) | 8.1(0.1) | 4.0(0.3) | 0.49(0.04) | 0.55(0.01) | 0.51(0.02) | 2.16(0.07) |
| | | 320 | 1.22(0.01) | 8.6(0.1) | 3.1(0.4) | 0.36(0.05) | 0.47(0.03) | 0.37(0.03) | 2.18(0.09) |
| | | 480 | 0.90(0.01) | 12.8(0.6) | 3.5(0.6) | 0.28(0.06) | 0.25(0.03) | 0.17(0.01) | 1.21(0.09) |

Initial N/P, the ratio of dissolved inorganic nitrogen to phosphate at the beginning of



experiment; L, light intensity (μmol photons m$^{-2}$ s$^{-1}$). See Table 2 for detailed
information. Data in the brackets are the standard deviations for four replicates.























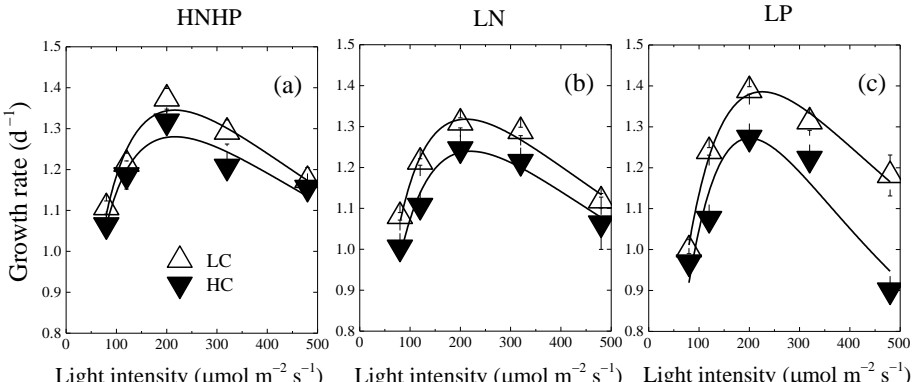





Figure 1












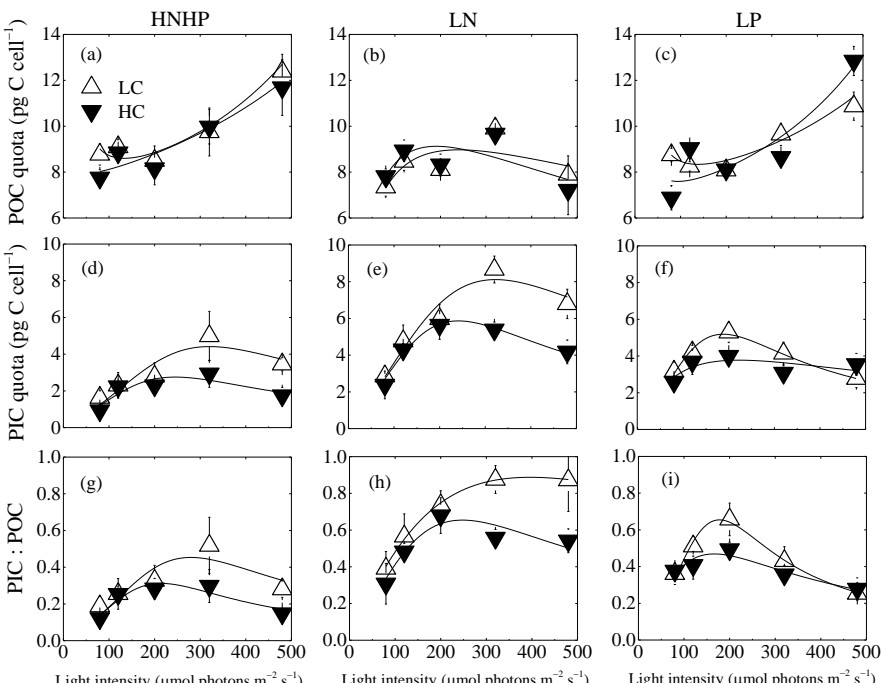





Figure 2









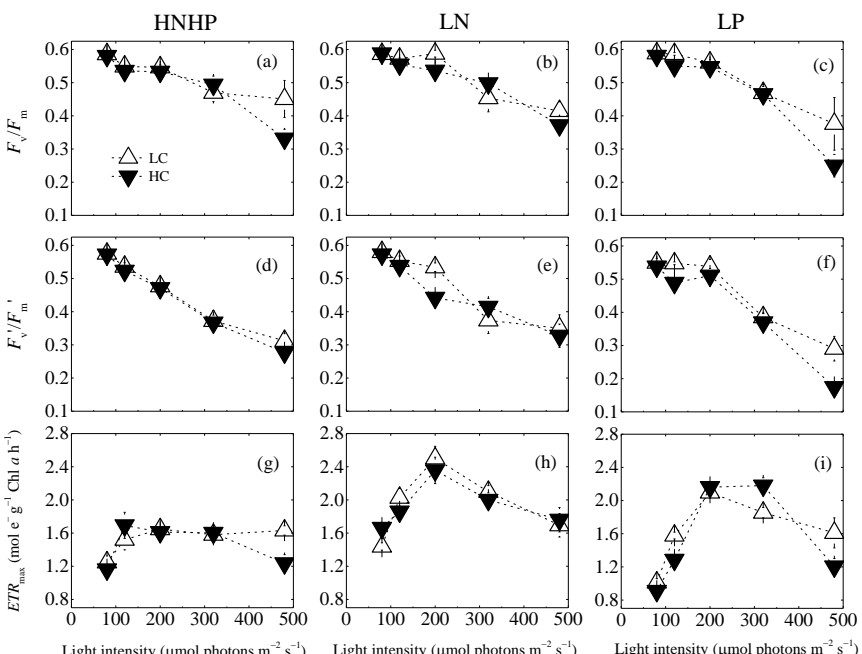





Figure 3











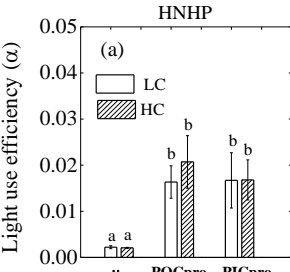 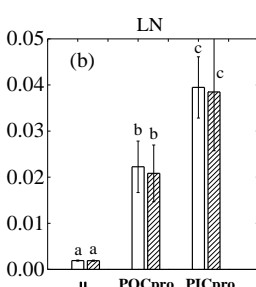 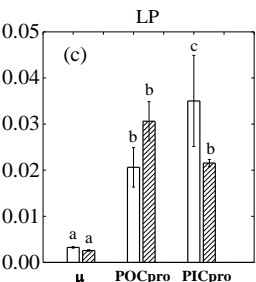




Figure 4











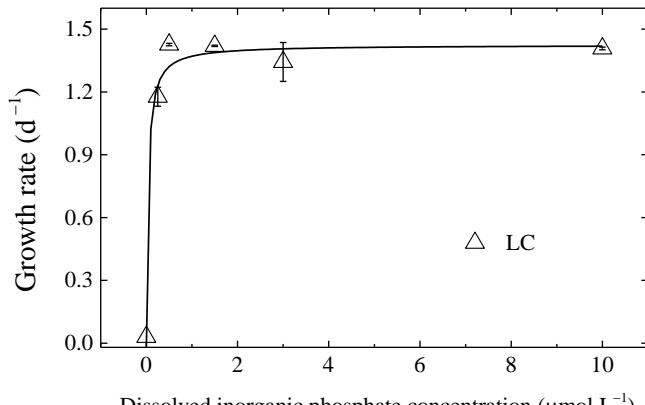





Figure 5