# Peer review of "Interactive effects of seawater carbonate chemistry, light intensity and nutrient availability on physiology and calcification of the coccolithophore *Emiliania huxleyi"

_Biogeosciences, 2018_

## Referee Comment (RC1) · Anonymous Referee #1 · 18 Feb 2018

Review on 'Interactive effects of seawater carbonate chemistry, light intensity and nutrient availability on physiology and calcification of the coccolithophore Emiliania huxleyi' by Zhang et al.

General comments:

The present manuscript presents a comprehensive dataset investigating the interactive effects of carbonate chemistry, light intensities and nutrient availabilities on the coccolithophore E. huxleyi. The dataset consists of an impressively high number of treatments, replicates and measured parameters therein. Given the fact that interaction of multiple drivers are often impossible to predict from simpler experiments, datasets as this one are indispensable to understand the expected natural complexity in climate change effects.

This vast amount of data is, however, somewhat overwhelming and it seems to me that the authors also got lost in it a bit. In its current form, the manuscript is poorly written (also with respect to language) and lacks a story line. While I do acknowledge the difficulty to find overarching patters in such a complicated dataset, the current manuscript does not make it easy for the reader to take home any conclusions. In the discussion, individual paragraphs are often not connected to each other (and sometimes even not between sentences therein). I suggest the authors to focus on the main aspects they want to interpret and discuss these in more than a few sentences, and to omit some of the other (side-)aspects. Likewise, parameters that do not get discussed in detail also do not need to be described in great detail in the (currently quite long) results section. In my opinion, quite some of this information could be sufficiently described in tables and the supplement.

With respect to the general interpretation of the data, I disagree with the way the nutrient treatments are regarded. Despite the fact that cells divided 1-2 times per day ($\mu$ >1 in almost all cases) and were clearly exhibiting non-limited exponential growth, the data is discussed as if the cells were nutrient limited and compared to previous studies that investigated strong nutrient limitation. Regarding nitrogen limitation, for example, residual DIN was 1.0 0.4 $\mu$mol L-1 in LN treatments, which is known to not limit growth, and the molar drawdown in HN and LN treatments is actually similarly high. The same is true for the molar drawdown of DIP. Thus, the discussion needs to be refocussed by considering different but not strongly limiting nutrient concentrations rather that limiting vs. non-limiting conditions. This is particularly the case as growth rates are integrated over the whole duration of the experiment (i.e. mixing phases of non-limited growth with potential limitation towards the end of the experiment), while photophysiological

measurements are only taken at the final (potentially more limited) stage.

Specific comments:

L33-34: The interaction between CO2 and N is actually the least significant term, why do you focus on this interaction and not the others?

L36-37: The authors do not provide any data that would allow to conclude on the competitive abilities of this species. If they want to, they would need to either conduct competition experiments, or compare nutrient uptake kinetics with those of competing species.

L56: Why only from media, not generally from seawater?

L60-65: I do not understand why the authors mention two opposing interpretations of multiple stressor effects (i.e. linearly increasing/decreasing/non-affected vs. optimum curve response) without clarifying why they use the linear trends even though they are aware of the fact the responses follow more complex optimum curves.

L65-67: Really? These is also plenty of evidence for the opposite effect, also published by some of the authors.

L67-70: Intraspecific differences are another well-established reason for differing responses (e.g. Langer et al. 2009).

L75: Photo-acclimation to HL or LL? Both are photo-acclimative processes.

L86-92: The same information is presented in the discussion. Is it really necessary to present it twice with the same level of detail?

L180: How did you measure the pressure inside the syringe filter?

L193: How similar were the PAM light values to those during the incubation? Please provide a quantitative comparison.

L198-203: How was the "cellular absorption value" determined? This parameter most

likely changes strongly with light-acclimation, so I do not think that one constant value can be used to convert relative ETR to absolute ones for all treatments. If the authors did not determine this values for each treatment, they should rather report the ETR in their relative unit.

L214: Is it really true that the authors did not even measure the initial cell count but just assumed inoculation to be perfectly equal among all bottles? I do not trust the growth rate estimates at all if this is the case, especially as small differences in the low abundance range will have huge effects on the final counts.

L234-235: Why were the two nutrient treatments analysed separately?

L266 ff.: It is no clear to me to which of the two tests (i.e. ANOVA vs. post hoc tests) the statements regarding the p values refer to. These are two different things. Please clearly state if you base a statement of "significance" on the ANOVA itself or a post-hoc test in the whole results section. If you describe an optimum-curve behaviour, for example, the ANOVA cannot capture both increasing and decreasing phases of it, but would indicate that one of the two is more dominant.

L412 ff: Quite often, single sentences are not clearly connected. The discussion thus seems like a long list of ideas, but without any structure or line of thought.

L414-416: Why "synergistic negative effects"? This a priory expectation is not stated (nor argued for) in the intro.

L423-430: What does the content of this paragraph mean for the interpretation of the results with respect to nutrient limitation?

L435-439: This could be explained by an excess of PSII reaction centres (Behrenfeld et al. 1998).

L445-447: See my comment regarding competition above.

L466-467: looking at the fit on figure 5, I am not convinced by this, as the fit does not

run close to the data in the relevant part of the curve (i.e. the slope).

L480-482: In the natural environment, 10 $\mu$M NO3 is definitely not "low nutrients".

L483-485: What is "alkaline phosphate activity"? There seems to be a word missing. Also, please explain why this is relevant.

L487-492: I do not understand how this is related to LP conditions. Wouldn't one expect that P limitation would increase energy demand due to upregulated P uptake machinery?

L498-499: This can be solely explained by increasing levels of energy saturation of C acquisition and fixation with increasing light.

L506-509: Seems completely unrelated to the presented and discussed data.

L509-511: Seems completely unrelated to previous discussion.

L515-527: Here, results from really nutrient-limited cultures are compared to the data from this study without discussing the lack of considerable nutrient-limitation of growth. Please rewrite this section by taking this into consideration. Also, take into account that under intermediate light levels, growth rates under P limitation and LC are as high as in the full media.

L535-536: ETRmax were measured at high light, so it cannot be limited by low energy input. Instead, previous acclimation to low light may have hampered usage of the provided energy.

L541: Please clarify that you have no data on CCM down-regulation but that this is speculation based on previous publications.

L547-550: Of course these processes are correlated. Can you provide something new that further elucidates this fact?

L555-558: I do not understand this line of thought. Please explain in more detail.

L563-566: The authors correctly state that highly labour-intensive experiments like the current one are necessary because interactions between multiple stressors cannot be inferred from isolated effects. I therefore do not understand why they speculate on an interaction they did not investigate.

Figure 5 legend: The method description should move into the method section and be more detailed, e.g. were growth rates integrated over 4 days? Were the cultures pre-acclimated to the conditions? If not, which conditions were they acclimate to before?

Technical corrections:

Generally, there are a lot of instances where grammar and wording need to be improved. I strongly suggest the native speakers in the author list to thoroughly correct the final revised version of this manuscript. Below a few examples:

L34-35: Please correct/rephrase this sentence.

L48-49: Please correct/rephrase this sentence.

L55: Replace "in the UML" by "therein"

L57-60: Please correct/rephrase this sentence. Why "counteract"?

L84: Consider replacing "decreased" by "suboptimal"

L86-92: Combine first two sentences into one.

L93-101: Indicate at which pCO2 levels these studies were conducted.

L119: Why "Even"?

L398-403: This is not discussion later on. Is it needed then?

L142-145: Please correct/rephrase this sentence.

L158: For clarity, please add "For each nutrient treatment, [. . .]"

L158-159: Add standard errors for light levels.

L165-167: Please correct/rephrase this sentence.

L178: "CO2 System" should read "CO2SYS"

L182: "Dickson et al. 2003" should read "Dickson et al (2003)".

L185: "equimolal" should read "equimolar".

L186-187: I assume you did not calculate K1 and K2, but used these constants from Roy et al. for your calculations. . . If so, please correct accordingly.

L194-196: Please correct/rephrase this sentence.

L226: Replace "their" by "cellular".

L256-264: Estimates of uncertainty are missing.

L274-279: Units of the treatments are missing.

L346-349: Replace "At each nutrient condition, at both LC and at HC" by "At all nutrient and CO2 levels".

L356: Why "both"?

L439-441: This sentence sounds as if the authors would have observed the first statement, and the reference refers to the latter, while the opposite is true. Please rephrase.

L465: "saturation condition, relationship" should read "saturated conditions, the relationship".

L467-471: Please correct/rephrase this sentence.

L480-482: Please correct/rephrase this sentence.

L496-502: Please correct/rephrase this sentence.

L503: omit first "-"

L516: Insert "could have" between "which" and "led".

L553-555: I do not understand this sentence. Please rephrase.

Figures: Consider using a dashed line for one of the fits to distinguish between the twoCO2 levels.

L869: Based on which test?

L902: Explain letters to abbreviate pCO2 and light intensity.

L920-921: Please rephrase to make it a sentence.

Figure 2: Indicate if the PIC:POC is molar- or weight-based.

---

## Referee Comment (RC2) · Anonymous Referee #2 · 27 Feb 2018

Review of Zhang et al. "Interactive effects of seawater carbonate chemistry, light intensity and nutrient availability on physiology and calcification of the coccolithophore Emiliania huxleyi" submitted to Biogeosciences.

The study by Zhang et al. is an important effort to address multifactorial control over the response to acidification of an important phytoplankton species, using an ambitious matrix of treatments. However, there are some major problems that must be resolved, as currently I am unsure of a major portion of the data or results interpretations as presented.

[Figure]

1. In the Introduction the authors plant the study as if it aims to mimic the natural environment presently or in the future in the laboratory, that is, that the nutrient, light, and CO2 conditions they chose are truly representative. I think this is not unnecessary and risks setting up an incorrect context for interpreting the study results. For example, the authors justify the choice of light regimes in the first paragraph by claiming that phytoplankton in the future ocean will be exposed to higher light levels in the mixed layer, citing two studies. I note also that neither of the studies cited (Gao et al. 2012 and Hutchins and Hu 2017) is relevant to cite, as one is a lab study and the other is a review of lab studies, and neither is a model study predicting average light fields at which phytoplankton will be exposed in the future ocean. In any case, it is difficult to imagine that changes in the stratification of the central ocean basins can only lead to an increase in light exposure. Light exposure is highly dynamic and depends on mixing regime, so yes, the light regime should change, but to model that in the lab with constant light levels is not reasonable. My comment here does not at all negate the study design: Even though we can never mimic the ocean in the lab, it still serves to understand how factors may interact. In the case of trying to predict the response to acidification, it at least serves us to understand how robust the lab results might be for predicting the direction of possible responses, and often as well helps provide insight into mechanisms underlying the responses. I do suggest that they consider revising the Intro.

2. There is at least one major problem with the growth rates reported, possibly many more:

a. It makes no sense to report a single growth rate as the response to nutrient-limitation in batch culture experiments. At inoculation of cultures, cells should be nutrient replete even in the LP and LN conditions. If they have been "acclimated" to growing previously in the same media, the inoculums likely are from cultures that have already exhausted the phosphate (in LP) or nitrate (in LN), so the cells will have to re-configure nutrient uptake and metabolism, begin to grow, then exhaust the nutrients, re-configuring nutrient and connected metabolism again. The growth rate most certainty will NOT be constant. A recent study where these effects can be seen would be that of Rokitta et al. (2014). The authors report only a single growth rate, not changes in cell density over time, no indication of when nutrient limitation may start nor how long cells have been in nutrient limited conditions. In this sense, a good study to look at would be the recent one by Müller et al. (2017) using a continuous culture approach to understand the interaction/independence of nutrient limitation and acidification effects (curiously, the authors cite the study in the Intro but do not discuss at all, despite its central relevance!). The results presented in the current manuscript are therefore completely uninterpretable.

b. The growth rate presented appears to be calculated only from an initial cell concentration and a final one, which is generally not adequate even in batch culture experiments when nutrient limitation is avoided, because it is necessary to understand if growth rate changes or not during the experiment.

c. The initial cell concentration appears not to have been measured, but to have been calculated, which causes many errors.

d. The growth rates provided seem high in comparison to most previous studies of this species. Most authors report that the maximum growth rate of Emiliania huxleyi in batch cultures under "optimum" nutrient and light conditions and a day:night lighting is in the range of 0.7-0.9, a little more than one doubling per day (for just a sampling of studies, see van Bleisjwiik et al. 1994; Zondervan et al. 2002; Rokitta et al. 2014; Müller et al. 2015). Higher growth rates are occasionally reported, but under longer light cycles, e.g. Langer et al. 2009, or a very nice study by the same first author (Zhang et al. 2014). The rates here seem quite high for a 12:12 light:dark cycle, and for that reason it's important to see the data (at least in supplementary), to have full confidence in the methods, and to have at least a brief mention of this.

e. I'm especially concerned in the Methods when they say that 4-6 days corresponds

to 14 generations. That would correspond to growth rates between 1.62 day-1 (at a "low" light level previous studies have found to nearly saturate growth rate) or 2.42 day-1, a level unachievable even for most diatoms (and not readily believable for a coccolithophore, even E. huxleyi). Perhaps this is a typographical error?

f. For cell counts they use a particle counter (presumably based on the Coulter principle, although the information provided is inadequate to identify the type of instrument). This is potentially very problematic particularly in the case of E. huxleyi. How can living cells be distinguished from detached coccoliths, agglomerations of detached coccoliths, and/or empty coccospheres, all of which are very abundant in E. huxleyi cultures? In limited conditions these other particles can actually dominate the suspended particles found in cultures and it can be difficult to distinguish cells. With all these issues, I really am not sure from the information provided that they are actually measuring cells. Cells should be counted under a microscope or with a flow cytometer (a Coulter-type particle counter can be used, if it is being checked, compared, calibrated with microscope or cytometer counts throughout the experiment). Details are needed.

Because of these unresolved methodological issues in measuring cell abundance, at the present time I cannot trust growth rate data or cell elemental quotas reported.

3. There is no way to know when nutrients became depleted. In the case of nitrate, it is not clear if that nutrient became limiting or sampling occurred when cells were just about to use up the last uM. In this sense, it is essentially impossible to interpret differences in any of the measured parameters between HNHP, LN, and LP conditions. The Fv/Fm data in Fig. 3 heightens my suspicion that cells never truly reached P starvation under LP conditions, as Fv/Fm doesn't show any clear drop in LP compared to HNHP condition at any light or CO2 treatment (compare to Rokitta et al. 2016, for example). In the case of phosphate, perhaps they became limiting at the end, but when? The fact that the increase in PIC/cell reported in many previous studies wasn't observed, but occurred under LN instead, is consistent with the suspicions that the presumed nutrient status was not limiting (and that cell abundance was not being

measured properly).

4. I think the approach for analyzing and interpreting the data could be more powerful:

a. The 3-way ANOVA ignores differences between LP and LN conditions

b. The 3-way ANOVA approach followed by a posthoc test to identify pairwise differences can be valid, but it doesn't help for identifying patterns. In this case, the Eilers and Peeters model they fit would help, but they only look at the fit of the alpha parameter, when the curves shown in their figures clearly indicate that the other fitted parameters (a, b, c) may be interesting as well.

5. What about cell volume effects? As reported recently by Müller et al. (2017), these could be crucial. If I understood that previous study correctly, nutrient limitation seemed to act independently rather than synergistically with ocean acidification when cell volume was accounted for. Of course, that study used continuous culture rather than batch culture/starvation conditions, but still it seems relevant at least to consider. Currently the Discussion seems to ignore some relevant studies such as Müller et al. 2017 that I previously cited, as well as Olson et al. 2016. Further, it needs to be much clearer. Finally, some revision of the English is suggested.

6. The Discussion focuses a lot on ETR and photophysiology (Fv/Fm, Fv'/Fm'), which doesn't make a lot of sense. Effects both of high CO2 and of supposed nutrient limitation on photophysiological parameters seem to be subtle in comparison to what they report on growth rates and cell quotas. The light dependence of photosynthesis in E. huxleyi has actually been comparatively well studied, and much of the discussion seems overly speculative and not to focus on some of the curious differences with what has been reported previously (e.g., Houdan et al. 2005 reporting that calcified cells are especially resistant to high PAR).

The doubts I have about the study are quite serious, and hopefully my major comments (above) and minor comments (below) help the authors determine where to clarify. Nevertheless, I think the study design may not be appropriate for investigating an interaction between nutrient limitation and acidification. The only way such an experimental design could potentially work for the question planted is with daily and trusted cell counts and nutrient measurements showing when nutrient depletion occurred.

There are many minor points through the manuscript to address as well. I mention a few:

Line 27 "and exposing phytoplankton to increased light intensities" and lines 52-53 later. I think this is too much over-simplification. At the base of the mixed layer and within the pycnocline, nutrients can be obtained by diffusion across the pycnocline, so phytoplankton will grow and increase in biomass until they compete for light. I do not see how the average light exposure of phytoplankton will necessarily increase. The references cited (Gao et al. 2012 and Hutchinson and Fu 2017) do not explain this (as I mentioned earlier).

Lines 101-103 "Interaction of rising CO2 with light appears to affect differentially coccolithophores when grown under different experimental setups." The sentence is not clear. What does "differentially affect coccolithophores" mean? Do those factors affect coccolithophores differently than other phytoplankton or do these factors have contrasting effects or ??

Lines 142-144: "added by 2200 $\mu$mol L−1 bicarbonate (as opposed to 2380 $\mu$mol L−1 in the original recipe), in order to reflect the alkalinity in the South and East China Seas of about 2200 $\mu$mol L−1" First, I don't understand why it's important to match the South and East China Seas if they are not specifically using strains isolated from those seas and trying to predict how organisms there will respond. Second, I don't see how bicarbonate concentration is equated with alkalinity, as CO32- also contributes to alkalinity, and for alkalinity every unit of CO32- counts twice. I think carbonate usually can contribute about a fifth or a fourth or so of total alkalinity (see Zeebe and Wolfgladrow 2001 or other references).

Lines 161-165: I mentioned above the problems with these lines.

Line 172: How often were nutrients measured?

Lines 182-184: Was pH measured immediately or after storage? pH should be measured immediately, as I understand (within a couple hours is best).

Lines 210-215: I already mentioned the major problems I have with the methodology as described here. Perhaps they can fix that.

Line 225: Do they mean "difference" instead of "variance"

Lines 251-255: It would be invaluable to know when nutrients were depleted. Do they have data on this?

Lines 256-264: This whole paragraph is basically redundant with Table 1. Also, it seems "(mean values for the beginning and end of incubations)" means that the beginning and ending values have been averaged together, while in Table S2 the beginning and ending values are given separately. Table S2 is far more useful, especially for assessing the changes in carbonate parameters during the experiment. I would place that table (S2) in the main text, using it to replace the current Table 1. Then in this text the focus should be more on how consistent were carbonate parameters over time and across treatments within the LC and within the HC treatments. Furthermore, when I calculate the averages using the values given in the text immediately before, I get different values (405 for LC and 918 for HC). What is happening? Were some replicates not used?

Lines 368-373: This is not a very good description. It seems to exaggerate small differences between HC and LC.

Line 467: I can't find any reference to Fig. 5 in the Results. Why does it appear suddenly in the Discussion? Further, I have problems with this Michaelis-Menten fit: It does not make any sense to fit growth rate in a batch culture (measured from initial and final concentrations) to the initial phosphate concentration. This seems to ignore

understanding of phytoplankton macronutrient physiology since Droop. But, as there isn't a clear description of this experiment, I am not sure. Finally, the ability to calculate a half-saturation from the data in the graph would be very limited because there is no value in an intermediate range of growth (growth rate is either 0 or saturated or nearly saturated). For this entire paragraph, the study was not designed to address the details of phosphate metabolism, which has already been fairly extensively studied in this species, and their discussion of the previous work is unclear.

Line 529: Do the authors consider the ballast effect to be completely irrelevant? Also, I have an issue with considering E. huxleyi as representative of the biogeochemically most important coccolithophores. It is the most numerically abundant, most widespread, and most easy to culture in the laboratory. However, E. huxleyi is definitely not the coccolithophore responsible for most sinking inorganic carbon and it may not be an appropriate model for the responses of other principal groups of coccolithophores, as it (and it's close relatives in the Gephyrocapsa genus) is different from most coccolithophores. For example, E. huxleyi does not require Si for calcification while most do.

In general, I have a hard time following the Discussion. It lacks clarity and focus, and seems to stray into inadequate review of important but peripheral themes. It's difficult for me to provide more detailed comments as I am not convinced that they know what state of nutrient limitation (or not) the cells were in when harvested.

―――――――――――――――――――――――――

---

## Short Comment (SC1) · 28 Feb 2018

Dear Referee,

Thank you for spending time to review our manuscript. We will make some changes according to your comments one by one. Best Regards

---

## Short Comment (SC2) · 28 Feb 2018

Dear Referee,

Thank you for spending time to review our manuscript. We will make some changes according to your comments one by one. Best Regards

---

## Author Comment (AC1) · 14 Apr 2018

**Dear Referees,**

We thank you for your supportive comments and the constructive reviews on our manuscript. Our detailed responses in blue text to your comments are attached. The changed contents in the revised manuscript are underlined.

**Responses to comments of referee 1 are shown as following:**

**General comments:**

The present manuscript presents a comprehensive dataset investigating the interactive effects of carbonate chemistry, light intensities and nutrient availabilities on the coccolithophore E. huxleyi. The dataset consists of an impressively high number of treatments, replicates and measured parameters therein. Given the fact that interaction of multiple drivers are often impossible to predict from simpler experiments, datasets as this one are indispensable to understand the expected natural complexity in climate change effects.

Response: We thank this referee for his (her) kind words.

This vast amount of data is, however, somewhat overwhelming and it seems to me that the authors also got lost in it a bit. In its current form, the manuscript is poorly written (also with respect to language) and lacks a story line. While I do acknowledge the difficulty to find overarching patters in such a complicated dataset, the current manuscript does not make it easy for the reader to take home any conclusions. In the discussion, individual paragraphs are often not connected to each other (and sometimes even not between sentences therein). I suggest the authors to focus on the main aspects they want to interpret and discuss these in more than a few sentences, and to omit some of the other (side-)aspects. Likewise, parameters that do not get discussed in detail also do not need to be described in great detail in the (currently quite long) results section. In my opinion, quite some of this information could be sufficiently described in tables and the supplement.

Response: We agree with the suggestions of this referee. The manuscript has been refocused on growth rate, POC and PIC production rates, and fitted alpha (*a*) and maximum values for growth, POC and PIC production rates. We omitted a description of the rETR. The take home conclusions are that: (1) low dissolved inorganic nitrogen (LN) concentration and high  $CO_2$  level synergistically reduced growth and POC production rate; (2) At high light intensity, low dissolved inorganic phosphate (LP) concentration did not limit growth rate at LC but led to increased high-light inhibition of growth rate at HC; (3) low nutrient concentrations (DIN or DIP) increased the maximum value and the light-use efficiencies of calcification rate. These changes are in Lines 35–39 and Lines 43–44 on page 2.

With respect to the general interpretation of the data, I disagree with the way the nutrient treatments are regarded. Despite the fact that cells divided 1-2 times per day ( $\mu$ >1 in almost all cases) and were clearly exhibiting non-limited exponential growth, the data is discussed as if the cells were nutrient limited and compared to previous studies that investigated strong

nutrient limitation. Regarding nitrogen limitation, for example, residual DIN was 1.0 0.4  $\mu$ mol L-1 in LN treatments, which is known to not limit growth, and the molar drawdown in HN and LN treatments is actually similarly high. The same is true for the molar drawdown of DIP. Thus, the discussion needs to be refocussed by considering different but not strongly limiting nutrient concentrations rather that limiting vs. non-limiting conditions. This is particularly the case as growth rates are integrated over the whole duration of the experiment (i.e. mixing phases of non-limited growth with potential limitation towards the end of the experiment), while photophysiological measurements are only taken at the final (potentially more limited) stage.

Response: Thanks for the important and supportive comments of this referee. We agree with this referee that growth, POC and PIC production rates of cells were not limited by low DIN or DIP concentration. We refocused on differences in growth, POC and PIC production rates caused by high and low nutrient concentrations rather than on liming and non-limiting nutrient conditions.

For different growth rates between HNHP and LN conditions, we described that:

LN concentration was shown to down-regulate transcripts of genes related to nitrate reductase (NRase) activity, synthesis of amino acids, RNA polymerases and nitrogen metabolism in *E. huxleyi* (Bruhn et al., 2010; Rouco et al., 2013; Rokitta et al., 2014), which led to lower overall biosynthetic activity and decreased the growth rates (Fig. 1). These changes are in Lines 620–625 on page 29.

For subsection in the discussion section: 'Effect of low dissolved inorganic phosphate concentration on growth rate was modulated by light intensity and  $CO_2$  level', we described that: 1. In this study, low light intensity not only limited cell growth but also was suggested to limit phosphate uptake rates (Nalewajko and Lee, 1983). In this case, compared to the HNHP condition, growth rates of *E. huxleyi* at LP condition were more likely to be limited by low-light intensity (Fig. 1a,c). High light intensity provided energy for cells to take up P, and cells at LP condition need to consume more energy to up-regulate P uptake (Nalewajko and Lee, 1983) which may lead to decreased high-light inhibition of growth rate at LP than at HNHP condition under LC. Furthermore, growth rate of *E. huxleyi* was nearly saturated at 0.25 µmol L-1 DIP and was saturated at 0.5 µmol L-1 DIP and above. This demonstrated that *E. huxleyi* possesses a high affinity for DIP (Fig. 5) which allowed *E. huxleyi* to take up  $PO_4^3$  efficiently. Rokitta et al. (2016)

showed that even though  $PO_4^{3-}$  concentration in the culture media declined to zero (undetectable),

cell number sustained to increase for 4 days, which indicates that *E. huxleyi* cells could store phosphorus for later use. Consequently, high energy consumption mechanisms, efficient uptake and storage capacity for phosphorus in *E. huxleyi* could account for there being no significant differences in growth rates between LP and HNHPatLC and high light intensities. These changes are in Lines 639–662 on pages 29 and 30.

2. Rising  $CO_2$  was found to lead to higher phosphorous requirements for growth, carbon fixation and nitrogen uptake in *E. huxleyi* (Matthiessen et al., 2012; Rouce et al., 2013). At HC, higher

phosphorous requirements may lead to lower growth rates at LP in comparison to HNHP (Fig. 1a,c). In addition, at LP, cell volume was 17% larger at HC than at LC under the highest light intensity (Table S1). Large cell volume can directly lead to lower growth rates and reduce nutrient uptake by cells, thereby limiting growth Another possible reason for low tolerance to high-light intensity in growth rate at LP and HC might be a combined effect of LP and HC on the carbon concentrating mechanism (CCM) of *E. huxleyi*. LP or HC is hypothesized to down-regulate the activity of CCM in the green algae *Chlorella emersonii* and in *E. huxleyi*, respectively (Rost and Riebesell, 2004; Beardall et al. 2005). When grown at HC, LP may minimize the activity of CCM of *E. hulxeyi*, which could lead to less energy cost for maintaining high efficient CCM. The saved energy in the HC- and LP-grown cells might have exacerbated photo-inhibition. In summary, high phosphorous requirement, large cell volume and less energy consumption at LP and HC conditions may lead to increased high-light inhibition of growth rates of *E. huxleyi* (Fig. 1). These changes are in Lines 670–692 on pages 31 and 32.

Nalewajko, C., Lee, K. : Light stimulation of phosphate uptake in marine phytoplankton, Mar. Bio., 74, 9–15, https://doi.org./10.1007/BF00394269, 1983.

Beardall, J., Roberts, S., Raven, J. A. : Regulation of inorganic carbon acquisition by phosphorus limitation in the green alga *Chlorella emersonii*, Can. J. Bot., 83, 859–864, https://doi.org/10.1139/b05-070, 2005.

**Specific comments:**

L33-34: The interaction between CO2 and N is actually the least significant term, why do you focus on this interaction and not the others?

Response: Synergistic effects of low dissolved inorganic nitrogen (DIN) concentration and high  $CO_2$  level on growth and POC production rates are one of main results in this study. We also refocused on interactive effects of DIP concentration,  $CO_2$  level and light intensity on growth and POC production rates, and effect of low nutrient concentrations on PIC production rate.

This sentence '*HC and LN synergistically decreased growth rates of E. huxleyi at all light intensities.*' were replaced by 'LN and HC synergistically reduced growth and POC production rates.' These changes are in Line 34–36 on page 2.

L36-37: The authors do not provide any data that would allow to conclude on the competitive abilities of this species. If they want to, they would need to either conduct competition experiments, or compare nutrient uptake kinetics with those of competing species.

Response: We thank to this referee for their suggestions.

This sentence 'These results indicate that the ability of E. huxleyi to compete for nitrate and phosphate may be reduced in future oceans with high  $CO_2$  and high light intensities.' was replaced by 'These results showed that effects of nutrient concentrations on physiological rates of E. huxleyi were modulated by  $CO_2$  level and light intensity.' These changes are in Lines 39–42 on page 2.

**L56: Why only from media, not generally from seawater?**

Response: 'Coccolithophores take up  $CO_2$  and/or  $HCO_3^-$  from seawater for carboxylation'. These

changes are in Line 66 on page 3.

L60-65: I do not understand why the authors mention two opposing interpretations of multiple stressor effects (i.e. linearly increasing/decreasing/non-affected vs. optimum curve response) without clarifying why they use the linear trends even though they are aware of the fact the responses follow more complex optimum curves.

Response: The text 'Growth rate, particulate organic (POC) and inorganic carbon (PIC) production rates of Emiliania huxleyi, the most abundant calcifying coccolithophore species, usually display optimum responses to a broad range of  $CO_2$  concentration, with growth, POC and PIC production rates increased, decreased or unaffected by rising  $CO_2$  treatments (Langer et al., 2009; Richier et al., 2011; Bach et al., 2015; Jin et al., 2017).' was replaced by 'Growth rate, particulate organic (POC) and inorganic carbon (PIC) production rates of *Emiliania huxleyi*, the most abundant calcifying coccolithophore species, usually display optimum responses to a broad range of  $CO_2$  concentration (Bach et al., 2011). Growth, POC and PIC production rates could increase, decrease and be unaffected by rising  $CO_2$  treatments across a narrow  $CO_2$  range, which is dependent on the optimal  $CO_2$  levels of these physiological rates and the selected  $CO_2$  range (Langer et al., 2009; Richier et al., 2011; Bach et al., 2011; Bach et al., 2015; Jin et al., 2017).' These changes are in Lines 70–78 on page 4.

**L65-67: Really? These is also plenty of evidence for the opposite effect, also published by some of the authors.**

Response: We deleted this sentence 'Increased light levels could counteract the negative effects of rising  $CO_2$  on calcification in E. huxleyi when grown under natural fluctuating sunlight (Jin et al., 2017).' These changes are lin Lines 78–80 on page 4.

**L67-70: Intraspecific differences are another well-established reason for differing responses (e.g. Langer et al. 2009).**

Response: Differences in sampling locations, experimental setups, deviations in the measuring methods and intraspecific differences can generally be responsible for the differential responses of growth, POC and PIC productions to rising  $CO_2$  in *E. huxleyi* (Langer et al., 2009; Meyer and Riebesell, 2015). These changes are in Lines 80–84 on page 4.

Langer, G., Nehrke, G., Probert, I., Ly, J., and Ziveri, P.: Strain-specific responses of *Emiliania huxleyi* to changing seawater carbonate chemistry, Biogeosciences, 6, 2637–2646, https://doi.org/10.5194/bg-6-2637-2009, 2009.

**L75: Photo-acclimation to HL or LL? Both are photo-acclimative processes**

Response: Reduction in pigment content and effective photochemical quantum yield  $(F'_v/F'_m)$  are

characteristics of photo-acclimation to high light intensity (Geider et al., 1997; Gao et al., 2012). These changes are in Lines 89 on page 4.

L86-92: The same information is presented in the discussion. Is it really necessary to present it twice with the same level of detail?

Response: This text 'Nevertheless, low nutrient concentrations often enhance the PIC quotas of E. huxleyi. This is due to the fact that low nutrient concentrations hold the cells in the G1 cell cycle phase where calcification occurs (Müller et al., 2008). A recent proteome study on E. huxleyi also shows that nutrient limitation arrests cell cycling (McKew et al., 2015). At molecular levels, nitrate or phosphate limitations down-regulate expression of genes involved in cell cycling, RNA and protein synthesis in E. huxleyi (Rokitta et al., 2014, 2016).' were replaced by 'Nevertheless, low nutrient concentrations often enhance the PIC quotas of E. huxleyi because low nutrient concentrations arrest cell cycling and lengthen the G1 phase where calcification occurs (Müller et al., 2008; McKew et al., 2015).' These changes are in Lines 100–108 on page 5.

**L180: How did you measure the pressure inside the syringe filter?**

Response: We cannot measure the pressure inside the syringe filter. But we used an instrument to pump seawater, which was filtered by the syringe filter. The pressure of the pump was 200 mbar.

In the final days of incubation, 25 mL samples for TA measurements were filtered (0.22  $\mu$ m pore size, Syringe Filter) by gentle pressure with 200 mbar in the pump (GM-0.5A, JINTENG) and stored at 4 °C for a maximum of 7 days. These changes are in **Lines 214** on page 10.

**L193: How similar were the PAM light values to those during the incubation? Please provide a quantitative comparison.**

Response: PAM light values are shown in the table R1. But we deleted the description of ETR in lines 232 –243 on page 11.

| Table K1. Comparison between LAW right values and incubation right intensity |    |    |     |     |     |     |     |      |      |
|------------------------------------------------------------------------------|----|----|-----|-----|-----|-----|-----|------|------|
| Light values                                                                 | 1  | 2  | 3   | 4   | 5   | 6   | 7   | 8    | 9    |
| $(\mu mol photons m^{-2} s^{-1})$                                            |    |    |     |     |     |     |     |      |      |
| PAM light                                                                    | 42 | 92 | 133 | 210 | 300 | 450 | 850 | 1126 | 1600 |
| Incubation light                                                             |    | 80 | 120 | 200 | 320 | 480 |     |      |      |

Table R1. Comparison between PAM light values and incubation light intensity

L198-203: How was the "cellular absorption value" determined? This parameter most likely changes strongly with light-acclimation, so I do not think that one constant value can be used to convert relative ETR to absolute ones for all treatments. If the authors did not determine this values for each treatment, they should rather report the ETR in their relative unit.

Response: We agree with this referee that cellular absorption value changes strongly with light-acclimation. But we deleted the description of ETR in lines 232 - 243 on page 11.

L214: Is it really true that the authors did not even measure the initial cell count but just assumed inoculation to be perfectly equal among all bottles? I do not trust the growth rate estimates at all if this is the case, especially as small differences in the low abundance range will have huge effects on the final counts.

Response: The bottles were filled with Aquil with no headspace to minimize gas exchange. The volume of the inoculum was calculated (see below) and the same volume of Aquil was taken out from 500 mL bottles before inoculation. These changes are in **Lines 179–182** on page 9.

There was 625 ml seawater in the 500 ml polycarbonate (PC) bottles. Before cells were inoculated to new seawater, finial cell concentrations (C0) were measured. Then we calculated the inoculated volumes (V) according to  $V = (200 \text{ cell/ml x } 625 \text{ ml})/C_0$ . And we don't think this method cause errors.

**L234-235: Why were the two nutrient treatments analysed separately?**

Response: We re-analyzed the data with a 3-way ANOVA, which shows individual and interactive effects of nutrient concentration,  $CO_2$  level and light intensity, and compares differences among HNHP, LN and LP conditions.

A three-way ANOVA was used to determine the main effect of dissolved inorganicnutrient concentration,  $pCO_2$ , light intensity and their interactions for these variables. A two-way ANOVA was performed to test the main effect of dissolved inorganic nutrient concentration,  $pCO_2$  and their interactions on fitted *a* and  $V_{max}$  of growth, POC and PIC production rates. When necessary, a Tukey Post hoc (Tukey HSD) test was used to identify the differences between two CO2 levels, nutrient concentrations or light intensities. These changes are in **Lines 290–298** on page 14.

**Table 2.** Results of three-way ANOVAs of the impacts of dissolved inorganic nutrient concentration,  $pCO_2$ , light intensity and their interaction on growth rate,  $\frac{F_v/F_{m_2}}{F_v}F'_{m_2}$  POC and

|                                 | Factor  | F value | p value |
|---------------------------------|---------|---------|----------------|
| Growth rate $(d^{-1})$          | Nut     | 264.7   | < 0.01         |
|                                 | С       | 875.6   | < 0.01         |
|                                 | L       | 2035.8  | < 0.01         |
|                                 | Nut×C   | 53.6    | < 0.01         |
|                                 | Nut×L   | 84.2    | < 0.01         |
|                                 | C×L     | 9.3     | < 0.01         |
|                                 | Nut×C×L | 26.8    | < 0.01         |
| $F_{\rm v}/F_{\rm m}$           | Nut     | 68.6    | < 0.01         |
|                                 | С       | 184.7   | < 0.01         |
|                                 | L       | 225.8   | < 0.01         |
|                                 | Nut×C   | 10.3    | < 0.01         |
|                                 | Nut×L   | 8.1     | < 0.01         |
|                                 | C×L     | 15      | < 0.01         |
|                                 | Nut×C×L | 5.2     | < 0.01         |
| F' / F'                         | Nut     | 63.9    | < 0.01         |
| $\Gamma_{\rm v}/\Gamma_{\rm m}$ | С       | 181.8   | < 0.01         |
|                                 | L       | 1161.8  | < 0.01         |
|                                 | Nut×C   | 51.9    | < 0.01         |
|                                 | Nut×L   | 15.3    | < 0.01         |
|                                 | C×L     | 9.9     | < 0.01         |
|                                 | Nut×C×L | 8.1     | < 0.01         |
| POC production rate             | Nut     | 11.8    | < 0.01         |

PIC production rates, and PIC:POC ratio.

| $(pg C cell^{-1} d^{-1})$ | С       | 128.9 | < 0.01 |
|---------------------------|---------|-------|--------|
|                           | L       | 293.7 | < 0.01 |
|                           | Nut×C   | 4.9   | =0.01  |
|                           | Nut×L   | 19.0  | < 0.01 |
|                           | C×L     | 8.47  | < 0.01 |
|                           | Nut×C×L | 1.94  | =0.06  |
| PIC production rate       | Nut     | 624.4 | < 0.01 |
| $(pg C cell^{-1} d^{-1})$ | С       | 142.0 | < 0.01 |
|                           | L       | 147.2 | < 0.01 |
|                           | Nut×C   | 1.9   | =0.16  |
|                           | Nut×L   | 17.3  | < 0.01 |
|                           | C×L     | 8.1   | < 0.01 |
|                           | Nut×C×L | 4.6   | < 0.01 |
| PIC:POC ratio             | Nut     | 326.7 | < 0.01 |
|                           | С       | 57.7  | < 0.01 |
|                           | L       | 41.8  | < 0.01 |
|                           | Nut×C   | 8.3   | < 0.01 |
|                           | Nut×L   | 12.5  | < 0.01 |
|                           | C×L     | 4.0   | < 0.01 |
|                           | Nut×C×L | 3.3   | < 0.01 |

Nut, dissolved inorganic nutrient concentrations ( $\mu$ mol L-1); C, *p*CO2 ( $\mu$ atm); L, light intensity ( $\mu$ mol photons m-2 s-1); POC and POC production rates, particulate organic and inorganic carbon production rates;  $F_v/F_m$ , maximum photochemical quantum yield;  $F'_v/F'_m$ , effective photochemical quantum yield. These changes are in Lines 1198–1209 on pages 56–58.

**Table 4.** Results of two-way ANOVAs of the effects of dissolved inorganic nutrient concentration and  $pCO_2$  on fitted *a* and maximum value ( $V_{max}$ ) of growth, POC and PIC production rates. More detailed information is given as in Table 2. These changes are in Lines 1246–1249 on page 62.

|               |                     | Factor              | F value | p value |
|---------------|---------------------|---------------------|---------|----------------|
| a             | Growth rate         | Nut                 | 18.08   | < 0.001        |
|               |                     | CO 2     | 0.186   | 0.6711         |
|               |                     | Nut×CO 2 | 0.398   | 0.6776         |
|               | POC production rate | Nut                 | 7.21    | 0.005          |
|               |                     | $CO_2$              | 7.78    | 0.0121         |
|               |                     | Nut×CO 2 | 2.50    | 0.11           |
|               | PIC production rate | Nut                 | 21.73   | < 0.001        |
|               |                     | $CO_2$              | 2.32    | 0.145          |
|               |                     | Nut×CO 2 | 2.56    | 0.105          |
| $V_{\rm max}$ | Growth rate         | Nut                 | 24.9    | < 0.001        |
|               |                     | $CO_2$              | 572.7   | < 0.001        |
|               |                     | Nut×CO 2 | 14.8    | < 0.001        |
|               | POC production rate | Nut                 | 7.301   | 0.0048         |

|                     | $CO_2$              | 15.95 | 0.0009  |
|---------------------|---------------------|-------|---------|
|                     | Nut×CO 2 | 1.91  | 0.177   |
| PIC production rate | Nut                 | 56.06 | < 0.001 |
|                     | $CO_2$              | 86.84 | < 0.001 |
|                     | $Nut \times CO_2$   | 0.168 | 0.85    |

L266 ff.: It is no clear to me to which of the two tests (i.e. ANOVA vs. post hoc tests) the statements regarding the p values refer to. These are two different things. Please clearly state if you base a statement of "significance" on the ANOVA itself or a posthoc test in the whole results section. If you describe an optimum-curve behaviour, for example, the ANOVA cannot capture both increasing and decreasing phases of it, but would indicate that one of the two is more dominant.

Response: p value in Table 2 (*see* above) in the manuscript refers to ANOVA, and p value in the results section refers to Tukey post hoc test. All Tukey Post hoc test in the results section were stated by 'Tukey HSD'. Alpha (*a*) and maximum value ( $V_{max}$ ) of growth, POC and PIC production rates (optimum-curve behaviour) were calculated from fitted parameters *a*, *b* and *c* based on model of Eilers and Peeters (1988). And a two-way ANOVA was used to test effects of nutrient concentration and CO2 level on *a* and  $V_{max}$  (Table 4).

The apparent light use efficiency, the slope ( $\alpha$ ), for each light response curve was estimated as  $\alpha = 1/c$ . The maximum values ( $V_{\text{max}}$ ) of growth, POC and PIC production rates were calculated

according to  $V_{\text{max}} = \frac{1}{b + 2\sqrt{ac}}$ . These changes are in Lines 286–289 on pages 13 and 14.

Eilers, P., and Peeters, J.: A model for the relationship between light intensity and the rate of photosynthesis in phytoplankton, Ecol. Model., 42, 199–215, https://doi.org/10.1016/0304-3800(88)90057-9, 1988.

L412 ff: Quite often, single sentences are not clearly connected. The discussion thus seems like a long list of ideas, but without any structure or line of thought.

Response: In the discussion section, we refocused on: (1) low dissolved inorganic nitrogen concentration and high  $CO_2$  level synergistically reduced growth rate; (2) Effect of low dissolved inorganic phosphate concentration on growth rate was modulated by light intensity and  $CO_2$  level; (3) low dissolved inorganic nutrient concentration (DIN or DIP) and high  $CO_2$  level synergistically reduced POC production rate; (4) low nutrient concentrations (DIN or DIP) facilitated PIC production rate. These topics have been shown as subsections of the discussion section in the revised BG manuscript.

**L414-416: Why "synergistic negative effects"? This a priory expectation is not stated (nor argued for) in the intro.**

Response: As shown in Fig. 4 in the revised manuscript, maximum growth rates were significantly

lower at LN than at HNHP under both LC and HC; and they were lower at HC than at LC. So LN and HC synergistically reduced growth rates. Previous studies generally reported effects of low nutrient concentration and rising  $CO_2$  on POC quota, so this expectation is not stated in the introduction (Sciandra et al., 2003; Rouco et al., 2013). We deleted these contents in Lines **581–583** on page 27.

Sciandra, A., Harlay, J., Lefévre, D., Lemée, R., Rimmelin, P., Denis, M., and Gattuso, J. P.: Response of coccolithophorid *Emiliania huxleyi* to elevated partial pressure of CO2 under nitrogen limitation, Mar. Ecol. Prog. Ser., 261, 111–122, https://doi.org/10.3354/meps261111, 2003. Rouco, M., Branson, O., Lebrato, M., and Iglesias-Rodríguez, M. D.: The effect of nitrate and phosphate availability on *Emiliania huxleyi* (NZEH) physiology under different CO2 scenarios, Front. Microbiol., 4, 155, https://doi.org/10.3389/fmicb.2013.00155, 2013.

**Line 1322:**

Figure 4. At both LC and HC, fitted a (a) and maximum (b) of growth rate at HNHP, LN and LP conditions. At both LC and HC, fitted a (c) and maximum (d) of POC prodution rate at HNHP, LN and LP conditions. At both LC and HC, fitted a (e) and maximum (f) of PIC production rate at HNHP, LN and LP conditions.  $\alpha$  was the slope of fitted lines for growth, POC and PIC production rates. Different letters showed statistical differences based on the Tukey post hoc test. The values represent the mean  $\pm$  standard deviation for four replicates. These changes are in Lines 1135–1141 on page 52.

**L423-430: What does the content of this paragraph mean for the interpretation of the results with respect to nutrient limitation?**

Response: In order to show that growth of E. huxleyi is in the exponential phase at the fourth to

sixth days during culturing, we cited Langer et al. (2013).

These contents in Lines 590–597 in page 27: 'Langer et al. (2013) detected that growth of cell on the fourth to sixth days during cultures was in the exponential phase even at 3  $\mu$ mol L-1 NO3 or

at 0.29  $\mu$ mol L-1 PO43- with the same *E. huxleyi* strain. In this study, all parameters were measured

on the fourth to the sixth days, and it is most likely that cells at all treatments were sampled in the exponential growth phase' were transferred to the materials and methods section in Lines 197–202 on pages 9 and 10.

**L435-439: This could be explained by an excess of PSII reaction centres (Behrenfeld et al. 1998).**

Response: We thank to this referee for his (her) nice suggestion. At high light intensity, increases in electron turnover rate through PSII can protect photosynthesis from photoinhibition. Once electron turnover rate started to decrease after it maximized, light-saturated photosynthetic rates decreased.

'because high light intensity can constantly damage the reaction centers of photosystem II (PSII) of *E. huxleyi* (Fig. 2) and maximize electron turnover rate through PSII centers (Behrenfeld et al. 1998; Ragni et al., 2008).' These changes are in Line 600–603 on page 28.

Behrenfeld, M. J., Prasil, O., Kolber, Z. S., Babin, M., Falkowski, P. G. : Compensatory changes in photosystem II electron turnover rates protect photosynthesis from photoinhibition, Photosynth. Res., 58, 259–268, http://doi.org/10.1023/A:1006138630573, 1998.

**L445-447: See my comment regarding competition above.**

Response: We agree with this referee and deleted this sentence in **lines 616–618** on page 28: '*E*. *huxleyi* appeared to be a poor competitor for inorganic nitrate under low levels of nitrate availability (Fig. 1).'

L466-467: looking at the fit on figure 5, I am not convinced by this, as the fit does not run close to the data in the relevant part of the curve (i.e. the slope).

Response: Agreed. In figure 5 (Line 1361 in page 71), we deleted the fitted line.

We changed these contents 'Under light saturation condition, relationship of growth rates of E. huxleyi with phosphate concentrations indicated a very high affinity for dissolved inorganic phosphate (DIP) with 0.04 µmol  $L^{-1}$  half-saturation constant for DIP (Fig. 5).' to 'Furthermore, growth rate of *E*. huxleyi is nearly saturated at 0.25 µmol  $L^{-1}$  DIP and is saturated at 0.5 µmol  $L^{-1}$ DIP and above. This demonstrated that *E*. huxleyi possesses a high affinity for DIP (Fig. 5) which

allowed *E. huxleyi* to take up  $PO_4^{3-}$  efficiently.' These changes are in Lines 646–655 on page 30.

Line 1361:

**Figure 5.** Growth rate of *E. huxleyi* as a function of dissolved inorganic phosphate (DIP) concentration. DIN concentration was 100 µmol  $L^{-1}$  in all culture media, and DIP concentrations were set up to 0.25 µmol  $L^{-1}$ , 0.5 µmol  $L^{-1}$ , 1.5 µmol  $L^{-1}$ , 3 µmol  $L^{-1}$  and 10 µmol  $L^{-1}$  in the culture media. All samples were incubated at 200 µmol photons m-2 s-1 and at 410 µatm *p*CO2 for 4 days, and the values represent the mean ± standard deviation for three replicates. These changes are in Lines 1150–1156 on page 53.

**L480-482: In the natural environment, 10 $\mu$ M NO3 is definitely not "low nutrients".**

Response: We agree with this referee that  $10 \,\mu\text{M}$  NO3- is definitely not "low nutrients".

We changed this text 'In natural waters, E. huxleyi usually starts to bloom following diatom blooms (Tyrrell and Merico, 2004). Therefore, our results also indicate that high growth rate of E. huxleyi at low nutrients concentrations may drive the succession of diatom to E. huxleyi.' to 'In natural seawaters, E. huxleyi usually starts to bloom following diatom blooms (Tyrrell and Merico, 2004), which may be related to high growth rate of E. huxleyi at low nutrient concentrations.' These changes are in Lines 665–669 on page 31.

**L483-485: What is "alkaline phosphate activity"? There seems to be a word missing. Also, please explain why this is relevant.**

Response: We changed 'alkaline phosphate (APase) activity' to 'alkaline phosphatase (APase) activity'. Alkaline phosphatase enzyme cleaves inorganic P from dissolved external organic sources (Dyhrman and Palenik, 2003). In our study, we did not add organic P into seawater. We have deleted ', *and to decrease alkaline phosphatase (APase) activity*' in Lines 671–672 on page 31.

Dyhrman, S. T., and Palenik, B.: Characterization of ectoenzyme activity and phosphate-regulated proteins in the coccolithophorid *Emiliania huxleyi*, J Plank. Res., 25, 1215–1225, https://doi.org/10.1093/plankt/fbg086, 2003.

L487-492: I do not understand how this is related to LP conditions. Wouldn't one expect that P limitation would increase energy demand due to upregulated P uptake machinery?

Response: In addition, at LP, cell volume was 17% larger at HC than at LC under the highest light intensity (Table S1 or R3, *see* below). Large cell volume can directly lead to lower growth rates and reduce nutrient uptake by cells which also limit growth of cells. Another possible reason for low tolerance to high-light intensity in growth rate at LP and HC might be a combined effect of LP and HC on the carbon concentrating mechanism (CCM) of *E. huxleyi*. LP or HC is hypothesized to down-regulate the activity of CCM in the green algae *Chlorella emersonii* and in *E. huxleyi*, respectively (Rost and Riebesell, 2004; Beardall et al. 2005). When grown at HC, LP may minimize the activity of CCM of *E. hukleyi*, which could lead to less energy cost for maintaining high efficient CCM. The saved energy in the HC- and LP-grown cells might have exacerbated photo-inhibition. In summary, high phosphorous requirement, large cell volume and less energy consumption at LP and HC conditions may lead to increased high-light inhibition of growth rates of *E. huxleyi* (Fig. 1). These contents are changed in Lines 679–692 on pages 31 and 32.

In this study, low light intensity not only limited cell growth but also was suggested to limit phosphate uptake rates (Nalewajko and Lee, 1983). In this case, compared to HNHP condition, growth rates of *E. huxleyi* at LP condition were more likely to be limited by low-light intensity (Fig. 1a,c). High light intensity provided energy for cells to take up P, and cells at LP condition need to consume more energy to up-regulate P uptake (Nalewajko and Lee, 1983) which may lead to decreased high-light inhibition of growth rate at LP than at HNHP condition under LC. These changes are in Lines 639–646 on pages 29 and 30.

Beardall, J., Roberts, S., Raven, J. A. : Regulation of inorganic carbon acquisition by phosphorus limitation in the green alga *Chlorella emersonii*, Can. J. Bot., 83, 859–864, https://doi.org/10.1139/b05-070, 2005.

Nalewajko, C., Lee, K. : Light stimulation of phosphate uptake in marine phytoplankton, Mar. Bio., 74, 9–15, https://doi.org./10.1007/BF00394269, 1983.

Rost, B., and Riebesell, U.: Coccolithophores and the biological pump: responses to environmental changes, in: Coccolithophores – From Molecular Biology to Global Impact, edited by: Thierstein, H. R. and Young, J. R., Springer, Berlin, 99–125, https://doi.org/10.1007/978-3-662-06278-4\_52004, 2004.

**L498-499: This can be solely explained by increasing levels of energy saturation of C acquisition and fixation with increasing light.**

Response: Kottmeier et al., (2016) provided a nice explanation for increased carbon acquisition and fixation with increasing light.

At LC, E. huxleyi mainly uses external HCO3 as an inorganic carbon source to synthesize POC

and PIC and increasing light intensity increases the HCO3 uptake rate (Kottmeier et al., 2016)

which results in large POC and PIC production rates at high light intensity (Fig. 3). However, at

HC, expression of gene related to the HCO3- transporter was down-regulated and the HCO3-

uptake rate was reduced (Rokitta et al., 2012; Kottmeier et al. 2016), which lead to lower PIC production rates at HC than at LC. Meanwhile, cells at HC can increase CO2 uptake to compensate for low HCO3-uptake for photosynthetic C fixation (Kottmeier et al., 2016), which explains the

similar POC quotas between HC and LC (Fig. S3). These changes are in Lines 702–711 on pages 32 and 33.

**L506-509: Seems completely unrelated to the presented and discussed data.**

Response: We deleted these contents in Lines 712–714 on page 33: 'LN down regulates expression of the rbcL gene coding for the large subunit of the ribulose-1,5-biphosphate carboxylase/oxygenase (RUBISCO) (Bruhn et al., 2010; Rokitta et al., 2014).'

**L509-511: Seems completely unrelated to previous discussion.**

Response: We changed these contents in Lines 714–724 on page 33: 'To conserve nitrogen, cells at LN prefer to shut down the synthesis of RUBISCO and then reduce carbon fixation (Falkowski et al., 1989) (Fig. 2b)' to 'LN was found to reduce the enzymatic function and cellular metabolic rates, such as reduced synthesis and activity of ribulose-1,5-biphosphate carboxylase/oxygenase (RUBISCO), which decreases POC quota at both LC and HC (Falkowski et al., 1989; Rokitta et al., 2014) (Fig. S3 and S6). Furthermore, in comparison to LC, lower cell division rates at HC further reduce POC production rates at LN. On the other hand, large cell volume at LP and HC condition was responsible for low cell division rate and low POC production rate (Figs 1, 3 and S3).'

L515-527: Here, results from really nutrient-limited cultures are compared to the data from this study without discussing the lack of considerable nutrient-limitation of growth. Please rewrite this section by taking this into consideration. Also, take into account that under intermediate light levels, growth rates under P limitation and LC are as high as in the full media.

Response: We agree with this referee and rewrite this paragraph.

This text 'Müller et al. (2008) found that calcification (PIC production) occurred only in the G1 cell cycle phase, and that LN or LP held cells in the G1 phase longer, which led to larger PIC quotas and calcification rates at LN or at LP than at HNHP (Figs. 2 and S5). LC and LP treatment decreased cell division rates, elongated cell cycle, and increased coccolith production of *E. huxleyi* in the darkness (Paasche and Brubak, 1994). In the present work, however, we found slightly faster cell division (growth) and identical calcification rates at LP and high light intensities (Figs. 1c, 2f and S5). LP has been shown to up-regulate the genes involved in calcium binding proteins such as the glutamic acid related to synthesize of coccolith, calcium homeostasis and transcription factor (cmyb) (Wahlund et al., 2004; Dyhrman et al., 2006), and facilitates the formation of cytoplasmic membrane bodies (Shemi et al., 2016). These are related to the pathways associated with production of coccoliths (Young and Henriksen, 2003) and may also be responsible for larger PIC quotas at LP.' were replaced by 'Nimer and Merrett (1993) reported that decreased DIN concentration facilitates calcification rate of *E. huxleyi*. This is consistent with our result. Due to lower photosynthetic carbon fixation rate and larger calcification rate at LN in comparison to HNHP (Fig. 3), we could expect that at LN, a high proportion of intracellular

HCO3 or CO2 was reallocated to synthesize particulate inorganic carbon. On the other hand, at

LP, slightly larger PIC production rate is likely due to larger cell volume in comparison to HNHP (Fig. 3).' These changes are in Lines 728–747 on pages 33 and 34.

In addition, we provided three reasons for similar growth rates between LP and HNHP at LC and intermediate light levels. These contents were shown in **lines 639–662** on pages 29 and 30: 'In this study, low light intensity not only limited cell growth but also was suggested to limit phosphate uptake rates (Nalewajko and Lee, 1983). In this case, compared to HNHP condition, growth rates of *E. huxleyi* at LP condition were more likely to be limited by low-light intensity (Fig. 1a,c). High light intensity provided energy for cells to take up P, and cells at LP condition need to consume more energy to up-regulate P uptake (Nalewajko and Lee, 1983) which may lead to decreased high-light inhibition of growth rate at LP than at HNHP condition under LC. Furthermore, growth rate of *E. huxleyi* was nearly saturated at 0.25  $\mu$ mol L-1 DIP and was saturated at 0.5  $\mu$ mol L-1 DIP and above. This demonstrated that *E. huxleyi* possesses a high affinity for DIP (Fig. 5) which

allowed *E. huxleyi* to take up  $PO_4^{3-}$  efficiently. Rokitta et al. (2016) showed that even though

 $PO_4^{3-}$  concentration in the culture media declined to zero (undetectable), cell number sustained an increase for 4 days, which indicates that *E. huxleyi* cells could store phosphorus for later use. Consequently, high energy consumption mechanism, efficient uptake and storage capacity for  $PO_4^{3-}$  in *E. huxleyi* could account for no significant differences in growth rates between LP and

HNHP at LC and high light intensities.'

Nalewajko, C., Lee, K. : Light stimulation of phosphate uptake in marine phytoplankton, Mar. Bio., 74, 9–15, https://doi.org./10.1007/BF00394269, 1983.

Rokitta, S. D., von Dassow, P., Rost, B., and John, U.: P- and N-depletion trigger similar cellular responses to promote senescence in eukaryotic phytoplankton, Front. Mar. Sci., 3, 109, https://doi.org/10.3389/fmars.2016.00109, 2016.

L535-536: ETRmax were measured at high light, so it cannot be limited by low energy input. Instead, previous acclimation to low light may have hampered usage of the provided energy. Response: We have deleted these contents in **lines 755–760** on page 35: '*At low light intensities, the*  $ETR_{max}$  values were severely limited by low energy input. Supraoptimal light intensities have *been found to significantly reduce the abundance of several proteins involved in repair and assembly of PSII, such as repair of photodamaged Psb D1 proteins in the reaction center of PSII of E. huxleyi (McKew et al., 2013). These suggest that high light intensity is likely to do great damage to the PSII structure and then reduce the*  $ETR_{max}$ .'

**L541: Please clarify that you have no data on CCM down-regulation but that this is speculation based on previous publications**

Response: We have deleted this text in **lines 760–762** on page 35: '*Especially at HC, supraoptimal light intensity and saved energy from down-regulation of CCM activity synergistically decreased*  $ETR_{max}$  (Fig. 3).'

**L547-550: Of course these processes are correlated. Can you provide something new that further elucidates this fact?**

Response: We have deleted these contents in **lines 763–770** on page 35: 'A previous study found that calcification can be an additional sink for electrons in E. huxleyi (Xu and Gao 2012). Compared with HNHP, larger  $ETR_{max}$  at LN or at LP and at saturating light intensities likely resulted from larger calcification rates (Figs. 2 and 3). On the other hand, growth, photosynthetic carbon fixation and nitrogen uptake need energy originating from electron transport (Zhang et al., 2015). At LP and at limiting levels of light intensity, lower growth, photosynthetic carbon and nitrate assimilation rates coincided with lower  $ETR_{max}$  (Figs. 1–3), implying correlations of these physiological processes.'

**L555-558: I do not understand this line of thought. Please explain in more detail.**

Response: Calcite process within vesicle is shown in equation 1. To calcify, *E. hulxeyi* cells need to take up  $\underline{\text{HCO}_3^-}$  and  $\text{Ca}^{2+}$  from the seawater, which consumes energy. Besides that, they also need to extrude H+ generated during calcification into the cytosol to favour the conversion of  $\underline{\text{HCO}_3^-}$  to  $\underline{\text{CO}_3^-}$ , which also needs some energy. Thus, calcification is a high-energy consumption process, and *E. huxleyi* needs to possess higher light-use efficiencies for their calcification.

**$HCO_3^- + Ca^{2+} \rightarrow CaCO_3^- + H^+$ equation 1.**

The text 'Calcification is an energy-dependent process (Riebesell and Tortell, 2011), and increased calcification rates at low nutrient concentrations could be aided by higher light-use efficiencies (Fig. 4). In addition, besides taking up inorganic carbon sources and  $Ca^{2+}$  from the seawater to calcify, cells need extra energy to expel H+ generated during calcification from the cells (Jin et al., 2017), these may also account for higher light-use efficiencies for PIC production rates.' was replaced by 'To calcify, *E. huxleyi* cells need to take up HCO3- and Ca2+ from the seawater, which consumes energy. Besides that, they also need to extrude H+ generated during calcification into the cytosol to favour the conversion of HCO3- to CO3-, which also needs some energy (Paasche 2002). Thus, calcification is an energy consuming process. To maintain large calcification rate at low nutrient concentration, cells possess high light-use efficiencies and can then obtain more energy to take up HCO3- and Ca2+, and extrude H+ into the cytosol.' These changes are in Lines 773–785 on pages 35 and 36.

Paasche, E. : A review of the coccolithophorid Emiliania huxleyi (Prymnesiophyceae) with particular reference to growth, coccolith formation, and calcification-photosynthesis interactions, Phycologia, 40, 503–529, 2002.

L563-566: The authors correctly state that highly labour-intensive experiments like the

**current one are necessary because interactions between multiple stressors cannot be inferred from isolated effects. I therefore do not understand why they speculate on an interaction they did not investigate.**

Response: Thanks for the comments of this referee. We have deleted these contents in **Lines 804–806** on page 37: '*In comparison to the current ocean environment, under HC and HL conditions as expected in future oceans, effects of LN and LP on carbon fixation of E. huxleyi may partly negate each other (Fig.2, Table 3).*'

**Figure 5 legend: The method description should move into the method section and be more detailed, e.g. were growth rates integrated over 4 days? Were the cultures preacclimated to the conditions? If not, which conditions were they acclimate to before?**

Response: We agree with this referee and moved method description in figure 5 legend to the materials and methods section in Lines 270–280 on page 13.

We added these contents:

**\*2.6** Response of growth rate of *E. huxleyi* to different dissolved inorganic phosphate (DIP) concentrations**

5 L Aquil media were enriched with 100  $\mu$ mol L-1 DIN, aerated for 24 h at 20 °C with air containing 400  $\mu$ atm *p*CO2, sterilized by filtration (0.22  $\mu$ m pore size, Polycap 75 AS, Whatman) and then pumped into autoclaved 250 mL PC bottles. 10  $\mu$ mol L-1, 3  $\mu$ mol L-1, 1.5  $\mu$ mol L-1, 0.5  $\mu$ mol L-1, 0.25  $\mu$ mol L-1 DIP (finial concentration) were respectively added into Aquil media with three replicates at each DIP concentration. 200 cells mL-1 was inoculated to Aquil media and all samples were cultured at 200  $\mu$ mol photons m-2 s-1 for 4 days before starting the experiment. Finial cell concentration was measured by using a Z2 Coulter Particle Count and Size Analyzer (Beckman Coulter).'

**Technical corrections:**

Generally, there are a lot of instances where grammar and wording need to be improved. I strongly suggest the native speakers in the author list to thoroughly correct the final revised version of this manuscript. Below a few examples:

Response: The native speakers in the author list have corrected the grammar and wording in the final revised manuscript.

**L34-35: Please correct/rephrase this sentence.**

Response: This sentence '*High light intensities compensated for inhibition of LP on growth rates at LC, but exacerbated inhibition of LP at HC.*' was replaced by 'At high light intensity, LP did not limit growth rate at LC, but led to increased high-light inhibition of growth rate at HC.' These changes are in **Lines 36–39** on page 2.

**L48-49: Please correct/rephrase this sentence.**

Response: Agreed. This sentence: 'Anthropogenic emission of  $CO_2$  is taken up by the oceans, decreasing pH of seawater and resulting in ocean acidification (OA)' was replaced by 'Rising atmospheric  $CO_2$  level leads to increasing seawater  $CO_2$  concentration and decreasing pH, which is known as ocean acidification (OA).' These changes are in Lines 54–57 on page 3.

L55: Replace "in the UML" by "therein"

Response: In Line 62 on page 3: 'in the UML' was replaced by 'therein'.

**L57-60: Please correct/rephrase this sentence. Why "counteract"?**

Response: We have deleted 'which counteracts with photosynthetic  $CO_2$  fixation,' in Lines 68–69 on page 4.

**L84: Consider replacing "decreased" by "suboptimal"**

Response: 'decreased' was replaced by 'suboptimal' in Line 98 on page 5.

**L86-92: Combine first two sentences into one.**

Response: This text 'Nevertheless, low nutrient concentrations often enhance the PIC quotas of E. huxleyi. This is due to the fact that low nutrient concentrations hold the cells in the G1 cell cycle phase where calcification occurs (Müller et al., 2008). A recent proteome study on E. huxleyi also shows that nutrient limitation arrests cell cycling (McKew et al., 2015). At molecular levels, nitrate or phosphate limitations down-regulate expression of genes involved in cell cycling, RNA and protein synthesis in E. huxleyi (Rokitta et al., 2014, 2016).' were replaced by 'Nevertheless, low nutrient concentrations often enhance the PIC quotas of E. huxleyi, because low nutrient concentrations arrest cell cycling and lengthen the G1 phase where calcification occurs (Müller et al., 2008; McKew et al., 2015).' These changes are in Lines 100–108 on page 5.

Müller, M. N., Antia, A. N., and LaRoche, J.: Influence of cell cycle phase on calcification in the coccolithophore *Emiliania huxleyi*, Limnol. Oceanogr., 53, 506–512, https://doi.org/10.4319/lo.2008.53.2.0506, 2008.

McKew, B. A., Metodieva, G., Raines, C. A., Metodier, M. V., and Geider, R. J.: Acclimation of *Emiliania huxleyi* (1516) to nutrient limitation involves precise modification of the proteome to scavenge alternative sources of N and P, Environ. Microbiol., 17, 4050–4062, https://doi.org/10.1111/1462-2920.12957, 2015.

**L93-101: Indicate at which pCO2 levels these studies were conducted.**

Response: Zhang et al. (2015) reported that at 50–800  $\mu$ mol photons m-2 s-1, 1050  $\mu$ atm CO2 decreased the maximum growth rate, POC and PIC production rate of *Gephyrocapsa oceanica* compared to 510  $\mu$ atm. These changes are in Lines 111–113 on pages 5 and 6.

Under natural solar radiation, Jin et al. (2017) reported that compared to 395  $\mu$ atm, 1000  $\mu$ atm CO2 increased the growth and POC production rates of *E. huxleyi* at high sunlight levels. These changes are in Lines 116–118 on page 6.

**L119: Why "Even"?**

Response: 'Even' was replaced by 'And'in Line 138 on page 7.

**L398-403: This is not discussion later on. Is it needed then?**

Response: Contents in Lines 771–785 on pages 35 and 36 explained why light-use efficiency of POC and PIC production rates was larger than that of growth rates, which is relevant with Lines 533–536 on page 25.

**L142-145: Please correct/rephrase this sentence.**

Response: This sentence in **Lines 162–169** on page 8: '*The synthetic seawater medium Aquil was* prepared according to Sunda et al. (2005), added by 2200  $\mu$ mol L-1 bicarbonate (as opposed to 2380  $\mu$ mol L-1 in the original recipe), in order to reflect the alkalinity in the South and East China Seas of about 2200  $\mu$ mol L-1 (Chou et al., 2005; Qu et al., 2017).' was replaced by 'The Aquil medium was prepared according to Sunda et al. (2005) with the addition of 2200  $\mu$ mol L-1 bicarbonate, resulting in initial concentrations of 2200  $\mu$ mol L-1 total alkalinity (TA). This reflects 2200  $\mu$ mol L-1 alkalinity in the South and East China Seas (Chou et al., 2005; Qu et al., 2017).'

**L158: For clarity, please add "For each nutrient treatment, [:::]"**

Response: We added 'For each nutrient treatment,' in Line 184 on page 9.

**L158-159: Add standard errors for light levels.**

Response: For each nutrient treatment, 20 bottles at each  $pCO_2$  level were incubated at light intensities of 80±5, 120±8, 200±17, 320±16, and 480±30 µmol photons m-2 s-1 of photosynthetically active radiation (PAR) (4 replicates each) measured using a PAR Detector (PMA 2132, Solar Light Company, Glenside). These changes are in **Lines 184–187** on page 9.

**L165-167: Please correct/rephrase this sentence.**

Response: This sentence: 'Bottles were rotated two times per day at 10:00 a.m. and 6:00 p.m. to make the cells can obtain light homogeniously.' was replaced by 'Culture bottles were rotated twice at 10:00 a.m. and 6:00 p.m.,' in Lines 192–195 on page 9.

**L178: "CO2 System" should read "CO2SYS"**

Response: 'CO2 System' was replaced by 'CO2SYS' in Line 211 on page 10.

**L182: "Dickson et al. 2003" should read "Dickson et al (2003)".**

Response: 'Dickson et al. 2003' was replaced by 'Dickson et al. (2003)' in Line 216 on page 10.

L185: "equimolal" should read "equimolar".

Response: 'equimolal' was replaced by 'equimolar' in Line 219 on page 10.

**L186-187: I assume you did not calculate K1 and K2, but used these constants from Roy et al. for your calculations: : : If so, please correct accordingly.**

Response: 'Carbonic acid constants  $K_1$  and  $K_2$  were taken from Roy et al. (1993).' This change is in Lines 221 on page 10.

**L194-196: Please correct/rephrase this sentence.**

Response: 3 mL samples were keptin the dark for 15 min at 20 °C, and  $F_v/F_m$  values were determined at a measuring light intensity of 0.3 µmol photons m-2 s-1 and at a saturation pulse of 5000 µmol photons m-2 s-1 with 0.8 s.' These changes are in **Lines 228–231** on page 11.

L226: Replace "their" by "cellular".

Response: '*their*' was replaced by 'cellular' in Line 266 on page 13.

**L256-264: Estimates of uncertainty are missing.**

**Response:** Table 1 in the original manuscript was replaced by Table S2 in the original supplement in the main text. The text '*The carbonate system parameters (mean values for the beginning and end of incubations) are shown in Table 1. For low CO*2 (*LC*) *condition, the pCO*2 *levels of the media were about 435 µatm at HNHP, 410 µatm at LN and 370 µatm at LP conditions, and the pH*T values (reported on the total scale) were about 8.10 at HNHP, 8.11 at LN and 8.16 at LP. For *high CO*2 (*HC*) *condition, the pCO*2 *levels of the media were about 970 µatm at HNHP, 935 µatm at LN and 850 µatm at LP, and the pH*T values were about 7.80 at HNHP, 7.80 at LN, and 7.85 at *LP conditions.*' was replaced by 'The carbonate system parameters of the seawater at the beginning and end of the incubation are shown in Table 1. Within the low CO2 (*LC*) treatment, *pCO*2 levels of the seawater declined by 16% at HNHP, 19% at LN and 8% at LP, and pH values increased by 0.07 at HNHP, 0.06 at LN and 0.02 at LP (Tukey HSD, all *p* < 0.05). Within the high CO2 (*HC*) treatment, *pCO*2 levels of the seawater declined by 23% at HNHP, 21% at LN and 32% at LP, and pH values increased by 0.1 at HNHP, 0.09 at LN and 0.15 at LP (Tukey HSD, all *p* < 0.05).' These changes are in **Lines 315–328** on page 15.

|      |    |        | p CO 2 | pН                     | TA                   | DIC                  | HCO 3     | CO 3 2- | $CO_2$            | Ω                     |
|------|----|--------|--------------------------|------------------------|----------------------|----------------------|----------------------|-------------------------------|-------------------|-----------------------|
|      |    |        | (µatm)                   | (total                 | (µmol                | (µmol                | (µmol                | (µmol                         | (µmol             | calcite               |
|      |    |        |                          | scale)                 | $L^{-1}$ )           | $L^{-1}$ )           | $L^{-1}$ )           | $L^{-1}$ )                    | $L^{-1}$ )        |                       |
| HNHP | LC | Before | 510±17 a      | 8.04±0.01 a | 2228±17 a | 2004±20 a | 1829±21ª             | 159±2ª                        | 16±1 a | 3.8±0.1ª              |
|      |    | End    | 428±57 b      | $8.11{\pm}0.05^{b}$    | 2225±24 a | 1967±22 b | $1773 \pm 34^{b}$    | 180±18 a           | 14±2 b | 4.3±0.5 a  |
|      | HC | Before | 1210±53 a     | 7.71±0.02 a | 2219±19 a | 2131±22 a | 2010±22 a | 81±2 a             | 39±2 a | 1.9±0.1ª              |
|      |    | End    | 935±139 b     | 7.81±0.06 b | 2225±24 a | 2098±12 b | 1966±17 b | 102±14 b           | 30±4 b | $2.4{\pm}0.3^{b}$     |
| LN   | LC | Before | 483±23 a      | 8.06±0.02 a | 2204±10 a | 1973±10 a | 1796±13 a | 162±6 a            | 16±1 a | 3.9±0.1 a  |
|      |    | End    | 391±39 b      | 8.12±0.03 b | 2123±38 b | 1866±45 b | 1679±48 b | 175±9 b            | 13±1 b | $4.2{\pm}0.2^{b}$     |
|      | HC | Before | 1126±66 a     | 7.73±0.02 a | 2201±3ª              | 2105±7 a  | 1983±9ª              | 85±4 a             | 36±2 a | 2.02±0.1 a |
|      |    | End    | 888±114 b     | $7.82{\pm}0.05^{b}$    | $2142 \pm 38^{b}$    | 2016±47 b | 1890±49 b | $98{\pm}8^{b}$                | $29 \pm 4^{b}$    | 2.4±0.2 b  |
| LP   | LC | Before | 397±16 a      | 8.14±0.02 a | 2248±30 a | 1982±22 a | 1777±17 a | 192±8 a            | 13±1 a | 4.6±0.2 a  |
|      |    | End    | 365±24 b      | 8.16±0.02 a | 2219±20 b | 1942±22 b | 1731±25 b | 199±8ª                        | 12±1 b | 4.8±0.2 a  |
|      | HC | Before | 1140±110 a    | 7.73±0.04 a | 2215±41 a | 2128±46 a | 2005±46 a | 86±7 a             | 37±4 a | $2.1{\pm}0.2^a$       |
|      |    | End    | 780±43 b      | $7.88{\pm}0.02^{b}$    | 2228±14 a | 2084±11 b | 1941±12 b | 117±6 b            | 25±1 b | 2.8±0.1 b  |

Table 1 (S2 in the original supplement). Carbonate chemistry parameters of the seawater at the beginning and end of incubations at different nutrient conditions and  $pCO_2$  levels.

HNHP, 101  $\mu$ mol L-1 dissolved inorganic nitrogen (DIN) and 10.5  $\mu$ mol L-1 dissolved inorganic phosphate (DIP); LN, 8.8  $\mu$ mol L-1 DIN; LP, 0.4  $\mu$ mol L-1 DIP. Different letters represent

statistically differences between the beginning and end of the experiments (Tukey Post hoc, p < 0.05). The values are expressed as mean values with standard deviation for four replicates. These changes are in Lines 1171–1181 on pages 54 and 55.

**L274-279: Units of the treatments are missing.**

Response: At LC, growth rate at LN was similar with that at HNHP under limited light intensity with 80 µmol photons m-2 s-1 (Tukey HSD, df = 1, p = 0.82), and was significantly lower than at HNHP under optimal and supra-optimal light intensities (Tukey HSD, both df = 1, p < 0.01 for 200 µmol photons m-2 s-1; p = 0.005 for 480 µmol photons m-2 s-1). At HC, growth rates at LN were significantly lower than those at HNHP under limited, optimal and supra-optimal light intensities (Tukey HSD, all df = 1, p < 0.01 for 80, 200, 480 µmol photons m-2 s-1). These changes are in Lines 339–345 on page 16.

**L346-349: Replace "At each nutrient condition, at both LC and at HC" by "At all nutrient and CO2 levels".**

Response: 'At each nutrient condition, at both LC and at HC' was replaced by 'At all nutrient and  $CO_2$  levels,' in Lines 414 on page 19.

**L356: Why "both"?**

Response: 'both' indicates 'at both LC and HC'. At both LC and HC, at 80–480 µmol photons m-2 s-1  $F_v/F_m$  did not show significant differences between LN and HNHP (Tukey HSD, all df = 1, all p > 0.05), and at 480 µmol photons m-2 s-1, they were lower at LP than at HNHP at both LC and HC (Tukey HSD, both df = 1, both p < 0.05) (Fig. 2a,c). These changes are in Lines 422–425 on page 20.

**L439-441: This sentence sounds as if the authors would have observed the first statement, and the reference refers to the latter, while the opposite is true. Please rephrase.**

Response: This sentence: 'At HC, the negative effect of high  $[H^+]$  on growth rate was larger than positive effects of increased CO2 and HCO7 concentrations, which could be attributed to lower

growth rates at HC than at LC (Fig. 1) (Bach et al., 2011).' was replaced by 'Lower growth rates at HC than at LC are due to the fact that at HC the negative effect of high  $[H^+]$  on growth rate was larger than positive effects of increased CO2 and HCO3 concentrations (Bach et al., 2011).' These

changes are in Lines 607–612 on page 28.

**L465: "saturation condition, relationship" should read "saturated conditions, the relationship".**

Response: We have deleted this sentence: 'Under light saturation condition, relationship of growth rates of E. huxleyi with phosphate concentrations indicated a very high affinity for dissolved inorganic phosphate (DIP) with 0.04  $\mu$ mol L-1 half-saturation constant for DIP.' in Lines 646–649 on page 30.

L467-471: Please correct/rephrase this sentence.

Response: these contents 'Under light saturation condition, relationship of growth rates of *E*. huxleyi with phosphate concentrations indicated a very high affinity for dissolved inorganic phosphate (DIP) with 0.04  $\mu$ mol L-1 half-saturation constant for DIP (Fig. 5). Since LP was reported to enhance expression of gene with a role in phosphorus assimilation or metabolism and synthesis of inorganic PO43- transporters (Dyhrman et al., 2006; McKew et al., 2015; Rokitta et

al., 2016), which allowed E. huxleyi to take up  $PO_4^{3-}$  efficiently enough, so that LP did not result in reduced growth rate at LC in this study (Fig. 1).' was replaced by 'Furthermore, growth rate of E. huxleyi was nearly saturated at 0.25 µmol L-1 DIP and was saturated at 0.5 DIP and above. This demonstrated that E. huxleyi possesses a high affinity for DIP (Fig. 4), which allowed E. huxleyi to take up  $PO_4^{3-}$  efficiently.' These changes are in Lines 646–655 on page 30.

**L480-482: Please correct/rephrase this sentence.**

Response: In natural seawaters, *E. huxleyi* usually starts to bloom following diatom blooms (Tyrrell and Merico, 2004) which may be related to a high growth rate of *E. huxleyi* at low nutrient concentrations.' These changes are in **Lines 665–669** on page 31.

**L496-502: Please correct/rephrase this sentence.**

Response: This text 'At LC, E. huxleyi mainly uses external  $HCO_3^-$  as an inorganic carbon source for photosynthesis and calcification, and increasing light intensities are able to increase  $HCO_3^-$  uptake rates (Kottmeier et al., 2016). This may explain why POC and PIC quotas and production rates increased with increasing light intensity (Figs. 2 and S5). HC down-regulates gene expression related to the  $HCO_3^-$  transporter (Rokitta et al., 2012) and decreases the

 $HCO_3^-$  uptake rate in E. huxleyi (Kottmeier et al. 2016), leading to lower PIC quotas at HC than at LC (Fig. 2).' were replaced by 'At LC, E. huxleyi mainly uses external  $HCO_3^-$  as an inorganic carbon source to synthesize POC and PIC and increasing light intensity increases the  $HCO_3^$ uptake rate (Kottmeier et al., 2016), which results in large POC and PIC production rates at high light intensity (Fig. 3). However, at HC, expression of gene related to the  $HCO_3^-$  transporter was down-regulated, and the  $HCO_3^-$  uptake rate was reduced (Rokitta et al., 2012; Kottmeier et al. 2016), which lead to lower PIC production rates at HC than at LC.' These changes are in Lines 696–708 on pages 32 and 33.

**L503: omit first "-"**

Response: 'low-HCO3-uptake' was replaced by 'low HCO3-uptake' in Line 709 on page 33.

**L516: Insert "could have" between "which" and "led".**

Response: We have deleted this content 'which led to' in Lines 729 on page 33.

**L553-555: I do not understand this sentence. Please rephrase.**

Response: This sentence 'Calcification is an energy-dependent process (Riebesell and Tortell, 2011), and increased calcification rates at low nutrient concentrations could be aided by higher light-use efficiencies (Fig. 4). In addition, besides taking up inorganic carbon sources and  $Ca^{2+}$  from the seawater to calcify, cells need extra energy to expel H+ generated during calcification from the cells (Jin et al., 2017), these may also account for higher light-use efficiencies for PIC production rates.' was replaced by 'To calcify, *E. hulxeyi* cells need to take up  $HCO_3^-$  and  $Ca^{2+}$  from the seawater, which consumes energy. Besides that, they also need to extrude H+ generated during calcification into the cytosol to favour the conversion of  $HCO_3^-$  to  $CO_3^-$ , which also consumes energy (Paasche 2002). Thus, calcification is an energy comsuming process. To maintain large calcification rates at low nutrient concentration, cells possess high light-use efficiencies and can then obtain more energy to take up  $HCO_3^-$  and  $Ca^{2+}$  and extrude H+ into

the cytosol.' These changes are in Lines 773–785 on pages 35 and 36.

**Figures: Consider using a dashed line for one of the fits to distinguish between the two CO2 levels.**

Response: Thanks for this nice suggestion of this referee. Dashed lines represent the fits at HC in all figures.

**L869: Based on which test?**

Response: 'Different letters showed statistical differences based on the Tukey post hoc test.' These changes are in Lines 1139–1140 on page 52.

**L902: Explain letters to abbreviate pCO2 and light intensity.**

Response: LC represented 410  $\mu$ atm *p*CO2, and light intensity was 200  $\mu$ mol photons m-2 s-1 in Line 1154–1156 on page 53.

This sentence 'All samples were incubated at 200  $\mu$ mol photons m-2 s-1 and at LC for 4 days.' was replaced by 'All samples were incubated at 200  $\mu$ mol photons m-2 s-1 and at 410  $\mu$ atm pCO2 for 4 days, and the values represent the mean  $\pm$  standard deviation for three replicates.' These changes are in Line 1154–1156 on page 53.

**L920-921: Please rephrase to make it a sentence.**

Response: These contents 'All samples were incubated at 200 µmol photons  $m^{-2} s^{-1}$  and at 410 µatm  $pCO_2$  for 4 days. The values represent the mean ± standard deviation for three replicates.' were replaced by 'All samples were incubated at 200 µmol photons  $m^{-2} s^{-1}$  and at 410 µatm  $pCO_2$  for 4 days, and the values represent the mean ± standard deviation for three replicates.' These changes are in Line 1154–1156 on page 53.

Response: PIC:POC ratio is based on weight in figure 3 in Line 1284 on page 65.

**Responses to comments of referee 2 are shown as following:**

The study by Zhang et al. is an important effort to address multifactorial control over the response to acidification of an important phytoplankton species, using an ambitious matrix of treatments. However, there are some major problems that must be resolved, as currently I am unsure of a major portion of the data or results interpretations as presented.

Response: We thank this referee for the positive comments. We refocused on growth rate, POC and PIC production rates. The conclusion of this study are that: (1) low dissolved inorganic nitrogen concentration and high  $CO_2$  level synergistically reduced growth rate; (2) Effect of low dissolved inorganic phosphate concentration on growth rate was modulated by light intensity and  $CO_2$  level; (3) low dissolved inorganic nutrient concentrations (DIN or DIP) and high  $CO_2$  level synergistically reduced POC production rate; (4) low dissolved inorganic nutrient concentrations (DIN or DIP) facilitated PIC production rate. These conclusions have been shown as subsections in the discussion section in the revised BG manuscript.

1. In the Introduction the authors plant the study as if it aims to mimic the natural environment presently or in the future in the laboratory, that is, that the nutrient, light, and CO2 conditions they chose are truly representative. I think this is not unnecessary and risks setting up an incorrect context for interpreting the study results. For example, the authors justify the choice of light regimes in the first paragraph by claiming that phytoplankton in the future ocean will be exposed to higher light levels in the mixed layer, citing two studies. I note also that neither of the studies cited (Gao et al. 2012 and Hutchins and Hu 2017) is relevant to cite, as one is a lab study and the other is a review of lab studies, and neither is a model study predicting average light fields at which phytoplankton will be exposed in the future ocean. In any case, it is difficult to imagine that changes in the stratification of the central ocean basins can only lead to an increase in light exposure. Light exposure is highly dynamic and depends on mixing regime, so yes, the light regime should change, but to model that in the lab with constant light levels is not reasonable. My comment here does not at all negate the study design: Even though we can never mimic the ocean in the lab, it still serves to understand how factors may interact. In the case of trying to predict the response to acidification, it at least serves us to understand how robust the lab results might be for predicting the direction of possible responses, and often as well helps provide insight into mechanisms underlying the responses. I do suggest that they consider revising the Intro.

Response: We agree with this referee that it is difficult to imagine that changes in the stratification of the central ocean basins can only lead to an increase in light exposure. However, it is true that light availability is tied to the mixed layer depth and sea ice fraction, and reduced primary

production correlates with increased stratification in the tropical, southern Pacific and North Atlantic in the CSM1.4 model (Steinacher et al. 2010). Thus, two references: Gao et al. (2012), and Hutchins and Hu (2017) were replaced by Steinacher et al. (2010) in **line 62** on page 3.

Some contents in the introduction were changed (underlined are altered text).

Rising atmospheric CO2 level leads to increasing seawater CO2 concentration and decreasing pH, which is known as ocean acidification (OA) (Caldeira and Wickett, 2003). On the other hand, rising atmospheric CO2 also leads to global and ocean warming, which enhances water column stratification and shoals the upper mixed layer (UML) (Wang et al., 2015). This affects light exposure of phytoplankton dwelling therein (Steinacher et al. 2010). In addition, enhanced stratification reduces the transport of nutrients from deep oceans to the UML (Behrenfeld et al., 2006), which reduces the nutrient concentrations in the UML. These changes are in Lines 54–65 on page 3.

Coccolithophores take up CO2 and/or  $HCO_{1}^{-}$  from seawater for carboxylation, and use  $HCO_{1}^{-}$  for

calcification which produces coccoliths. Calcification processes generate  $CO_2$  due to production of protons, and therefore influencing  $CO_2$  influx into the oceans (Rost and Riebesell, 2004). Growth rate, particulate organic (POC) and inorganic carbon (PIC) production rates of *Emiliania huxleyi*, the most abundant calcifying coccolithophore species, usually display optimum responses to a broad range of  $CO_2$  concentration (Bach et al., 2011). Growth, POC and PIC production rates could increase, decrease and be unaffected by rising  $CO_2$  treatments across a narrow  $CO_2$  range, which is dependent on the optimal  $CO_2$  levels of these physiological rates and the selected  $CO_2$ range (Langer et al., 2009; Richier et al., 2011; Bach et al., 2015; Jin et al., 2017). Differences in sampling locations, experimental setups, deviations in the measuring methods and intraspecific differences can generally be responsible for the differential responses of growth, POC and PIC productions to rising  $CO_2$  in *E. huxleyi* (Langer et al., 2009; Meyer and Riebesell, 2015). These changes are in **Lines 66–84** on pages 3 and 4.

These contents '*This is due to the fact that low nutrient concentrations hold the cells in the G1 cell cycle phase where calcification occurs (Müller et al., 2008). A recent proteome study on E. huxleyi also shows that nutrient limitation arrests cell cycling (McKew et al., 2015). At molecular levels, nitrate or phosphate limitations down-regulate expression of genes involved in cell cycling, RNA and protein synthesis in E. huxleyi (Rokitta et al., 2014, 2016).' were replaced by 'because low nutrient concentrations arrest cell cycling and lengthen the G1 phase where calcification occurs (Müller et al., 2008; McKew et al., 2015).' in Lines 101–108 on page 5.*

Recently, several studies investigated interactive effects of rising CO2 and light intensity on physiological rates of coccolithophores (Feng et al., 2008; Jin et al., 2017). Zhang et al. (2015) reported that at 50–800 µmol photons m-2 s-1, 1050 µatm CO2 decreased the maximum growth rate, POC and PIC production rates of *Gephyrocapsa oceanica* compared to 510 µatm. At low light levels, coccolithophores increase CO2 uptake to compensate for inhibition of  $HCO_3^-$  uptake on photosynthesis, while at high light intensity they don't increase CO2 uptake (Kottmeier et al.,

2016). Under natural solar radiation, Jin et al. (2017) reported that compared to 395  $\mu$ atm, 1000  $\mu$ atm CO\_2-increased the growth and POC production rates of *E. huxleyi* at high sunlight levels. These indicate that during growth under different experimental conditions, rising CO2 showed contrasting effects on growth and POC production rates of *E. huxleyi* and *G. oceanica*. These changes are in **Lines 109–122** on pages 5 and 6.

Steinacher, M., Joos, F., Frö licher, T. L., Bopp, L., Cadule, P., Cocco, V., Doney, S. C., Gehlen, M., Lindsay, K., Moore, J. K., Schneider, B., Segschneider, J. : Projected 21st century decrease in marine productivity: a multi-model analysis, Biogeosciences, 7, 979–1005, 2010.

**2. There is at least one major problem with the growth rates reported, possibly many more:**

a. It makes no sense to report a single growth rate as the response to nutrient-limitation in batch culture experiments. At inoculation of cultures, cells should be nutrient replete even in the LP and LN conditions. If they have been "acclimated" to growing previously in the same media, the inoculums likely are from cultures that have already exhausted the phosphate (in LP) or nitrate (in LN), so the cells will have to re-configure nutrient uptake and metabolism, begin to grow, then exhaust the nutrients, re-configuring nutrient and connected metabolism again. The growth rate most certainty will NOT be constant. A recent study where these effects can be seen would be that of Rokitta et al. (2014). The authors report only a single growth rate, not changes in cell density over time, no indication of when nutrient limitation may start nor how long cells have been in nutrient limited conditions. In this sense, a good study to look at would be the recent one by Müller et al. (2017) using a continuous culture approach to understand the interaction/independence of nutrient limitation and acidification effects (curiously, the authors cite the study in the Intro but do not discuss at all, despite its central relevance!). The results presented in the current manuscript are therefore completely uninterpretable.

Response: Thanks for the comments and suggestions of this referee. To prevent seawater-air  $CO_2$  exchange, incubation bottles were filled with seawater with no headspace and tightly closed during incubations. This is one of the reasons for measuring cell concentration at the end of the incubation and reporting a single growth rate. More importantly, studies of Rokitta et al. (2014) reported that cell number of *E. huxleyi* increased exponentially on the third to sixth days during incubation, and showed that growth rates were similar at the fourth, fifth and sixth days. We agree with this referee that growth rates are not constant, however, variation in growth rates at different days were much lower than variation in growth rates between different treatments.

**Low DIN and DIP concentration did not limit growth in this study, the reasons are that:**

In this study, growth rates of *E. huxleyi* were larger than 1 in almost all treatments, and cells divided 1–2 times per day (Fig .1), which indicates non-limiting nutrient conditions during the incubation. Based on measured PON quota and cell concentration in this study (Figs. 1 and S6 in the manuscript), PON concentrations at the end of incubations were estimated to be 7.8–9.3 µmol  $L^{-1}$  at different nutrient conditions (Table S2). These data were closely correlated with molar drawdown of dissolved inorganic nitrogen (DIN) during the incubations. Furthermore, residual 1 µmol  $L^{-1}$  DIN in the final day of the incubation showed non-limitation of growth and POC production rates by nitrogen. On the other hand, Rokitta et al. (2016) reported that  $F_y/F_m$  of *E*.

huxleyi was 50% lower at P-depleted than at P-replete conditions. In this study,  $F_v/F_m$  and POC quota were very similar between LP and HNHP treatments (Figs. 2 and S3), which suggests that LP did not limit growth and carbon fixation. This text was added in the first paragraph in the discussion section in Lines 568–579 on pages 26 and 27.

Comparison between the study of Müller et al. (2017) and ours are shown as following:

Using a chemostat culture, Müller et al. (2017) reported that DIN or DIP limitation decreased the POC and PIC production rates (in pg C cell-1 d-1) by 50% and rising  $pCO_2$  levels did not affect POC production rates. However, when normalized to cell volume, nutrient limitation did not affect POC and PIC production rates (in pg C cellV-1 d-1), and rising  $pCO_2$  levels reduced POC and PIC production rates. In our study, decreased DIN or DIP concentration reduced the normalized POC production rates (in pg C cellV-1 d-1), and increased the normalized PIC production rates at both LC and HC (Fig. S5). Differing results between the study of Müller et al. (2017) and ours may result from different experimental setup. Growth was really limited by N or P, cells were cultured in a continuous photon flux, and cell growth was in the stable phase when POC and PIC samples were taken in the study of Müller et al. (2017). While we took POC and PIC samples in the exponential growth phase, and LN or LP did not really limit growth of *E. huxleyi* in our study. These contents were added in the discussion section in Lines 786–798 on pages 36 and 37.

Rokitta, S. D., von Dassow, P., Rost, B., and John, U.: *Emiliania huxleyi* endures N-limitation with an efficient metabolic budgeting and effective ATP synthesis, BMC Genomics, 15, 1051–1064, https://doi.org/10.1186/1471-2164-15-1051, 2014.

Müller, M. N., Trull, T. W., and Hallegraeff, G. M.: Independence of nutrient limitation and carbon dioxide impacts on the Southern Ocean coccolithophore *Emiliania huxleyi*, ISME J., 11, 1777–1787, https://doi.org/10.1038/ismej.2017.53, 2017.

---

## Author Comment (AC2) · 14 Apr 2018

Dear Referees,

We thank you for your supportive comments and the constructive reviews on our manuscript. Our detailed responses in blue text to your comments are attached. **The changed contents in the revised manuscript are underlined.**

**Responses to comments of referee 1 are shown as following:**

General comments:
The present manuscript presents a comprehensive dataset investigating the interactive effects of carbonate chemistry, light intensities and nutrient availabilities on the coccolithophore E. huxleyi. The dataset consists of an impressively high number of treatments, replicates and measured parameters therein. Given the fact that interaction of multiple drivers are often impossible to predict from simpler experiments, datasets as this one are indispensable to understand the expected natural complexity in climate change effects.
Response: We thank this referee for his (her) kind words.

This vast amount of data is, however, somewhat overwhelming and it seems to me that the authors also got lost in it a bit. In its current form, the manuscript is poorly written (also with respect to language) and lacks a story line. While I do acknowledge the difficulty to find overarching patters in such a complicated dataset, the current manuscript does not make it easy for the reader to take home any conclusions. In the discussion, individual paragraphs are often not connected to each other (and sometimes even not between sentences therein). I suggest the authors to focus on the main aspects they want to interpret and discuss these in more than a few sentences, and to omit some of the other (side-)aspects. Likewise, parameters that do not get discussed in detail also do not need to be described in great detail in the (currently quite long) results section. In my opinion, quite some of this information could be sufficiently described in tables and the supplement.
Response: We agree with the suggestions of this referee. The manuscript has been refocused on growth rate, POC and PIC production rates, and fitted alpha ($a$) and maximum values for growth, POC and PIC production rates. We omitted a description of the rETR. The take home conclusions are that: (1) low dissolved inorganic nitrogen (LN) concentration and high $CO_2$ level synergistically reduced growth and POC production rate; (2) At high light intensity, low dissolved inorganic phosphate (LP) concentration did not limit growth rate at LC but led to increased high-light inhibition of growth rate at HC; (3) low nutrient concentrations (DIN or DIP) increased the maximum value and the light-use efficiencies of calcification rate. These changes are in **Lines 35–39** and **Lines 43–44** on page 2.

With respect to the general interpretation of the data, I disagree with the way the nutrient treatments are regarded. Despite the fact that cells divided 1-2 times per day ($\mu$>1 in almost all cases) and were clearly exhibiting non-limited exponential growth, the data is discussed as if the cells were nutrient limited and compared to previous studies that investigated strong nutrient limitation. Regarding nitrogen limitation, for example, residual DIN was 1.0 0.4 $\mu$mol L-1 in LN treatments, which is known to not limit growth, and the molar drawdown in HN and LN treatments is actually similarly high. The same is true for the molar drawdown of DIP. Thus, the discussion needs to be refocussed by considering different but not strongly limiting nutrient concentrations rather that limiting vs. non-limiting conditions. This is particularly the case as growth rates are integrated over the whole duration of the experiment (i.e. mixing phases of non-limited growth with potential limitation towards the end of the experiment), while photophysiological measurements are only taken at the final (potentially more limited) stage.

Response: Thanks for the important and supportive comments of this referee. We agree with this referee that growth, POC and PIC production rates of cells were not limited by low DIN or DIP concentration. We refocused on differences in growth, POC and PIC production rates caused by high and low nutrient concentrations rather than on liming and non-limiting nutrient conditions.

For different growth rates between HNHP and LN conditions, we described that:
LN concentration was shown to down-regulate transcripts of genes related to nitrate reductase (NRase) activity, synthesis of amino acids, RNA polymerases and nitrogen metabolism in *E. huxleyi* (Bruhn et al., 2010; Rouco et al., 2013; Rokitta et al., 2014), which led to lower overall biosynthetic activity and decreased the growth rates (Fig. 1). These changes are in **Lines 620–625** on page 29.

For subsection in the discussion section: 'Effect of low dissolved inorganic phosphate concentration on growth rate was modulated by light intensity and $CO_2$ level', we described that:
1. In this study, low light intensity not only limited cell growth but also was suggested to limit phosphate uptake rates (Nalewajko and Lee, 1983). In this case, compared to the HNHP condition, growth rates of *E. huxleyi* at LP condition were more likely to be limited by low-light intensity (Fig. 1a,c). High light intensity provided energy for cells to take up P, and cells at LP condition need to consume more energy to up-regulate P uptake (Nalewajko and Lee, 1983) which may lead to decreased high-light inhibition of growth rate at LP than at HNHP condition under LC. Furthermore, growth rate of *E. huxleyi* was nearly saturated at 0.25 μmol $L^{-1}$ DIP and was saturated at 0.5 μmol $L^{-1}$ DIP and above. This demonstrated that *E. huxleyi* possesses a high affinity for DIP (Fig. 5) which allowed *E. huxleyi* to take up $PO_4^{3-}$ efficiently. Rokitta et al. (2016)

showed that even though $PO_4^{3-}$ concentration in the culture media declined to zero (undetectable), cell number sustained to increase for 4 days, which indicates that *E. huxleyi* cells could store phosphorus for later use. Consequently, high energy consumption mechanisms, efficient uptake and storage capacity for phosphorus in *E. huxleyi* could account for there being no significant differences in growth rates between LP and HNHP at LC and high light intensities. These changes are in **Lines 639–662** on pages 29 and 30.

2. Rising $CO_2$ was found to lead to higher phosphorous requirements for growth, carbon fixation and nitrogen uptake in *E. huxleyi* (Matthiessen et al., 2012; Rouce et al., 2013). At HC, higher phosphorous requirements may lead to lower growth rates at LP in comparison to HNHP (Fig. 1a,c). In addition, at LP, cell volume was 17% larger at HC than at LC under the highest light intensity (Table S1). Large cell volume can directly lead to lower growth rates and reduce nutrient uptake by cells, thereby limiting growth Another possible reason for low tolerance to high-light intensity in growth rate at LP and HC might be a combined effect of LP and HC on the carbon concentrating mechanism (CCM) of *E. huxleyi*. LP or HC is hypothesized to down-regulate the activity of CCM in the green algae *Chlorella emersonii* and in *E. huxleyi*, respectively (Rost and Riebesell, 2004; Beardall et al. 2005). When grown at HC, LP may minimize the activity of CCM of *E. hulxeyi*, which could lead to less energy cost for maintaining high efficient CCM. The saved energy in the HC- and LP-grown cells might have exacerbated photo-inhibition. In summary, high phosphorous requirement, large cell volume and less energy consumption at LP and HC conditions may lead to increased high-light inhibition of growth rates of *E. huxleyi* (Fig. 1). These changes are in **Lines 670–692** on pages 31 and 32.

Nalewajko, C., Lee, K. : Light stimulation of phosphate uptake in marine phytoplankton, Mar. Bio., 74, 9–15, https://doi.org./10.1007/BF00394269, 1983.
Beardall, J., Roberts, S., Raven, J. A. : Regulation of inorganic carbon acquisition by phosphorus limitation in the green alga *Chlorella emersonii*, Can. J. Bot., 83, 859–864, https://doi.org/10.1139/b05-070, 2005.

Specific comments:
L33-34: The interaction between CO2 and N is actually the least significant term, why do you focus on this interaction and not the others?
Response: Synergistic effects of low dissolved inorganic nitrogen (DIN) concentration and high $CO_2$ level on growth and POC production rates are one of main results in this study. We also refocused on interactive effects of DIP concentration, $CO_2$ level and light intensity on growth and POC production rates, and effect of low nutrient concentrations on PIC production rate.
This sentence '*HC and LN synergistically decreased growth rates of E. huxleyi at all light intensities.*' were replaced by 'LN and HC synergistically reduced growth and POC production rates.' These changes are in **Line 34–36** on page 2.

L36-37: The authors do not provide any data that would allow to conclude on the competitive abilities of this species. If they want to, they would need to either conduct competition experiments, or compare nutrient uptake kinetics with those of competing species.
Response: We thank to this referee for their suggestions.
This sentence '*These results indicate that the ability of E. huxleyi to compete for nitrate and phosphate may be reduced in future oceans with high CO₂ and high light intensities.*' was replaced by 'These results showed that effects of nutrient concentrations on physiological rates of *E. huxleyi* were modulated by $CO_2$ level and light intensity.' These changes are in **Lines 39–42** on page 2.

L56: Why only from media, not generally from seawater?

Response: 'Coccolithophores take up $CO_2$ and/or $HCO_3^-$ from seawater for carboxylation'. These changes are in **Line 66** on page 3.

L60-65: I do not understand why the authors mention two opposing interpretations of multiple stressor effects (i.e. linearly increasing/decreasing/non-affected vs. optimum curve response) without clarifying why they use the linear trends even though they are aware of the fact the responses follow more complex optimum curves.

Response: The text '*Growth rate, particulate organic (POC) and inorganic carbon (PIC) production rates of Emiliania huxleyi, the most abundant calcifying coccolithophore species, usually display optimum responses to a broad range of $CO_2$ concentration, with growth, POC and PIC production rates increased, decreased or unaffected by rising $CO_2$ treatments (Langer et al., 2009; Richier et al., 2011; Bach et al., 2015; Jin et al., 2017).*' was replaced by 'Growth rate, particulate organic (POC) and inorganic carbon (PIC) production rates of Emiliania huxleyi, the most abundant calcifying coccolithophore species, usually display optimum responses to a broad range of $CO_2$ concentration (Bach et al., 2011). Growth, POC and PIC production rates could increase, decrease and be unaffected by rising $CO_2$ treatments across a narrow $CO_2$ range, which is dependent on the optimal $CO_2$ levels of these physiological rates and the selected $CO_2$ range (Langer et al., 2009; Richier et al., 2011; Bach et al., 2015; Jin et al., 2017).' These changes are in **Lines 70–78** on page 4.

L65-67: Really? These is also plenty of evidence for the opposite effect, also published by some of the authors.

Response: We deleted this sentence '*Increased light levels could counteract the negative effects of rising $CO_2$ on calcification in E. huxleyi when grown under natural fluctuating sunlight (Jin et al., 2017).*' These changes are lin **Lines 78–80** on page 4.

L67-70: Intraspecific differences are another well-established reason for differing responses (e.g. Langer et al. 2009).

Response: Differences in sampling locations, experimental setups, deviations in the measuring methods and intraspecific differences can generally be responsible for the differential responses of growth, POC and PIC productions to rising $CO_2$ in *E. huxleyi* (Langer et al., 2009; Meyer and Riebesell, 2015). These changes are in **Lines 80–84** on page 4.

Langer, G., Nehrke, G., Probert, I., Ly, J., and Ziveri, P.: Strain-specific responses of *Emiliania huxleyi* to changing seawater carbonate chemistry, Biogeosciences, 6, 2637–2646, https://doi.org/10.5194/bg-6-2637-2009, 2009.

L75: Photo-acclimation to HL or LL? Both are photo-acclimative processes

Response: Reduction in pigment content and effective photochemical quantum yield ($F_v^{'}/F_m^{'}$) are characteristics of photo-acclimation to high light intensity (Geider et al., 1997; Gao et al., 2012). These changes are in **Lines 89** on page 4.

L86-92: The same information is presented in the discussion. Is it really necessary to present it twice with the same level of detail?

Response: This text '*Nevertheless, low nutrient concentrations often enhance the PIC quotas of E. huxleyi. This is due to the fact that low nutrient concentrations hold the cells in the G1 cell cycle phase where calcification occurs (Müller et al., 2008). A recent proteome study on E. huxleyi also shows that nutrient limitation arrests cell cycling (McKew et al., 2015). At molecular levels, nitrate or phosphate limitations down-regulate expression of genes involved in cell cycling, RNA and protein synthesis in E. huxleyi (Rokitta et al., 2014, 2016).*' were replaced by 'Nevertheless, low nutrient concentrations often enhance the PIC quotas of *E. huxleyi* because low nutrient concentrations arrest cell cycling and lengthen the G1 phase where calcification occurs (Müller et al., 2008; McKew et al., 2015).' These changes are in **Lines 100–108** on page 5.

L180: How did you measure the pressure inside the syringe filter?
Response: We cannot measure the pressure inside the syringe filter. But we used an instrument to pump seawater, which was filtered by the syringe filter. The pressure of the pump was 200 mbar.

In the final days of incubation, 25 mL samples for TA measurements were filtered (0.22 μm pore size, Syringe Filter) by gentle pressure with 200 mbar in the pump (GM-0.5A, JINTENG) and stored at 4 $^{\circ}$C for a maximum of 7 days. These changes are in **Lines 214** on page 10.

L193: How similar were the PAM light values to those during the incubation? Please provide a quantitative comparison.
Response: PAM light values are shown in the table R1. But we deleted the description of ETR in lines **232 –243** on page 11.

Table R1. Comparison between PAM light values and incubation light intensity

| Light values (μmol photons m$^{-2}$ s$^{-1}$) | 1 | 2 | 3 | 4 | 5 | 6 | 7 | 8 | 9 |
|---|---|---|---|---|---|---|---|---|---|
| PAM light | 42 | 92 | 133 | 210 | 300 | 450 | 850 | 1126 | 1600 |
| Incubation light | | 80 | 120 | 200 | 320 | 480 | | | |

L198-203: How was the "cellular absorption value" determined? This parameter most likely changes strongly with light-acclimation, so I do not think that one constant value can be used to convert relative ETR to absolute ones for all treatments. If the authors did not determine this values for each treatment, they should rather report the ETR in their relative unit.
Response: We agree with this referee that cellular absorption value changes strongly with light-acclimation. But we deleted the description of ETR in lines **232 –243** on page 11.

L214: Is it really true that the authors did not even measure the initial cell count but just assumed inoculation to be perfectly equal among all bottles? I do not trust the growth rate estimates at all if this is the case, especially as small differences in the low abundance range will have huge effects on the final counts.
Response: The bottles were filled with Aquil with no headspace to minimize gas exchange. The volume of the inoculum was calculated (see below) and the same volume of Aquil was taken out from 500 mL bottles before inoculation. These changes are in **Lines 179–182** on page 9.

There was 625 ml seawater in the 500 ml polycarbonate (PC) bottles. Before cells were inoculated to new seawater, finial cell concentrations ($C_0$) were measured. Then we calculated the inoculated volumes (V) according to V = (200 cell/ml x 625 ml)/$C_0$. And we don't think this method cause errors.

L234-235: Why were the two nutrient treatments analysed separately?

Response: We re-analyzed the data with a 3-way ANOVA, which shows individual and interactive effects of nutrient concentration, $CO_2$ level and light intensity, and compares differences among HNHP, LN and LP conditions.

A three-way ANOVA was used to determine the main effect of dissolved inorganic nutrient concentration, $p CO_2$, light intensity and their interactions for these variables. A two-way ANOVA was performed to test the main effect of dissolved inorganic nutrient concentration, $p CO_2$ and their interactions on fitted $a$ and $V_{max}$ of growth, POC and PIC production rates. When necessary, a Tukey Post hoc (Tukey HSD) test was used to identify the differences between two $CO_2$ levels, nutrient concentrations or light intensities. These changes are in **Lines 290–298** on page 14.

**Table 2.** Results of three-way ANOVAs of the impacts of dissolved inorganic nutrient concentration, $p CO_2$, light intensity and their interaction on growth rate, $F_v/F_m$, $F_v'/F_m'$, POC and PIC production rates, and PIC:POC ratio.

|  | Factor | F value | p value |
|---|---|---|---|
| Growth rate (d$^{-1}$) | Nut | 264.7 | <0.01 |
|  | C | 875.6 | <0.01 |
|  | L | 2035.8 | <0.01 |
|  | Nut×C | 53.6 | <0.01 |
|  | Nut×L | 84.2 | <0.01 |
|  | C×L | 9.3 | <0.01 |
|  | Nut×C×L | 26.8 | <0.01 |
| $F_v/F_m$ | Nut | 68.6 | <0.01 |
|  | C | 184.7 | <0.01 |
|  | L | 225.8 | <0.01 |
|  | Nut×C | 10.3 | <0.01 |
|  | Nut×L | 8.1 | <0.01 |
|  | C×L | 15 | <0.01 |
|  | Nut×C×L | 5.2 | <0.01 |
| $F_v'/F_m'$ | Nut | 63.9 | <0.01 |
|  | C | 181.8 | <0.01 |
|  | L | 1161.8 | <0.01 |
|  | Nut×C | 51.9 | <0.01 |
|  | Nut×L | 15.3 | <0.01 |
|  | C×L | 9.9 | <0.01 |
|  | Nut×C×L | 8.1 | <0.01 |
| POC production rate | Nut | 11.8 | <0.01 |

| | | | |
|---|---|---|---|
| (pg C cell$^{-1}$ d$^{-1}$) | C | 128.9 | <0.01 |
| | L | 293.7 | <0.01 |
| | Nut×C | 4.9 | =0.01 |
| | Nut×L | 19.0 | <0.01 |
| | C×L | 8.47 | <0.01 |
| | Nut×C×L | 1.94 | =0.06 |
| PIC production rate | Nut | 624.4 | <0.01 |
| (pg C cell$^{-1}$ d$^{-1}$) | C | 142.0 | <0.01 |
| | L | 147.2 | <0.01 |
| | Nut×C | 1.9 | =0.16 |
| | Nut×L | 17.3 | <0.01 |
| | C×L | 8.1 | <0.01 |
| | Nut×C×L | 4.6 | <0.01 |
| PIC:POC ratio | Nut | 326.7 | <0.01 |
| | C | 57.7 | <0.01 |
| | L | 41.8 | <0.01 |
| | Nut×C | 8.3 | <0.01 |
| | Nut×L | 12.5 | <0.01 |
| | C×L | 4.0 | <0.01 |
| | Nut×C×L | 3.3 | <0.01 |

Nut, dissolved inorganic nutrient concentrations (μmol L$^{-1}$); C, $pCO_2$ (μatm); L, light intensity (μmol photons m$^{-2}$ s$^{-1}$); POC and POC production rates, particulate organic and inorganic carbon production rates; $F_v/F_m$, maximum photochemical quantum yield; $F_v'/F_m'$, effective photochemical quantum yield. These changes are in **Lines 1198–1209** on pages 56–58.

**Table 4.** Results of two-way ANOVAs of the effects of dissolved inorganic nutrient concentration and $pCO_2$ on fitted $a$ and maximum value ($V_{max}$) of growth, POC and PIC production rates. More detailed information is given as in Table 2. These changes are in **Lines 1246–1249** on page 62.

| | | Factor | $F$ value | $p$ value |
|---|---|---|---|---|
| $a$ | Growth rate | Nut | 18.08 | <0.001 |
| | | CO$_2$ | 0.186 | 0.6711 |
| | | Nut×CO$_2$ | 0.398 | 0.6776 |
| | POC production rate | Nut | 7.21 | 0.005 |
| | | CO$_2$ | 7.78 | 0.0121 |
| | | Nut×CO$_2$ | 2.50 | 0.11 |
| | PIC production rate | Nut | 21.73 | <0.001 |
| | | CO$_2$ | 2.32 | 0.145 |
| | | Nut×CO$_2$ | 2.56 | 0.105 |
| $V_{max}$ | Growth rate | Nut | 24.9 | <0.001 |
| | | CO$_2$ | 572.7 | <0.001 |
| | | Nut×CO$_2$ | 14.8 | <0.001 |
| | POC production rate | Nut | 7.301 | 0.0048 |

|  | CO$_2$ | 15.95 | 0.0009 |
|---|---|---|---|
|  | Nut×CO$_2$ | 1.91 | 0.177 |
| PIC production rate | Nut | 56.06 | <0.001 |
|  | CO$_2$ | 86.84 | <0.001 |
|  | Nut×CO$_2$ | 0.168 | 0.85 |

L266 ff.: It is no clear to me to which of the two tests (i.e. ANOVA vs. post hoc tests) the statements regarding the p values refer to. These are two different things. Please clearly state if you base a statement of "significance" on the ANOVA itself or a posthoc test in the whole results section. If you describe an optimum-curve behaviour, for example, the ANOVA cannot capture both increasing and decreasing phases of it, but would indicate that one of the two is more dominant.

Response: p value in Table 2 (*see* above) in the manuscript refers to ANOVA, and p value in the results section refers to Tukey post hoc test. All Tukey Post hoc test in the results section were stated by 'Tukey HSD'. Alpha (*a*) and maximum value ($V_{max}$) of growth, POC and PIC production rates (optimum-curve behaviour) were calculated from fitted parameters *a*, *b* and *c* based on model of Eilers and Peeters (1988). And a two-way ANOVA was used to test effects of nutrient concentration and CO$_2$ level on *a* and $V_{max}$ (Table 4).

The apparent light use efficiency, the slope (*α*), for each light response curve was estimated as $α = 1/c$. The maximum values ($V_{max}$) of growth, POC and PIC production rates were calculated according to $V_{max} = \dfrac{1}{b + 2\sqrt{ac}}$. These changes are in **Lines 286–289** on pages 13 and 14.

Eilers, P., and Peeters, J.: A model for the relationship between light intensity and the rate of photosynthesis in phytoplankton, Ecol. Model., 42, 199–215, https://doi.org/10.1016/0304-3800(88)90057-9, 1988.

L412 ff: Quite often, single sentences are not clearly connected. The discussion thus seems like a long list of ideas, but without any structure or line of thought.

Response: In the discussion section, we refocused on: (1) low dissolved inorganic nitrogen concentration and high CO$_2$ level synergistically reduced growth rate; (2) Effect of low dissolved inorganic phosphate concentration on growth rate was modulated by light intensity and CO$_2$ level; (3) low dissolved inorganic nutrient concentration (DIN or DIP) and high CO$_2$ level synergistically reduced POC production rate; (4) low nutrient concentrations (DIN or DIP) facilitated PIC production rate. These topics have been shown as subsections of the discussion section in the revised BG manuscript.

L414-416: Why "synergistic negative effects"? This a priory expectation is not stated (nor argued for) in the intro.

Response: As shown in Fig. 4 in the revised manuscript, maximum growth rates were significantly lower at LN than at HNHP under both LC and HC; and they were lower at HC than at LC. So LN and HC synergistically reduced growth rates. Previous studies generally reported effects of low nutrient concentration and rising $CO_2$ on POC quota, so this expectation is not stated in the introduction (Sciandra et al., 2003; Rouco et al., 2013). We deleted these contents in **Lines 581–583** on page 27.

Sciandra, A., Harlay, J., Lefévre, D., Lemée, R., Rimmelin, P., Denis, M., and Gattuso, J. P.: Response of coccolithophorid *Emiliania huxleyi* to elevated partial pressure of $CO_2$ under nitrogen limitation, Mar. Ecol. Prog. Ser., 261, 111–122, https://doi.org/10.3354/meps261111, 2003.
Rouco, M., Branson, O., Lebrato, M., and Iglesias-Rodríguez, M. D.: The effect of nitrate and phosphate availability on *Emiliania huxleyi* (NZEH) physiology under different $CO_2$ scenarios, Front. Microbiol., 4, 155, https://doi.org/10.3389/fmicb.2013.00155, 2013.

**Line 1322:**

[Figure]

Figure 4. At both LC and HC, fitted $a$ (**a**) and maximum (**b**) of growth rate at HNHP, LN and LP conditions. At both LC and HC, fitted $a$ (**c**) and maximum (**d**) of POC prodution rate at HNHP, LN and LP conditions. At both LC and HC, fitted $a$ (**e**) and maximum (**f**) of PIC production rate at HNHP, LN and LP conditions. $\alpha$ was the slope of fitted lines for growth, POC and PIC production rates. Different letters showed statistical differences based on the Tukey post hoc test. The values represent the mean ± standard deviation for four replicates. These changes are in **Lines 1135–1141** on page 52.

L423-430: What does the content of this paragraph mean for the interpretation of the results with respect to nutrient limitation?
Response: In order to show that growth of *E. huxleyi* is in the exponential phase at the fourth to sixth days during culturing, we cited Langer et al. (2013).

These contents in **Lines 590–597** in page 27: 'Langer et al. (2013) detected that growth of cell on the fourth to sixth days during cultures was in the exponential phase even at 3 µmol L$^{-1}$ $NO_3^-$ or at 0.29 µmol L$^{-1}$ $PO_4^{3-}$ with the same *E. huxleyi* strain. In this study, all parameters were measured on the fourth to the sixth days, and it is most likely that cells at all treatments were sampled in the exponential growth phase' were transferred to the materials and methods section in **Lines 197–202** on pages 9 and 10.

L435-439: This could be explained by an excess of PSII reaction centres (Behrenfeld et al. 1998).
Response: We thank to this referee for his (her) nice suggestion. At high light intensity, increases in electron turnover rate through PSII can protect photosynthesis from photoinhibition. Once electron turnover rate started to decrease after it maximized, light-saturated photosynthetic rates decreased.
'because high light intensity can constantly damage the reaction centers of photosystem II (PSII) of *E. huxleyi* (Fig. 2) and maximize electron turnover rate through PSII centers (Behrenfeld et al. 1998; Ragni et al., 2008).' These changes are in **Line 600–603** on page 28.

Behrenfeld, M. J., Prasil, O., Kolber, Z. S., Babin, M., Falkowski, P. G. : Compensatory changes in photosystem II electron turnover rates protect photosynthesis from photoinhibition, Photosynth. Res., 58, 259–268, http://doi,org/10.1023/A:1006138630573, 1998.

L445-447: See my comment regarding competition above.
Response: We agree with this referee and deleted this sentence in **lines 616–618** on page 28: '*E. huxleyi* appeared to be a poor competitor for inorganic nitrate under low levels of nitrate availability (Fig. 1).'

L466-467: looking at the fit on figure 5, I am not convinced by this, as the fit does not run close to the data in the relevant part of the curve (i.e. the slope).
Response: Agreed. In figure 5 (Line 1361 in page 71), we deleted the fitted line.
We changed these contents '*Under light saturation condition, relationship of growth rates of E. huxleyi with phosphate concentrations indicated a very high affinity for dissolved inorganic phosphate (DIP) with 0.04 µmol L$^{-1}$ half-saturation constant for DIP (Fig. 5).*' to 'Furthermore, growth rate of *E. huxleyi* is nearly saturated at 0.25 µmol L$^{-1}$ DIP and is saturated at 0.5 µmol L$^{-1}$ DIP and above. This demonstrated that *E. huxleyi* possesses a high affinity for DIP (Fig. 5) which allowed *E. huxleyi* to take up $PO_4^{3-}$ efficiently.' These changes are in **Lines 646–655** on page 30.

[Figure]

**Figure 5.** Growth rate of *E. huxleyi* as a function of dissolved inorganic phosphate (DIP) concentration. DIN concentration was 100 μmol L$^{-1}$ in all culture media, and DIP concentrations were set up to 0.25 μmol L$^{-1}$, 0.5 μmol L$^{-1}$, 1.5 μmol L$^{-1}$, 3 μmol L$^{-1}$ and 10 μmol L$^{-1}$ in the culture media. All samples were incubated at 200 μmol photons m$^{-2}$ s$^{-1}$ and at 410 μatm $p$CO$_2$ for 4 days, and the values represent the mean ± standard deviation for three replicates. These changes are in **Lines 1150–1156** on page 53.

L480-482: In the natural environment, 10 μM NO3 is definitely not "low nutrients".

Response: We agree with this referee that 10 μM $NO_3^-$ is definitely not "low nutrients".

We changed this text '*In natural waters, E. huxleyi usually starts to bloom following diatom blooms (Tyrrell and Merico, 2004). Therefore, our results also indicate that high growth rate of E. huxleyi at low nutrients concentrations may drive the succession of diatom to E. huxleyi.*' to 'In natural seawaters, *E. huxleyi* usually starts to bloom following diatom blooms (Tyrrell and Merico, 2004), which may be related to high growth rate of *E. huxleyi* at low nutrient concentrations.' These changes are in **Lines 665–669** on page 31.

L483-485: What is "alkaline phosphate activity"? There seems to be a word missing. Also, please explain why this is relevant.
Response: We changed 'alkaline phosphate (APase) activity' to 'alkaline phosphatase (APase) activity'. Alkaline phosphatase enzyme cleaves inorganic P from dissolved external organic sources (Dyhrman and Palenik, 2003). In our study, we did not add organic P into seawater. We have deleted '*, and to decrease alkaline phosphatase (APase) activity*' in **Lines 671–672** on page 31.

Dyhrman, S. T., and Palenik, B.: Characterization of ectoenzyme activity and phosphate-regulated proteins in the coccolithophorid *Emiliania huxleyi*, J Plank. Res., 25, 1215–1225, https://doi.org/10.1093/plankt/fbg086, 2003.

L487-492: I do not understand how this is related to LP conditions. Wouldn't one expect that P limitation would increase energy demand due to upregulated P uptake machinery?

Response: In addition, at LP, cell volume was 17% larger at HC than at LC under the highest light intensity (Table S1 or R3, *see* below). Large cell volume can directly lead to lower growth rates and reduce nutrient uptake by cells which also limit growth of cells. Another possible reason for low tolerance to high-light intensity in growth rate at LP and HC might be a combined effect of LP and HC on the carbon concentrating mechanism (CCM) of *E. huxleyi*. LP or HC is hypothesized to down-regulate the activity of CCM in the green algae *Chlorella emersonii* and in *E. huxleyi*, respectively (Rost and Riebesell, 2004; Beardall et al. 2005). When grown at HC, LP may minimize the activity of CCM of *E. hulxeyi*, which could lead to less energy cost for maintaining high efficient CCM. The saved energy in the HC- and LP-grown cells might have exacerbated photo-inhibition. In summary, high phosphorous requirement, large cell volume and less energy consumption at LP and HC conditions may lead to increased high-light inhibition of growth rates of *E. huxleyi* (Fig. 1). These contents are changed in **Lines 679–692** on pages 31 and 32.

In this study, low light intensity not only limited cell growth but also was suggested to limit phosphate uptake rates (Nalewajko and Lee, 1983). In this case, compared to HNHP condition, growth rates of *E. huxleyi* at LP condition were more likely to be limited by low-light intensity (Fig. 1a,c). High light intensity provided energy for cells to take up P, and cells at LP condition need to consume more energy to up-regulate P uptake (Nalewajko and Lee, 1983) which may lead to decreased high-light inhibition of growth rate at LP than at HNHP condition under LC. These changes are in **Lines 639–646** on pages 29 and 30.

Beardall, J., Roberts, S., Raven, J. A. : Regulation of inorganic carbon acquisition by phosphorus limitation in the green alga *Chlorella emersonii*, Can. J. Bot., 83, 859–864, https://doi.org/10.1139/b05-070, 2005.

Nalewajko, C., Lee, K. : Light stimulation of phosphate uptake in marine phytoplankton, Mar. Bio., 74, 9–15, https://doi.org./10.1007/BF00394269, 1983.

Rost, B., and Riebesell, U.: Coccolithophores and the biological pump: responses to environmental changes, in: Coccolithophores – From Molecular Biology to Global Impact, edited by: Thierstein, H. R. and Young, J. R., Springer, Berlin, 99–125, https://doi.org/10.1007/978-3-662-06278-4_52004, 2004.

L498-499: This can be solely explained by increasing levels of energy saturation of C acquisition and fixation with increasing light.

Response: Kottmeier et al., (2016) provided a nice explanation for increased carbon acquisition and fixation with increasing light.

At LC, *E. huxleyi* mainly uses external $HCO_3^-$ as an inorganic carbon source to synthesize POC

and PIC and increasing light intensity increases the $HCO_3^-$ uptake rate (Kottmeier et al., 2016)

which results in large POC and PIC production rates at high light intensity (Fig. 3). However, at

HC, expression of gene related to the $HCO_3^-$ transporter was down-regulated and the $HCO_3^-$

uptake rate was reduced (Rokitta et al., 2012; Kottmeier et al. 2016), which lead to lower PIC production rates at HC than at LC. Meanwhile, cells at HC can increase $CO_2$ uptake to compensate for low $HCO_3^-$-uptake for photosynthetic C fixation (Kottmeier et al., 2016), which explains the similar POC quotas between HC and LC (Fig. S3). These changes are in **Lines 702–711** on pages 32 and 33.

L506-509: Seems completely unrelated to the presented and discussed data.
Response: We deleted these contents in **Lines 712–714** on page 33: '*LN down regulates expression of the rbcL gene coding for the large subunit of the ribulose-1,5-biphosphate carboxylase/oxygenase (RUBISCO) (Bruhn et al., 2010; Rokitta et al., 2014).*'

L509-511: Seems completely unrelated to previous discussion.
Response: We changed these contents in **Lines 714–724** on page 33: '*To conserve nitrogen, cells at LN prefer to shut down the synthesis of RUBISCO and then reduce carbon fixation (Falkowski et al., 1989) (Fig. 2b)*' to 'LN was found to reduce the enzymatic function and cellular metabolic rates, such as reduced synthesis and activity of ribulose-1,5-biphosphate carboxylase/oxygenase (RUBISCO), which decreases POC quota at both LC and HC (Falkowski et al., 1989; Rokitta et al., 2014) (Fig. S3 and S6). Furthermore, in comparison to LC, lower cell division rates at HC further reduce POC production rates at LN. On the other hand, large cell volume at LP and HC condition was responsible for low cell division rate and low POC production rate (Figs 1, 3 and S3).'

L515-527: Here, results from really nutrient-limited cultures are compared to the data from this study without discussing the lack of considerable nutrient-limitation of growth. Please rewrite this section by taking this into consideration. Also, take into account that under intermediate light levels, growth rates under P limitation and LC are as high as in the full media.
Response: We agree with this referee and rewrite this paragraph.
This text '*Müller et al. (2008) found that calcification (PIC production) occurred only in the G1 cell cycle phase, and that LN or LP held cells in the G1 phase longer, which led to larger PIC quotas and calcification rates at LN or at LP than at HNHP (Figs. 2 and S5). LC and LP treatment decreased cell division rates, elongated cell cycle, and increased coccolith production of E. huxleyi in the darkness (Paasche and Brubak, 1994). In the present work, however, we found slightly faster cell division (growth) and identical calcification rates at LP and high light intensities (Figs. 1c, 2f and S5). LP has been shown to up-regulate the genes involved in calcium binding proteins such as the glutamic acid related to synthesize of coccolith, calcium homeostasis and transcription factor (cmyb) (Wahlund et al., 2004; Dyhrman et al., 2006), and facilitates the formation of cytoplasmic membrane bodies (Shemi et al., 2016). These are related to the pathways associated with production of coccoliths (Young and Henriksen, 2003) and may also be responsible for larger PIC quotas at LP.*' were replaced by 'Nimer and Merrett (1993) reported that decreased DIN concentration facilitates calcification rate of E. huxleyi. This is consistent with our result. Due to lower photosynthetic carbon fixation rate and larger calcification rate at LN in comparison to HNHP (Fig. 3), we could expect that at LN, a high proportion of intracellular $HCO_3^-$ or $CO_2$ was reallocated to synthesize particulate inorganic carbon. On the other hand, at

LP, slightly larger PIC production rate is likely due to larger cell volume in comparison to HNHP (Fig. 3).' These changes are in **Lines 728–747** on pages 33 and 34.

In addition, we provided three reasons for similar growth rates between LP and HNHP at LC and intermediate light levels. These contents were shown in **lines 639–662** on pages 29 and 30: 'In this study, low light intensity not only limited cell growth but also was suggested to limit phosphate uptake rates (Nalewajko and Lee, 1983). In this case, compared to HNHP condition, growth rates of *E. huxleyi* at LP condition were more likely to be limited by low-light intensity (Fig. 1a,c). High light intensity provided energy for cells to take up P, and cells at LP condition need to consume more energy to up-regulate P uptake (Nalewajko and Lee, 1983) which may lead to decreased high-light inhibition of growth rate at LP than at HNHP condition under LC. Furthermore, growth rate of *E. huxleyi* was nearly saturated at 0.25 μmol L$^{-1}$ DIP and was saturated at 0.5 μmol L$^{-1}$ DIP and above. This demonstrated that *E. huxleyi* possesses a high affinity for DIP (Fig. 5) which allowed *E. huxleyi* to take up $PO_4^{3-}$ efficiently. Rokitta et al. (2016) showed that even though

$PO_4^{3-}$ concentration in the culture media declined to zero (undetectable), cell number sustained an increase for 4 days, which indicates that *E. huxleyi* cells could store phosphorus for later use. Consequently, high energy consumption mechanism, efficient uptake and storage capacity for

$PO_4^{3-}$ in *E. huxleyi* could account for no significant differences in growth rates between LP and

HNHP at LC and high light intensities.'

Nalewajko, C., Lee, K. : Light stimulation of phosphate uptake in marine phytoplankton, Mar. Bio., 74, 9–15, https://doi.org./10.1007/BF00394269, 1983.
Rokitta, S. D., von Dassow, P., Rost, B., and John, U.: P- and N-depletion trigger similar cellular responses to promote senescence in eukaryotic phytoplankton, Front. Mar. Sci., 3, 109, https://doi.org/10.3389/fmars.2016.00109, 2016.

L535-536: ETRmax were measured at high light, so it cannot be limited by low energy input. Instead, previous acclimation to low light may have hampered usage of the provided energy.
Response: We have deleted these contents in **lines 755–760** on page 35: '*At low light intensities, the ETR$_{max}$ values were severely limited by low energy input. Supraoptimal light intensities have been found to significantly reduce the abundance of several proteins involved in repair and assembly of PSII, such as repair of photodamaged Psb D1 proteins in the reaction center of PSII of E. huxleyi (McKew et al., 2013). These suggest that high light intensity is likely to do great damage to the PSII structure and then reduce the ETR$_{max}$.*'

L541: Please clarify that you have no data on CCM down-regulation but that this is speculation based on previous publications
Response: We have deleted this text in **lines 760–762** on page 35: '*Especially at HC, supraoptimal light intensity and saved energy from down-regulation of CCM activity synergistically decreased ETR$_{max}$ (Fig. 3).*'

L547-550: Of course these processes are correlated. Can you provide something new that further elucidates this fact?

Response: We have deleted these contents in **lines 763–770** on page 35: '*A previous study found that calcification can be an additional sink for electrons in E. huxleyi (Xu and Gao 2012). Compared with HNHP, larger $ETR_{max}$ at LN or at LP and at saturating light intensities likely resulted from larger calcification rates (Figs. 2 and 3). On the other hand, growth, photosynthetic carbon fixation and nitrogen uptake need energy originating from electron transport (Zhang et al., 2015). At LP and at limiting levels of light intensity, lower growth, photosynthetic carbon and nitrate assimilation rates coincided with lower $ETR_{max}$ (Figs. 1–3), implying correlations of these physiological processes.*'

L555-558: I do not understand this line of thought. Please explain in more detail.

Response: Calcite process within vesicle is shown in equation 1. To calcify, *E. hulxeyi* cells need to take up $HCO_3^-$ and $Ca^{2+}$ from the seawater, which consumes energy. Besides that, they also need to extrude $H^+$ generated during calcification into the cytosol to favour the conversion of $HCO_3^-$ to $CO_3^-$, which also needs some energy. Thus, calcification is a high-energy consumption process, and *E. huxleyi* needs to possess higher light-use efficiencies for their calcification.

$$HCO_3^- + Ca^{2+} \rightarrow CaCO_3 + H^+ \qquad \text{equation 1.}$$

The text '*Calcification is an energy-dependent process (Riebesell and Tortell, 2011), and increased calcification rates at low nutrient concentrations could be aided by higher light-use efficiencies (Fig. 4). In addition, besides taking up inorganic carbon sources and $Ca^{2+}$ from the seawater to calcify, cells need extra energy to expel $H^+$ generated during calcification from the cells (Jin et al., 2017), these may also account for higher light-use efficiencies for PIC production rates.*' was replaced by 'To calcify, *E. huxleyi* cells need to take up $HCO_3^-$ and $Ca^{2+}$ from the seawater, which consumes energy. Besides that, they also need to extrude $H^+$ generated during calcification into the cytosol to favour the conversion of $HCO_3^-$ to $CO_3^-$, which also needs some energy (Paasche 2002). Thus, calcification is an energy consuming process. To maintain large calcification rate at low nutrient concentration, cells possess high light-use efficiencies and can then obtain more energy to take up $HCO_3^-$ and $Ca^{2+}$, and extrude $H^+$ into the cytosol.' These changes are in **Lines 773–785** on pages 35 and 36.

Paasche, E. : A review of the coccolithophorid Emiliania huxleyi (Prymnesiophyceae) with particular reference to growth, coccolith formation, and calcification-photosynthesis interactions, Phycologia, 40, 503–529, 2002.

L563-566: The authors correctly state that highly labour-intensive experiments like the current one are necessary because interactions between multiple stressors cannot be inferred from isolated effects. I therefore do not understand why they speculate on an interaction they did not investigate.

Response: Thanks for the comments of this referee. We have deleted these contents in **Lines 804–806** on page 37: '*In comparison to the current ocean environment, under HC and HL conditions as expected in future oceans, effects of LN and LP on carbon fixation of E. huxleyi may partly negate each other (Fig.2, Table 3).*'

Figure 5 legend: The method description should move into the method section and be more detailed, e.g. were growth rates integrated over 4 days? Were the cultures preacclimated to the conditions? If not, which conditions were they acclimate to before?

Response: We agree with this referee and moved method description in figure 5 legend to the materials and methods section in **Lines 270–280** on page 13.

We added these contents:

**'2.6 Response of growth rate of *E. huxleyi* to different dissolved inorganic phosphate (DIP) concentrations**

L Aquil media were enriched with 100 μmol $L^{-1}$ DIN, aerated for 24 h at 20 $^{o}$C with air containing 400 μatm $p$CO$_2$, sterilized by filtration (0.22 μm pore size, Polycap 75 AS, Whatman) and then pumped into autoclaved 250 mL PC bottles. 10 μmol $L^{-1}$, 3 μmol $L^{-1}$, 1.5 μmol $L^{-1}$, 0.5 μmol $L^{-1}$, 0.25 μmol $L^{-1}$ DIP (finial concentration) were respectively added into Aquil media with three replicates at each DIP concentration. 200 cells $mL^{-1}$ was inoculated to Aquil media and all samples were cultured at 200 μmol photons $m^{-2}$ $s^{-1}$ for 4 days before starting the experiment. Finial cell concentration was measured by using a Z2 Coulter Particle Count and Size Analyzer (Beckman Coulter).'

Technical corrections:

Generally, there are a lot of instances where grammar and wording need to be improved. I strongly suggest the native speakers in the author list to thoroughly correct the final revised version of this manuscript. Below a few examples:

Response: The native speakers in the author list have corrected the grammar and wording in the final revised manuscript.

L34-35: Please correct/rephrase this sentence.

Response: This sentence '*High light intensities compensated for inhibition of LP on growth rates at LC, but exacerbated inhibition of LP at HC.*' was replaced by 'At high light intensity, LP did not limit growth rate at LC, but led to increased high-light inhibition of growth rate at HC.' These changes are in **Lines 36–39** on page 2.

L48-49: Please correct/rephrase this sentence.

Response: Agreed. This sentence: '*Anthropogenic emission of CO$_2$ is taken up by the oceans, decreasing pH of seawater and resulting in ocean acidification (OA)*' was replaced by 'Rising atmospheric CO$_2$ level leads to increasing seawater CO$_2$ concentration and decreasing pH, which is known as ocean acidification (OA).' These changes are in **Lines 54–57** on page 3.

L55: Replace "in the UML" by "therein"
Response: In **Line 62** on page 3: 'in the UML' was replaced by 'therein'.

L57-60: Please correct/rephrase this sentence. Why "counteract"?
Response: We have deleted '*which counteracts with photosynthetic $CO_2$ fixation,*' in **Lines 68–69** on page 4.

L84: Consider replacing "decreased" by "suboptimal"
Response: 'decreased' was replaced by 'suboptimal' in **Line 98** on page 5.

L86-92: Combine first two sentences into one.
Response: This text '*Nevertheless, low nutrient concentrations often enhance the PIC quotas of E. huxleyi. This is due to the fact that low nutrient concentrations hold the cells in the G1 cell cycle phase where calcification occurs (Müller et al., 2008). A recent proteome study on E. huxleyi also shows that nutrient limitation arrests cell cycling (McKew et al., 2015). At molecular levels, nitrate or phosphate limitations down-regulate expression of genes involved in cell cycling, RNA and protein synthesis in E. huxleyi (Rokitta et al., 2014, 2016).*' were replaced by 'Nevertheless, low nutrient concentrations often enhance the PIC quotas of *E. huxleyi,* because low nutrient concentrations arrest cell cycling and lengthen the G1 phase where calcification occurs (Müller et al., 2008; McKew et al., 2015).' These changes are in **Lines 100–108** on page 5.

Müller, M. N., Antia, A. N., and LaRoche, J.: Influence of cell cycle phase on calcification in the coccolithophore *Emiliania huxleyi*, Limnol. Oceanogr., 53, 506–512, https://doi.org/10.4319/lo.2008.53.2.0506, 2008.
McKew, B. A., Metodieva, G., Raines, C. A., Metodier, M. V., and Geider, R. J.: Acclimation of *Emiliania huxleyi* (1516) to nutrient limitation involves precise modification of the proteome to scavenge alternative sources of N and P, Environ. Microbiol., 17, 4050–4062, https://doi.org/10.1111/1462-2920.12957, 2015.

L93-101: Indicate at which pCO2 levels these studies were conducted.
Response: Zhang et al. (2015) reported that at 50–800 μmol photons $m^{-2}$ $s^{-1}$, 1050 μatm $CO_2$ decreased the maximum growth rate, POC and PIC production rate of *Gephyrocapsa oceanica* compared to 510 μatm. These changes are in **Lines 111–113** on pages 5 and 6.
Under natural solar radiation, Jin et al. (2017) reported that compared to 395 μatm, 1000 μatm $CO_2$ increased the growth and POC production rates of *E. huxleyi* at high sunlight levels. These changes are in **Lines 116–118** on page 6.

L119: Why "Even"?
Response: '*Even*' was replaced by 'And' in **Line 138** on page 7.

L398-403: This is not discussion later on. Is it needed then?
Response: Contents in **Lines 771–785** on pages 35 and 36 explained why light-use efficiency of POC and PIC production rates was larger than that of growth rates, which is relevant with **Lines 533–536** on page 25.

L142-145: Please correct/rephrase this sentence.

Response: This sentence in **Lines 162–169** on page 8: '*The synthetic seawater medium Aquil was prepared according to Sunda et al. (2005), added by 2200 μmol L$^{-1}$ bicarbonate (as opposed to 2380 μmol L$^{-1}$ in the original recipe), in order to reflect the alkalinity in the South and East China Seas of about 2200 μmol L$^{-1}$ (Chou et al., 2005; Qu et al., 2017).*' was replaced by 'The Aquil medium was prepared according to Sunda et al. (2005) with the addition of 2200 μmol L$^{-1}$ bicarbonate, resulting in initial concentrations of 2200 μmol L$^{-1}$ total alkalinity (TA). This reflects 2200 μmol L$^{-1}$ alkalinity in the South and East China Seas (Chou et al., 2005; Qu et al., 2017).'

L158: For clarity, please add "For each nutrient treatment, [: : :]"

Response: We added 'For each nutrient treatment, ' in **Line 184** on page 9.

L158-159: Add standard errors for light levels.

Response: For each nutrient treatment, 20 bottles at each $p$CO$_2$ level were incubated at light intensities of 80±5, 120±8, 200±17, 320±16, and 480±30 μmol photons m$^{-2}$ s$^{-1}$ of photosynthetically active radiation (PAR) (4 replicates each) measured using a PAR Detector (PMA 2132, Solar Light Company, Glenside). These changes are in **Lines 184–187** on page 9.

L165-167: Please correct/rephrase this sentence.

Response: This sentence: '*Bottles were rotated two times per day at 10:00 a.m. and 6:00 p.m. to make the cells can obtain light homogeniously.*' was replaced by 'Culture bottles were rotated twice at 10:00 a.m. and 6:00 p.m..' in **Lines 192–195** on page 9.

L178: "CO2 System" should read "CO2SYS"

Response: '*CO2 System*' was replaced by 'CO2SYS' in **Line 211** on page 10.

L182: "Dickson et al. 2003" should read "Dickson et al (2003)".

Response: '*Dickson et al. 2003*' was replaced by 'Dickson et al. (2003)' in **Line 216** on page 10.

L185: "equimolal" should read "equimolar".

Response: '*equimolal*' was replaced by 'equimolar' in **Line 219** on page 10.

L186-187: I assume you did not calculate K1 and K2, but used these constants from Roy et al. for your calculations: : : If so, please correct accordingly.

Response: 'Carbonic acid constants K$_1$ and K$_2$ were taken from Roy et al. (1993).' This change is in **Lines 221** on page 10.

L194-196: Please correct/rephrase this sentence.

Response: 3 mL samples were kept in the dark for 15 min at 20 $^{o}$C, and $F_v$ / $F_m$ values were determined at a measuring light intensity of 0.3 μmol photons m$^{-2}$ s$^{-1}$ and at a saturation pulse of 5000 μmol photons m$^{-2}$ s$^{-1}$ with 0.8 s.' These changes are in **Lines 228–231** on page 11.

L226: Replace "their" by "cellular".

Response: '*their*' was replaced by 'cellular' in **Line 266** on page 13.

L256-264: Estimates of uncertainty are missing.

Response: Table 1 in the original manuscript was replaced by Table S2 in the original supplement in the main text. The text '*The carbonate system parameters (mean values for the beginning and end of incubations) are shown in Table 1. For low $CO_2$ (LC) condition, the $pCO_2$ levels of the media were about 435 µatm at HNHP, 410 µatm at LN and 370 µatm at LP conditions, and the $pH_T$ values (reported on the total scale) were about 8.10 at HNHP, 8.11 at LN and 8.16 at LP. For high $CO_2$ (HC) condition, the $pCO_2$ levels of the media were about 970 µatm at HNHP, 935 µatm at LN and 850 µatm at LP, and the $pH_T$ values were about 7.80 at HNHP, 7.80 at LN, and 7.85 at LP conditions.*' was replaced by 'The carbonate system parameters of the seawater at the beginning and end of the incubation are shown in Table 1. Within the low $CO_2$ (LC) treatment, $pCO_2$ levels of the seawater declined by 16% at HNHP, 19% at LN and 8% at LP, and pH values increased by 0.07 at HNHP, 0.06 at LN and 0.02 at LP (Tukey HSD, all $p < 0.05$). Within the high $CO_2$ (HC) treatment, $pCO_2$ levels of the seawater declined by 23% at HNHP, 21% at LN and 32% at LP, and pH values increased by 0.1 at HNHP, 0.09 at LN and 0.15 at LP (Tukey HSD, all $p < 0.05$).' These changes are in **Lines 315–328** on page 15.

**Table 1** (S2 in the original supplement)**.** Carbonate chemistry parameters of the seawater at the beginning and end of incubations at different nutrient conditions and $pCO_2$ levels.

| | | | $pCO_2$ (µatm) | pH (total scale) | TA (µmol $L^{-1}$) | DIC (µmol $L^{-1}$) | $HCO_3^-$ (µmol $L^{-1}$) | $CO_3^{2-}$ (µmol $L^{-1}$) | $CO_2$ (µmol $L^{-1}$) | $\Omega$ calcite |
|---|---|---|---|---|---|---|---|---|---|---|
| HNHP | LC | Before | 510±17[a] | 8.04±0.01[a] | 2228±17[a] | 2004±20[a] | 1829±21[a] | 159±2[a] | 16±1[a] | 3.8±0.1[a] |
| | | End | 428±57[b] | 8.11±0.05[b] | 2225±24[a] | 1967±22[b] | 1773±34[b] | 180±18[a] | 14±2[b] | 4.3±0.5[a] |
| | HC | Before | 1210±53[a] | 7.71±0.02[a] | 2219±19[a] | 2131±22[a] | 2010±22[a] | 81±2[a] | 39±2[a] | 1.9±0.1[a] |
| | | End | 935±139[b] | 7.81±0.06[b] | 2225±24[a] | 2098±12[b] | 1966±17[b] | 102±14[b] | 30±4[b] | 2.4±0.3[b] |
| LN | LC | Before | 483±23[a] | 8.06±0.02[a] | 2204±10[a] | 1973±10[a] | 1796±13[a] | 162±6[a] | 16±1[a] | 3.9±0.1[a] |
| | | End | 391±39[b] | 8.12±0.03[b] | 2123±38[b] | 1866±45[b] | 1679±48[b] | 175±9[b] | 13±1[b] | 4.2±0.2[b] |
| | HC | Before | 1126±66[a] | 7.73±0.02[a] | 2201±3[a] | 2105±7[a] | 1983±9[a] | 85±4[a] | 36±2[a] | 2.02±0.1[a] |
| | | End | 888±114[b] | 7.82±0.05[b] | 2142±38[b] | 2016±47[b] | 1890±49[b] | 98±8[b] | 29±4[b] | 2.4±0.2[b] |
| LP | LC | Before | 397±16[a] | 8.14±0.02[a] | 2248±30[a] | 1982±22[a] | 1777±17[a] | 192±8[a] | 13±1[a] | 4.6±0.2[a] |
| | | End | 365±24[b] | 8.16±0.02[a] | 2219±20[b] | 1942±22[b] | 1731±25[b] | 199±8[a] | 12±1[b] | 4.8±0.2[a] |
| | HC | Before | 1140±110[a] | 7.73±0.04[a] | 2215±41[a] | 2128±46[a] | 2005±46[a] | 86±7[a] | 37±4[a] | 2.1±0.2[a] |
| | | End | 780±43[b] | 7.88±0.02[b] | 2228±14[a] | 2084±11[b] | 1941±12[b] | 117±6[b] | 25±1[b] | 2.8±0.1[b] |

HNHP, 101 µmol $L^{-1}$ dissolved inorganic nitrogen (DIN) and 10.5 µmol $L^{-1}$ dissolved inorganic phosphate (DIP); LN, 8.8 µmol $L^{-1}$ DIN; LP, 0.4 µmol $L^{-1}$ DIP. Different letters represent statistically differences between the beginning and end of the experiments (Tukey Post hoc, $p <$ 0.05). The values are expressed as mean values with standard deviation for four replicates. These changes are in Lines **1171–1181 on pages 54 and 55.**

L274-279: Units of the treatments are missing.

Response: At LC, growth rate at LN was similar with that at HNHP under limited light intensity with 80 μmol photons m$^{-2}$ s$^{-1}$ (Tukey HSD, df = 1, $p$ = 0.82), and was significantly lower than at HNHP under optimal and supra-optimal light intensities (Tukey HSD, both df = 1, $p$ < 0.01 for 200 μmol photons m$^{-2}$ s$^{-1}$; $p$ = 0.005 for 480 μmol photons m$^{-2}$ s$^{-1}$). At HC, growth rates at LN were significantly lower than those at HNHP under limited, optimal and supra-optimal light intensities (Tukey HSD, all df = 1, $p$ < 0.01 for 80, 200, 480 μmol photons m$^{-2}$ s$^{-1}$). These changes are in **Lines 339–345** on page 16.

L346-349: Replace "At each nutrient condition, at both LC and at HC" by "At all nutrient and CO2 levels".

Response: '*At each nutrient condition, at both LC and at HC*' was replaced by 'At all nutrient and CO$_2$ levels,' in **Lines 414** on page 19.

L356: Why "both"?

Response: 'both' indicates 'at both LC and HC'. At both LC and HC, at 80–480 μmol photons m$^{-2}$ s$^{-1}$ $F_{\sqrt{}}/F_m$ did not show significant differences between LN and HNHP (Tukey HSD, all df = 1, all $p$ > 0.05), and at 480 μmol photons m$^{-2}$ s$^{-1}$, they were lower at LP than at HNHP at both LC and HC (Tukey HSD, both df = 1, both $p$ < 0.05) (Fig. 2a,c). These changes are in **Lines 422–425** on page 20.

L439-441: This sentence sounds as if the authors would have observed the first statement, and the reference refers to the latter, while the opposite is true. Please rephrase.

Response: This sentence: '*At HC, the negative effect of high [H$^+$] on growth rate was larger than positive effects of increased CO$_2$ and* $HCO_3^-$ *concentrations, which could be attributed to lower growth rates at HC than at LC (Fig. 1) (Bach et al., 2011).*' was replaced by 'Lower growth rates at HC than at LC are due to the fact that at HC the negative effect of high [H$^+$] on growth rate was larger than positive effects of increased CO$_2$ and $HCO_3^-$ concentrations (Bach et al., 2011).' These changes are in **Lines 607–612** on page 28.

L465: "saturation condition, relationship" should read "saturated conditions, the relationship".

Response: We have deleted this sentence: '*Under light saturation condition, relationship of growth rates of E. huxleyi with phosphate concentrations indicated a very high affinity for dissolved inorganic phosphate (DIP) with 0.04 μmol L$^{-1}$ half-saturation constant for DIP.*' in **Lines 646–649** on page 30.

L467-471: Please correct/rephrase this sentence.

Response: these contents '*Under light saturation condition, relationship of growth rates of E. huxleyi with phosphate concentrations indicated a very high affinity for dissolved inorganic phosphate (DIP) with 0.04 μmol L$^{-1}$ half-saturation constant for DIP (Fig. 5). Since LP was reported to enhance expression of gene with a role in phosphorus assimilation or metabolism and synthesis of inorganic $PO_4^{3-}$ transporters (Dyhrman et al., 2006; McKew et al., 2015; Rokitta et al., 2016), which allowed E. huxleyi to take up $PO_4^{3-}$ efficiently enough, so that LP did not result in reduced growth rate at LC in this study (Fig. 1).*' was replaced by 'Furthermore, growth rate of E. huxleyi was nearly saturated at 0.25 μmol L$^{-1}$ DIP and was saturated at 0.5 DIP and above. This demonstrated that E. huxleyi possesses a high affinity for DIP (Fig. 4), which allowed E. huxleyi to take up $PO_4^{3-}$ efficiently.' These changes are in **Lines 646–655** on page 30.

L480-482: Please correct/rephrase this sentence.

Response: In natural seawaters, E. huxleyi usually starts to bloom following diatom blooms (Tyrrell and Merico, 2004) which may be related to a high growth rate of E. huxleyi at low nutrient concentrations.' These changes are in **Lines 665–669** on page 31.

L496-502: Please correct/rephrase this sentence.

Response: This text '*At LC, E. huxleyi mainly uses external $HCO_3^-$ as an inorganic carbon source for photosynthesis and calcification, and increasing light intensities are able to increase $HCO_3^-$ uptake rates (Kottmeier et al., 2016). This may explain why POC and PIC quotas and production rates increased with increasing light intensity (Figs. 2 and S5). HC down-regulates gene expression related to the $HCO_3^-$ transporter (Rokitta et al., 2012) and decreases the $HCO_3^-$ uptake rate in E. huxleyi (Kottmeier et al. 2016), leading to lower PIC quotas at HC than at LC (Fig. 2).*' were replaced by 'At LC, E. huxleyi mainly uses external $HCO_3^-$ as an inorganic carbon source to synthesize POC and PIC and increasing light intensity increases the $HCO_3^-$ uptake rate (Kottmeier et al., 2016), which results in large POC and PIC production rates at high light intensity (Fig. 3). However, at HC, expression of gene related to the $HCO_3^-$ transporter was down-regulated, and the $HCO_3^-$ uptake rate was reduced (Rokitta et al., 2012; Kottmeier et al. 2016), which lead to lower PIC production rates at HC than at LC.' These changes are in **Lines 696–708** on pages 32 and 33.

L503: omit first "-"

Response: '*low-$HCO_3^-$-uptake*' was replaced by 'low $HCO_3^-$-uptake' in **Line 709** on page 33.

L516: Insert "could have" between "which" and "led".

Response: We have deleted this content '*which led to*' in **Lines 729** on page 33.

L553-555: I do not understand this sentence. Please rephrase.

Response: This sentence '*Calcification is an energy-dependent process (Riebesell and Tortell, 2011), and increased calcification rates at low nutrient concentrations could be aided by higher light-use efficiencies (Fig. 4). In addition, besides taking up inorganic carbon sources and Ca$^{2+}$ from the seawater to calcify, cells need extra energy to expel H$^{+}$ generated during calcification from the cells (Jin et al., 2017), these may also account for higher light-use efficiencies for PIC production rates.*' was replaced by 'To calcify, *E. hulxeyi* cells need to take up $HCO_3^-$ and Ca$^{2+}$ from the seawater, which consumes energy. Besides that, they also need to extrude H$^{+}$ generated during calcification into the cytosol to favour the conversion of $HCO_3^-$ to $CO_3^-$, which also consumes energy (Paasche 2002). Thus, calcification is an energy comsuming process. To maintain large calcification rates at low nutrient concentration, cells possess high light-use efficiencies and can then obtain more energy to take up $HCO_3^-$ and Ca$^{2+}$ and extrude H$^{+}$ into the cytosol.' These changes are in **Lines 773–785** on pages 35 and 36.

Figures: Consider using a dashed line for one of the fits to distinguish between the two CO2 levels.

Response: Thanks for this nice suggestion of this referee. Dashed lines represent the fits at HC in all figures.

L869: Based on which test?

Response: 'Different letters showed statistical differences based on the Tukey post hoc test.' These changes are in **Lines 1139–1140** on page 52.

L902: Explain letters to abbreviate pCO2 and light intensity.

Response: LC represented 410 μatm $p$CO$_2$, and light intensity was 200 μmol photons m$^{-2}$ s$^{-1}$ in **Line 1154–1156** on page 53.

This sentence '*All samples were incubated at 200 μmol photons m$^{-2}$ s$^{-1}$ and at LC for 4 days.*' was replaced by 'All samples were incubated at 200 μmol photons m$^{-2}$ s$^{-1}$ and at 410 μatm $p$CO$_2$ for 4 days, and the values represent the mean ± standard deviation for three replicates.' These changes are in **Line 1154–1156** on page 53.

L920-921: Please rephrase to make it a sentence.

Response: These contents '*All samples were incubated at 200 μmol photons m$^{-2}$ s$^{-1}$ and at 410 μatm pCO$_2$ for 4 days. The values represent the mean ± standard deviation for three replicates.*' were replaced by 'All samples were incubated at 200 μmol photons m$^{-2}$ s$^{-1}$ and at 410 μatm $p$CO$_2$ for 4 days, and the values represent the mean ± standard deviation for three replicates.' These changes are in **Line 1154–1156** on page 53.

Figure 2: Indicate if the PIC:POC is molar- or weight-based.
Response: PIC:POC ratio is based on weight in figure 3 in **Line 1284** on page 65.

**Responses to comments of referee 2 are shown as following:**

The study by Zhang et al. is an important effort to address multifactorial control over the response to acidification of an important phytoplankton species, using an ambitious matrix of treatments. However, there are some major problems that must be resolved, as currently I am unsure of a major portion of the data or results interpretations as presented.
Response: We thank this referee for the positive comments. We refocused on growth rate, POC and PIC production rates. The conclusion of this study are that: (1) low dissolved inorganic nitrogen concentration and high $CO_2$ level synergistically reduced growth rate; (2) Effect of low dissolved inorganic phosphate concentration on growth rate was modulated by light intensity and $CO_2$ level; (3) low dissolved inorganic nutrient concentrations (DIN or DIP) and high $CO_2$ level synergistically reduced POC production rate; (4) low dissolved inorganic nutrient concentrations (DIN or DIP) facilitated PIC production rate. These conclusions have been shown as subsections in the discussion section in the revised BG manuscript.

1. In the Introduction the authors plant the study as if it aims to mimic the natural environment presently or in the future in the laboratory, that is, that the nutrient, light, and CO2 conditions they chose are truly representative. I think this is not unnecessary and risks setting up an incorrect context for interpreting the study results. For example, the authors justify the choice of light regimes in the first paragraph by claiming that phytoplankton in the future ocean will be exposed to higher light levels in the mixed layer, citing two studies. I note also that neither of the studies cited (Gao et al. 2012 and Hutchins and Hu 2017) is relevant to cite, as one is a lab study and the other is a review of lab studies, and neither is a model study predicting average light fields at which phytoplankton will be exposed in the future ocean. In any case, it is difficult to imagine that changes in the stratification of the central ocean basins can only lead to an increase in light exposure. Light exposure is highly dynamic and depends on mixing regime, so yes, the light regime should change, but to model that in the lab with constant light levels is not reasonable. My comment here does not at all negate the study design: Even though we can never mimic the ocean in the lab, it still serves to understand how factors may interact. In the case of trying to predict the response to acidification, it at least serves us to understand how robust the lab results might be for predicting the direction of possible responses, and often as well helps provide insight into mechanisms underlying the responses. I do suggest that they consider revising the Intro.
Response: We agree with this referee that it is difficult to imagine that changes in the stratification of the central ocean basins can only lead to an increase in light exposure. However, it is true that light availability is tied to the mixed layer depth and sea ice fraction, and reduced primary production correlates with increased stratification in the tropical, southern Pacific and North Atlantic in the CSM1.4 model (Steinacher et al. 2010). Thus, two references: Gao et al. (2012), and Hutchins and Hu (2017) were replaced by Steinacher et al. (2010) in **line 62** on page 3.

Some contents in the introduction were changed (underlined are altered text).
Rising atmospheric $CO_2$ level leads to increasing seawater $CO_2$ concentration and decreasing pH, which is known as ocean acidification (OA) (Caldeira and Wickett, 2003). On the other hand, rising atmospheric $CO_2$ also leads to global and ocean warming, which enhances water column stratification and shoals the upper mixed layer (UML) (Wang et al., 2015). This affects light exposure of phytoplankton dwelling therein (Steinacher et al. 2010). In addition, enhanced stratification reduces the transport of nutrients from deep oceans to the UML (Behrenfeld et al., 2006), which reduces the nutrient concentrations in the UML. These changes are in **Lines 54–65** on page 3.

Coccolithophores take up $CO_2$ and/or $HCO_3^-$ from seawater for carboxylation, and use $HCO_3^-$ for calcification which produces coccoliths. Calcification processes generate $CO_2$ due to production of protons, and therefore influencing $CO_2$ influx into the oceans (Rost and Riebesell, 2004). Growth rate, particulate organic (POC) and inorganic carbon (PIC) production rates of *Emiliania huxleyi*, the most abundant calcifying coccolithophore species, usually display optimum responses to a broad range of $CO_2$ concentration (Bach et al., 2011). Growth, POC and PIC production rates could increase, decrease and be unaffected by rising $CO_2$ treatments across a narrow $CO_2$ range, which is dependent on the optimal $CO_2$ levels of these physiological rates and the selected $CO_2$ range (Langer et al., 2009; Richier et al., 2011; Bach et al., 2015; Jin et al., 2017). Differences in sampling locations, experimental setups, deviations in the measuring methods and intraspecific differences can generally be responsible for the differential responses of growth, POC and PIC productions to rising $CO_2$ in *E. huxleyi* (Langer et al., 2009; Meyer and Riebesell, 2015). These changes are in **Lines 66–84** on pages 3 and 4.

These contents '*This is due to the fact that low nutrient concentrations hold the cells in the G1 cell cycle phase where calcification occurs (Müller et al., 2008). A recent proteome study on E. huxleyi also shows that nutrient limitation arrests cell cycling (McKew et al., 2015). At molecular levels, nitrate or phosphate limitations down-regulate expression of genes involved in cell cycling, RNA and protein synthesis in E. huxleyi (Rokitta et al., 2014, 2016).*' were replaced by 'because low nutrient concentrations arrest cell cycling and lengthen the G1 phase where calcification occurs (Müller et al., 2008; McKew et al., 2015).' in **Lines 101–108** on page 5.

Recently, several studies investigated interactive effects of rising $CO_2$ and light intensity on physiological rates of coccolithophores (Feng et al., 2008; Jin et al., 2017). Zhang et al. (2015) reported that at 50–800 μmol photons $m^{-2}$ $s^{-1}$, 1050 μatm $CO_2$ decreased the maximum growth rate, POC and PIC production rates of *Gephyrocapsa oceanica* compared to 510 μatm. At low light levels, coccolithophores increase $CO_2$ uptake to compensate for inhibition of $HCO_3^-$ uptake on photosynthesis, while at high light intensity they don't increase $CO_2$ uptake (Kottmeier et al.,

2016). Under natural solar radiation, Jin et al. (2017) reported that compared to 395 µatm, 1000 µatm $CO_2$ increased the growth and POC production rates of *E. huxleyi* at high sunlight levels. These indicate that during growth under different experimental conditions, rising $CO_2$ showed contrasting effects on growth and POC production rates of *E. huxleyi* and *G. oceanica*. These changes are in **Lines 109–122** on pages 5 and 6.

Steinacher, M., Joos, F., Frö licher, T. L., Bopp, L., Cadule, P., Cocco, V., Doney, S. C., Gehlen, M., Lindsay, K., Moore, J. K., Schneider, B., Segschneider, J. : Projected 21st century decrease in marine productivity: a multi-model analysis, Biogeosciences, 7, 979–1005, 2010.

2. There is at least one major problem with the growth rates reported, possibly many more:
a. It makes no sense to report a single growth rate as the response to nutrient-limitation in batch culture experiments. At inoculation of cultures, cells should be nutrient replete even in the LP and LN conditions. If they have been "acclimated" to growing previously in the same media, the inoculums likely are from cultures that have already exhausted the phosphate (in LP) or nitrate (in LN), so the cells will have to re-configure nutrient uptake and metabolism, begin to grow, then exhaust the nutrients, re-configuring nutrient and connected metabolism again. The growth rate most certainty will NOT be constant. A recent study where these effects can be seen would be that of Rokitta et al. (2014). The authors report only a single growth rate, not changes in cell density over time, no indication of when nutrient limitation may start nor how long cells have been in nutrient limited conditions. In this sense, a good study to look at would be the recent one by Müller et al. (2017) using a continuous culture approach to understand the interaction/independence of nutrient limitation and acidification effects (curiously, the authors cite the study in the Intro but do not discuss at all, despite its central relevance!). The results presented in the current manuscript are therefore completely uninterpretable.

Response: Thanks for the comments and suggestions of this referee. To prevent seawater-air $CO_2$ exchange, incubation bottles were filled with seawater with no headspace and tightly closed during incubations. This is one of the reasons for measuring cell concentration at the end of the incubation and reporting a single growth rate. More importantly, studies of Rokitta et al. (2014) reported that cell number of *E. huxleyi* increased exponentially on the third to sixth days during incubation, and showed that growth rates were similar at the fourth, fifth and sixth days. We agree with this referee that growth rates are not constant, however, variation in growth rates at different days were much lower than variation in growth rates between different treatments.

Low DIN and DIP concentration did not limit growth in this study, the reasons are that:
In this study, growth rates of *E. huxleyi* were larger than 1 in almost all treatments, and cells divided 1–2 times per day (Fig .1), which indicates non-limiting nutrient conditions during the incubation. Based on measured PON quota and cell concentration in this study (Figs. 1 and S6 in the manuscript), PON concentrations at the end of incubations were estimated to be 7.8–9.3 µmol $L^{-1}$ at different nutrient conditions (Table S2). These data were closely correlated with molar drawdown of dissolved inorganic nitrogen (DIN) during the incubations. Furthermore, residual 1 µmol $L^{-1}$ DIN in the final day of the incubation showed non-limitation of growth and POC production rates by nitrogen. On the other hand, Rokitta et al. (2016) reported that $F_v/F_m$ of *E.*

*huxleyi* was 50% lower at P-depleted than at P-replete conditions. In this study, $F_v/F_m$ and POC quota were very similar between LP and HNHP treatments (Figs. 2 and S3), which suggests that LP did not limit growth and carbon fixation. This text was added in the first paragraph in the discussion section in **Lines 568–579** on pages 26 and 27.

Comparison between the study of Müller et al. (2017) and ours are shown as following:
Using a chemostat culture, Müller et al. (2017) reported that DIN or DIP limitation decreased the POC and PIC production rates (in pg C cell$^{-1}$ d$^{-1}$) by 50% and rising $p$CO$_2$ levels did not affect POC production rates. However, when normalized to cell volume, nutrient limitation did not affect POC and PIC production rates (in pg C cellV$^{-1}$ d$^{-1}$), and rising $p$CO$_2$ levels reduced POC and PIC production rates. In our study, decreased DIN or DIP concentration reduced the normalized POC production rates (in pg C cellV$^{-1}$ d$^{-1}$), and increased the normalized PIC production rates at both LC and HC (Fig. S5). Differing results between the study of Müller et al. (2017) and ours may result from different experimental setup. Growth was really limited by N or P, cells were cultured in a continuous photon flux, and cell growth was in the stable phase when POC and PIC samples were taken in the study of Müller et al. (2017). While we took POC and PIC samples in the exponential growth phase, and LN or LP did not really limit growth of *E. huxleyi* in our study. These contents were added in the discussion section in **Lines 786–798** on pages 36 and 37.

Rokitta, S. D., von Dassow, P., Rost, B., and John, U.: *Emiliania huxleyi* endures N-limitation with an efficient metabolic budgeting and effective ATP synthesis, BMC Genomics, 15, 1051–1064, https://doi.org/10.1186/1471-2164-15-1051, 2014.
Müller, M. N., Trull, T. W., and Hallegraeff, G. M.: Independence of nutrient limitation and carbon dioxide impacts on the Southern Ocean coccolithophore *Emiliania huxleyi*, ISME J., 11, 1777–1787, https://doi.org/10.1038/ismej.2017.53, 2017.

[Figure]

**Figure S5.** At both LC and HC, normalized POC production rate (pg C cellV$^{-1}$ d$^{-1}$) of *E. huxleyi* as a function of light intensity at HNHP (**a**), LN (**b**) and LP (**c**) conditions. At both LC and HC, light response of normalized PIC production rate (pg C cellV$^{-1}$ d$^{-1}$) of *E. huxleyi* at HNHP (**d**), LN (**e**) and LP (**f**) conditions. The values represent the mean ± standard deviation for four replicates. These contents were added in the supplement.

b. The growth rate presented appears to be calculated only from an initial cell concentration and a final one, which is generally not adequate even in batch culture experiments when nutrient limitation is avoided, because it is necessary to understand if growth rate changes or not during the experiment

Response: When cell growth is in the exponential phase, cell concentration increased exponentially with incubation days, and growth rates should be very similar.

Langer et al. (2013) found that growth of cells on the fourth to sixth days of batch cultures was in the exponential phase even at 3 µmol L$^{-1}$ $NO_3^-$ or at 0.29 µmol L$^{-1}$ $PO_4^{3-}$ with the same *E. huxleyi* strain. In this study, all parameters were measured on the fourth to the sixth days, so it is most likely that cells in all treatments were sampled in the exponential growth phase. These contents are shown in **Lines 197–202** on pages 9 and 10.

Langer, G., Oetjen, K., and Brenneis, T.: Coccolithophores do not increase particulate carbon production under nutrient limitation: A case study using *Emiliania huxleyi* (PML92/11), J. Exp. Mar. Biol. Ecol., 443, 155−161, 2013

c. The initial cell concentration appears not to have been measured, but to have been calculated, which causes many errors.

Response: The bottles were filled with Aquil with no headspace to minimize gas exchange. The volume of the inoculum was calculated (see below) and the same volume of Aquil was taken out from 500 mL bottles before inoculation. These changes are in **Lines 179–182** on page 9.

There was 625 ml seawater in the 500 ml polycarbonate (PC) bottles. Before cells were inoculated to new seawater, finial cell concentrations ($C_0$) were measured. Then we calculated the inoculated volumes (V) according to V = (200 cell/ml x 625 ml)/$C_0$. And we don't think this method cause errors.

d. The growth rates provided seem high in comparison to most previous studies of this species. Most authors report that the maximum growth rate of Emiliania huxleyi in batch cultures under "optimum" nutrient and light conditions and a day:night lighting is in the range of 0.7-0.9, a little more than one doubling per day (for just a sampling of studies, see van Bleisjwiik et al. 1994; Zondervan et al. 2002; Rokitta et al. 2014; Müller et al. 2015). Higher growth rates are occasionally reported, but under longer light cycles, e.g. Langer et al. 2009, or a very nice study by the same first author (Zhang et al. 2014). The rates here seem quite high for a 12:12 light:dark cycle, and for that reason it's important to see the data (at least in supplementary), to have full confidence in the methods, and to have at least a brief mention of this.

Response: Growth rate of *Emiliania huxleyi* was affected by light intensity, light cycle, temperature and dissolved inorganic nitrogen and phosphate concentrations and so on. I summarized the culture conditions of some studies (Table R2 in the response letter), and found that high incubation temperature (20 $^{\circ}$C) in our study may lead to higher growth rates compared studies of Bleisjwiik et al. (1994); Zondervan et al. (2002); Rokitta et al. (2014) and Müller et al. (2015). Final cell concentration in this study was shown in Table R3 (or Table S1 in the supplement).

van Bleijswijk, J. D. L., Kemper, R. S., Veldhuis, M. J., Westbroek, P. : Cell and growth characteristics of types A and B of *Emiliania huxleyi* (prymnesiophyceae) as determined by flow cytometry and chemical analyses, J. Phycol., 30, 230–241, 1994.
Zondervan, I., Rost, B., Riebesell, U. : Effect of CO2 concentration on the PIC/POC ratio in the coccolithophore *Emiliania huxleyi* grown under light-limiting conditions and different daylengths, J. Exp. Mar. Biol. Ecol., 272, 55–70, 2002.
Müller, M. N., Trull, T. W., and Hallegraeff, G. M.: Differing responses of three Southern Ocean *Emiliania huxleyi* ecotypes to changing seawater carbonate chemistry, Mar. Ecol. Prog. Ser., 531, 81–90, 2015.
Langer, G., Nehrke, G., Probert, I., Ly, J., and Ziveri, P.: Strain-specific responses of *Emiliania huxleyi* to changing seawater carbonate chemistry, Biogeosciences, 6, 2637–2646, 2009.
Zhang, Y., Klapper, R., Lohbeck, K. T., Bach, L. T., Schulz, K. G., Reusch, T. B. H., and Riebesell, U.: Between- and within-population variations in thermal reaction norms of the coccolithophore *Emiliania huxleyi*, Limnol. Oceanogr., 59, 1570–1580, 2014.

Table R2. Growth rates and experimental culture conditions of some studies.

| Reference | Growth rate (d$^{-1}$) | Light intensity (μmol photons m$^{-2}$ s$^{-1}$) | Light cycle (Light/Dark) | Temperature (°C) | DIN concentration (μmol L$^{-1}$) | DIP concentration (μmol L$^{-1}$) |
|---|---|---|---|---|---|---|
| Bleijswijk et al. 1994 | 0.8 | 70 or 140 | 16:8 | 10 or 15 | 30 to 39 | 0.2 to 0.4 |
| Zondervan et al. 2002 | 1.1 | 150 | 16:8 | 15 | 100 | 6.25 |
| Rokitta et al. 2014 | 0.8 | 250 | 16:8 | 15 | 100 | 6.25 |
| Müller et al. 2015 | 0.3–0.6 | 100–115 | 24:0 | 14 | 88 | 3.6 |
| Langer et al. 2009 | 1.2–1.6 | 400 | 16:8 | 17–20 | 100 | 6.25 |
| Zhang et al. 2014 | 1.1–1.6 | 160 | 16:8 | 15–22 | 64 | 4 |
| This study | 1.2–1.3 | 200 | 12:12 | 20 | 100 or 8 | 10 or 0.4 |

Table R3 (S1). Final cell concentration and cell volume at the end of the incubation, and incubation period. Data in the brackets are the standard deviations for four replicates. These contents were added in supplement as Table S1.

| Initial N/P | $p\mathrm{CO_2}$ | L | Final cell concentration (cell mL$^{-1}$) | Incubation time (d) | cell volume (μm$^3$) |
|---|---|---|---|---|---|
| 101/10.5 | 435 | 80 | 153,960(14,490) | 6 | 39.82(1.33) |
| | | 120 | 86,910(11,650) | 5 | 51.67(0.96) |
| | | 200 | 40,060(5,180) | 4 | 62.22(0.97) |
| | | 320 | 35,250(4,280) | 4 | 54.88(1.13) |
| | | 480 | 22,010(2,860) | 4 | 52.47(3.08) |
| | 970 | 80 | 119,180(9,560) | 6 | 46.99(1.49) |
| | | 120 | 76,330(13,560) | 5 | 50.49(0.52) |
| | | 200 | 38,950(1,620) | 4 | 57.36(0.68) |
| | | 320 | 25,050(1,480) | 4 | 51.92(0.78) |
| | | 480 | 20,390(616) | 4 | 50.58(2.34) |
| 8.8/10.5 | 410 | 80 | 131,030(7,160) | 6 | 52.50(0.55) |
| | | 120 | 86,350(3,350) | 5 | 66.66(0.80) |
| | | 200 | 37,630(1,810) | 4 | 65.00(0.31) |
| | | 320 | 125,460(6,320) | 5 | 62.08(1.74) |
| | | 480 | 53,920(4,930) | 5 | 59.94(4.42) |
| | 936 | 80 | 83,060(3,410) | 6 | 51.79(0.27) |
| | | 120 | 50,630(1,520) | 5 | 56.65(0.67) |

| | | 200 | 29,110(1,030) | 4 | 59.27(0.79) |
|---|---|---|---|---|---|
| | | 320 | 86,510(1,680) | 5 | 60.52(1.40) |
| | | 480 | 42,240(11,370) | 5 | 56.16(3.16) |
| 101/0.4 | 372 | 80 | 81,230(11,000) | 6 | 61.75(2.19) |
| | | 120 | 98,630(4,490) | 5 | 59.65(0.91) |
| | | 200 | 51,750(1,920) | 4 | 58.28(0.58) |
| | | 320 | 38,220(3,120) | 4 | 53.70(1.16) |
| | | 480 | 75,040(16,940) | 5 | 60.93(1.83) |
| | 852 | 80 | 67,400(8,450) | 6 | 48.56(3.20) |
| | | 120 | 43,320(2,130) | 5 | 64.40(0.88) |
| | | 200 | 116,630(1,760) | 5 | 61.35(0.81) |
| | | 320 | 90,170(2,960) | 5 | 64.1(0.95) |
| | | 480 | 44,490(2,150) | 6 | 71.66(1.33) |

e. I'm especially concerned in the Methods when they say that 4-6 days corresponds to 14 generations. That would correspond to growth rates between 1.62 day-1 (at a "low" light level previous studies have found to nearly saturate growth rate) or 2.42 day-1, a level unachievable even for most diatoms (and not readily believable for a coccolithophore, even E. huxleyi). Perhaps this is a typographical error?

Response: Cells were cultured at each experimental treatment for 4 to 6 days, which corresponds to 7 to 8 generations, and then inoculated to new seawater and cultured for another 4 to 6 days.

'cells were acclimated to the experimental treatments for at least 7 generations before starting the experiment' in **Lines 189–190** on page 9.

f. For cell counts they use a particle counter (presumably based on the Coulter principle, although the information provided is inadequate to identify the type of instrument). This is potentially very problematic particularly in the case of E. huxleyi. How can living cells be distinguished from detached coccoliths, agglomerations of detached coccoliths, and/or empty coccospheres, all of which are very abundant in E. huxleyi cultures? In limited conditions these other particles can actually dominate the suspended particles found in cultures and it can be difficult to distinguish cells. With all these issues, I really am not sure from the information provided that they are actually measuring cells. Cells should be counted under a microscope or with a flow cytometer (a Coulter-type particle counter can be used, if it is being checked, compared, calibrated with microscope or cytometer counts throughout the experiment). Details are needed.

Response: We thank this referee for their suggestion.

Cell densities were measured using a Z2 Coulter Particle Count and Size Analyzer (Beckman Coulter). The diameter of detected particles was set to 3 to 7 μm in the instrument, which excludes detached coccoliths because the diameter of coccolith is less than 3 μm (Müller et al., 2012). These changes are in **Lines 247–250** on page 12.

Recently, we measured cell concentration using a Cell Lab Quanta SC flow cytometer (Beckman Coulter) and a Z2 Coulter Particle Count and Size Analyzer. Cell concentration was 14,550 cells mL$^{-1}$ when it was measured by a flow cytometer (Fig. R1 and R2 in the response letter) and was 15, 210 cells mL$^{-1}$ when it was measured by the Z2 Coulter Particle Count and Size Analyzer (Fig. R3 in the response letter). Variation in measured cell concentration between two methods was 4.3%. Thus, we don't think that the cell concentration measured using a Z2 Coulter Particle Count and Size Analyzer cause error.

Cell concentration was also measured by a Cell Lab Quanta SC flow cytometer (Beckman Coulter), and variation in measured cell concentration between two methods was about 4.3%. This sentence was added in **Lines 250–253** on page 12.

[Figure]

Figure R1 Signal shown in flow cytometer.

Tube Name: 1zy
Sample ID:

| Population | ents/µL(V) | % T... | % P... | Mean FSC-A | Mean SSC-A | Median FSC-A | Median SSC-A | CV FSC-A | CV SSC- |
|---|---|---|---|---|---|---|---|---|---|
| 0.2um | 130.99 | 19.5... | 19.5... | 4839.3 | 2082.9 | 4702.0 | 2112.8 | 131.43% | 14. |
| 10UM | 0.00 | 0.00% | 0.00% | #### | #### | #### | #### | #### | # |
| 3UM | 0.00 | 0.00% | 0.00% | #### | #### | #### | #### | #### | # |
| 3UM-1 | 0.40 | 0.06% | 0.06% | 802676.6 | 907325.0 | 802676.6 | 907325.0 | 42.89% | 8. |
| 10UM-1 | 0.00 | 0.00% | 0.00% | #### | #### | #### | #### | #### | # |
| 6UM | 0.40 | 0.06% | 0.06% | 806525.1 | 1929917.1 | 806525.1 | 1929917.1 | 75.85% | 13. |
| P4 | 506.61 | 75.5... | 75.5... | 7334.9 | 4553.8 | 6904.6 | 3084.0 | 111.04% | 144. |
| P2 | 14.55 | 2.17% | 2.17% | 650772.1 | 149084.1 | 647158.3 | 123957.0 | 21.58% | 79. |
| P1 | 634.01 | 94.5... | 94.5... | 12763.8 | 19283.6 | 8495.8 | 3651.6 | 234.81% | 263. |

Figure R2 Calculated cell concentration by a flow cytometer.

[Figure]

Figure R3 Cell concentration shown by a Z2 Coulter Particle Count and Size Analyzer

Müller, M. N., Beaufort, L., Bernard, O., Pedrotti, M. L., Talec, A., Sciandra, A. : Influence of $CO_2$ and nitrogen limitation on the coccolith volume of *Emiliania huxleyi* (Haptophyta), Biogeosciences, 9, 4155–4167, 2012.

Because of these unresolved methodological issues in measuring cell abundance, at the present time I cannot trust growth rate data or cell elemental quotas reported.

Response: As mentioned above, cell abundance was measured using a suitable method and growth rate was correctly calculated. Cellular carbon content was measured using a Perkin Elmer Series II CHNS/O Analyzer 2400 instrument (Perkin Elmer Waltham, MA).
Variations in measured carbon content between the four replicates were calculated to be 1–13% in this study. This sentence was added in **Lines 267–268** on page 13.

3. There is no way to know when nutrients became depleted. In the case of nitrate, it is not clear if that nutrient became limiting or sampling occurred when cells were just about to use up the last uM. In this sense, it is essentially impossible to interpret differences in any of the measured parameters between HNHP, LN, and LP conditions. The Fv/Fm data in Fig. 3 heightens my suspicion that cells never truly reached P starvation under LP conditions, as Fv/Fm doesn't show any clear drop in LP compared to HNHP condition at any light or CO2 treatment (compare to Rokitta et al. 2016, for example). In the case of phosphate, perhaps they became limiting at the end, but when? The fact that the increase in PIC/cell reported in many previous studies wasn't observed, but occurred under LN instead, is consistent with the suspicions that the presumed nutrient status was not limiting (and that cell abundance was not being measured properly).

Response: Low DIN or DIP concentration in this study did not limit growth and carbon fixation rates. The reasons are as follows (**Lines 568–579** on pages 26 and 27): (1) In this study, growth rates of *E. huxleyi* were larger than 1 in almost all treatments, and cells divided 1–2 times per day (Fig .1 in the manuscript), which indicates non-limiting nutrient conditions during the incubation. (2) Based on measured PON quota and cell concentration in this study (Figs. 1 and S6), PON concentrations at the end of incubations were estimated to be 7.8–9.3 $\mu mol\ L^{-1}$ at different nutrient conditions (Table S2). These data were closely correlated with molar drawdown of dissolved inorganic nitrogen (DIN) during the incubations. Furthermore, residual 1 $\mu mol\ L^{-1}$ DIN in the final day of the incubation showed non-limitation of growth and POC production rates by nitrogen.

(3) On the other hand, Rokitta et al. (2016) reported that $F_v/F_m$ of _E. huxleyi_ was 50% lower at P-depleted than at P-replete conditions. In this study, $F_v/F_m$ and POC quota were very similar between LP and HNHP treatments (Figs. 2 and S3), which suggest that LP did not limit growth and carbon fixation.

4. I think the approach for analyzing and interpreting the data could be more powerful:
a. The 3-way ANOVA ignores differences between LP and LN conditions
Response: Thanks for the comments of this referee. We re-analyzed the data with a 3-way ANOVA, which shows individual and interactive effects of nutrient concentration, $CO_2$ level and light intensity, and compares differences among HNHP, LN and LP conditions.

A three-way ANOVA was used to determine the main effect of dissolved inorganic nutrient concentration, $pCO_2$, light intensity and their interactions for these variables. A two-way ANOVA was performed to test the main effect of dissolved inorganic nutrient concentration, $pCO_2$ and their interactions on fitted $a$ and $V_{max}$ of growth, POC and PIC production rates. When necessary, a Tukey Post hoc (Tukey HSD) test was used to identify the differences between two $CO_2$ levels, nutrient concentrations or light intensities. These changes are in **Lines 290–298** on page 14.

**Table 2.** Results of three-way ANOVAs of the impacts of dissolved inorganic nutrient concentration, $pCO_2$, light intensity and their interaction on growth rate, $F_v/F_m$, $F_v^{'}/F_m^{'}$, POC and PIC production rates, and PIC:POC ratio.

|  | Factor | _F_ value | _p_ value |
|---|---|---|---|
| Growth rate (d$^{-1}$) | Nut | 264.7 | <0.01 |
|  | C | 875.6 | <0.01 |
|  | L | 2035.8 | <0.01 |
|  | Nut×C | 53.6 | <0.01 |
|  | Nut×L | 84.2 | <0.01 |
|  | C×L | 9.3 | <0.01 |
|  | Nut×C×L | 26.8 | <0.01 |
| $F_v/F_m$ | Nut | 68.6 | <0.01 |
|  | C | 184.7 | <0.01 |
|  | L | 225.8 | <0.01 |
|  | Nut×C | 10.3 | <0.01 |
|  | Nut×L | 8.1 | <0.01 |
|  | C×L | 15 | <0.01 |
|  | Nut×C×L | 5.2 | <0.01 |
| $F_v^{'}/F_m^{'}$ | Nut | 63.9 | <0.01 |
|  | C | 181.8 | <0.01 |
|  | L | 1161.8 | <0.01 |
|  | Nut×C | 51.9 | <0.01 |
|  | Nut×L | 15.3 | <0.01 |
|  | C×L | 9.9 | <0.01 |
|  | Nut×C×L | 8.1 | <0.01 |

| | | | |
|---|---|---|---|
| POC production rate (pg C cell$^{-1}$ d$^{-1}$) | Nut | 11.8 | <0.01 |
| | C | 128.9 | <0.01 |
| | L | 293.7 | <0.01 |
| | Nut×C | 4.9 | =0.01 |
| | Nut×L | 19.0 | <0.01 |
| | C×L | 8.47 | <0.01 |
| | Nut×C×L | 1.94 | =0.06 |
| PIC production rate (pg C cell$^{-1}$ d$^{-1}$) | Nut | 624.4 | <0.01 |
| | C | 142.0 | <0.01 |
| | L | 147.2 | <0.01 |
| | Nut×C | 1.9 | =0.16 |
| | Nut×L | 17.3 | <0.01 |
| | C×L | 8.1 | <0.01 |
| | Nut×C×L | 4.6 | <0.01 |
| PIC:POC ratio | Nut | 326.7 | <0.01 |
| | C | 57.7 | <0.01 |
| | L | 41.8 | <0.01 |
| | Nut×C | 8.3 | <0.01 |
| | Nut×L | 12.5 | <0.01 |
| | C×L | 4.0 | <0.01 |
| | Nut×C×L | 3.3 | <0.01 |

Nut, dissolved inorganic nutrient concentrations (μmol L$^{-1}$); C, $p$CO$_2$ (μatm); L, light intensity (μmol photons m$^{-2}$ s$^{-1}$); POC and POC production rates, particulate organic and inorganic carbon production rates; $F_v/F_m$, maximum photochemical quantum yield; $F_v'/F_m'$ , effective photochemical quantum yield. These changes are in **Lines 1198–1209** on pages 56–58.

b. The 3-way ANOVA approach followed by a posthoc test to identify pairwise differences can be valid, but it doesn't help for identifying patterns. In this case, the Eilers and Peeters model they fit would help, but they only look at the fit of the alpha parameter, when the curves shown in their figures clearly indicate that the other fitted parameters (a, b, c) may be interesting as well.

Response: As suggested by this referee, we used the model of Eilers and Peeters (1988) to fit growth, POC and PIC production rates, and calculated alpha ($a$) and maximum values ($V_{max}$) of growth, POC and PIC production rates. Then a 2-way ANOVA was used to test effects of nutrient and CO$_2$ level on $a$ and $V_{max}$.

A two-way ANOVA was performed to test the main effect of dissolved inorganic nutrient concentration, $p$CO$_2$ and their interactions on fitted $a$ and $V_{max}$ of growth, POC and PIC production rates. This sentence was added in **Lines 294–296** on page 14.

**Table 4.** Results of two-way ANOVAs of the effects of dissolved inorganic nutrient concentration and $pCO_2$ on fitted $a$ and maximum value ($V_{max}$) of growth, POC and PIC production rates. More detailed information is given as in Table 2. These changes are in **Lines 1246–1249** on page 62.

| | | Factor | $F$ value | $p$ value |
|---|---|---|---|---|
| $a$ | Growth rate | Nut | 18.08 | <0.001 |
| | | $CO_2$ | 0.186 | 0.6711 |
| | | Nut×$CO_2$ | 0.398 | 0.6776 |
| | POC production rate | Nut | 7.21 | 0.005 |
| | | $CO_2$ | 7.78 | 0.0121 |
| | | Nut×$CO_2$ | 2.50 | 0.11 |
| | PIC production rate | Nut | 21.73 | <0.001 |
| | | $CO_2$ | 2.32 | 0.145 |
| | | Nut×$CO_2$ | 2.56 | 0.105 |
| $V_{max}$ | Growth rate | Nut | 24.9 | <0.001 |
| | | $CO_2$ | 572.7 | <0.001 |
| | | Nut×$CO_2$ | 14.8 | <0.001 |
| | POC production rate | Nut | 7.301 | 0.0048 |
| | | $CO_2$ | 15.95 | 0.0009 |
| | | Nut×$CO_2$ | 1.91 | 0.177 |
| | PIC production rate | Nut | 56.06 | <0.001 |
| | | $CO_2$ | 86.84 | <0.001 |
| | | Nut×$CO_2$ | 0.168 | 0.85 |

5. What about cell volume effects? As reported recently by Müller et al. (2017), these could be crucial. If I understood that previous study correctly, nutrient limitation seemed to act independently rather than synergistically with ocean acidification when cell volume was accounted for. Of course, that study used continuous culture rather than batch culture/starvation conditions, but still it seems relevant at least to consider. Currently the Discussion seems to ignore some relevant studies such as Müller et al. 2017 that I previously cited, as well as Olson et al. 2016. Further, it needs to be much clearer. Finally, some revision of the English is suggested.

Response: Cell volume is shown in Table R3 (or Table S1 in the supplement). POC and PIC production rates are normalized by cell volume, and the normalized POC and PIC production rates were shown in Figure S5 in the Supplement.

Comparison between the study of Müller et al. (2017) and ours are shown as following:
Using a chemostat culture, Müller et al. (2017) reported that DIN or DIP limitation decreased the POC and PIC production rates (in pg C cell$^{-1}$ d$^{-1}$) by 50% and rising $pCO_2$ levels did not affect POC production rates. However, when normalized to cell volume, nutrient limitation did not affect POC and PIC production rates (in pg C cellV$^{-1}$ d$^{-1}$), and rising $pCO_2$ levels reduced POC and PIC production rates. In our study, decreased DIN or DIP concentration reduced the normalized POC production rates (in pg C cellV$^{-1}$ d$^{-1}$), and increased the normalized PIC production rates at both LC and HC (Fig. S5 in the supplyment). Differing results between the study of Müller et al. (2017)

and ours may result from different experimental setups. Growth was really limited by N or P, cells were cultured in a continuous photon flux, and cell growth was in the stable phase when POC and PIC samples were taken in the study of Müller et al. (2017). While we took POC and PIC samples in the exponential growth phase, and LN or LP did not really limit growth of *E. huxleyi* in our study. These contents were added in the discussion section in **Lines 786–798** on pages 36 and 37.

At 15 °C, 140 μmol photons m$^{-2}$ s$^{-1}$, 28 μmol L$^{-1}$ DIN and 2.4 μmol L$^{-1}$ DIP conditions, rising CO$_2$ increased POC quota (pg C cell$^{-1}$) of *E.huxleyi* strain s2668, while decreased normalized POC quota (pg C cellV$^{-1}$) in study of Olson et al. (2016). In our study, rising CO$_2$ did not significantly affect POC quota (Fig. S3) and normalized POC quota (Fig. S4) at 120 μmol photons m$^{-2}$ s$^{-1}$ and HNHP conditions.

Olson, M. B., Wuori, T. A., Love, B. A., Strom, S. L. : Ocean acidification effects on haploid and diploid *Emiliania huxleyi* strains: Why changes in cell size matter, J. Exp. Mar. Biol. Ecol., 488, 72–82, 2017.

[Figure]

**Figure S4.** At both LC and HC, normalized POC quota (pg C cellV$^{-1}$) of *E. huxleyi* as a function of light intensity at HNHP (**a**), LN (**b**) and LP (**c**) conditions. At both LC and HC, light response of normalized PIC quota (pg C cellV$^{-1}$) of *E. huxleyi* at HNHP (**d**), LN (**e**) and LP (**f**) conditions. The values represent the mean ± standard deviation for four replicates.

6. The Discussion focuses a lot on ETR and photophysiology (Fv/Fm, Fv'/Fm'), which doesn't make a lot of sense. Effects both of high CO2 and of supposed nutrient limitation on photophysiological parameters seem to be subtle in comparison to what they report on growth rates and cell quotas. The light dependence of photosynthesis in E. huxleyi has actually been comparatively well studied, and much of the discussion seems overly speculative and not to focus on some of the curious differences with what has been reported previously (e.g., Houdanetal. 2005 reporting that calcified cells are especially resistant to high PAR).

Response: Thanks for this comment of this referee. We have deleted descriptions of *ETR* in **Lines 232–243, Lines 498–523, Lines 755–770.**

The doubts I have about the study are quite serious, and hopefully my major comments (above) and minor comments (below) help the authors determine where to clarify. Nevertheless, I think the study design may not be appropriate for investigating an interaction between nutrient limitation and acidification. The only way such an experimental design could potentially work for the question planted is with daily and trusted cell counts and nutrient measurements showing when nutrient depletion occurred.

Response: The comments and suggestions of this referee are helpful and useful. As mentioned above, we are confident that cell concentration was measured correctly, and that nutrient concentrations did not limit growth and POC production rates. This is clear.

There are many minor points through the manuscript to address as well. I mention a few:

Line 27 "and exposing phytoplankton to increased light intensities" and lines 52-53 later. I think this is too much over-simplification. At the base of the mixed layer and within the pycnocline, nutrients can be obtained by diffusion across the pycnocline, so phytoplankton will grow and increase in biomass until they compete for light. I do not see how the average light exposure of phytoplankton will necessarily increase. The references cited (Gao et al. 2012 and Hutchinson and Fu 2017) do not explain this (as I mentioned earlier).

Response: Thanks for suggestion of this referee.

This content '*exposing phytoplankton to increased light intensities.*' was replaced by 'affecting the light intensity to which phytoplankton are exposed.' in **Line 27–28** on page 2.

These contents '*This exposes phytoplankton dwelling in the UML to higher light intensities (Gao et al., 2012; Hutchins and Fu, 2017).*' were replaced by 'This affects light exposure of phytoplankton dwelling therein (Steinacher et al. 2010).' These changes are in **Lines 60–62** on page 3.

Lines 101-103 "Interaction of rising CO2 with light appears to affect differentially coccolithophores when grown under different experimental setups." The sentence is not clear. What does "differentially affect coccolithophores" mean? Do those factors affect coccolithophores differently than other phytoplankton or do these factors have contrasting effects or ??

Response: Zhang et al. (2015) reported that compared to 510 µatm, 1050 µatm $CO_2$ decreased growth and POC production rate of *Gephyrocapsa oceanica* at high light intensity. Jin et al. (2017) reported that compared to 395 µatm, 1000 µatm $CO_2$ increased growth and POC production rates of *E. huxleyi* at high sunlight levels. Thus, the studies of Jin et al. (2017) and Zhang et al. (2015) reported contrasting response of growth and POC production rates of *E. huxleyi* and *G. oceanica* to rising $CO_2$. And rising $CO_2$ have contrasting effects on growth and POC production rates of the coccolithophores *E. huxleyi* and *G. oceanica*.

This sentence '*Interaction of rising CO$_2$ with light appears to affect differentially coccolithophores when grown under different experimental setups.*' was replaced by 'These indicate that during growth under different experimental conditions, rising CO$_2$ showed contrasting effects on growth and POC production rates of *E. huxleyi* and *G. oceanica*.' These changes are in **Lines 118–122** on page 6.

Lines 142-144: "added by 2200 µmol L$^{-1}$ bicarbonate (as opposed to 2380 µmol L$^{-1}$ in the original recipe), in order to reflect the alkalinity in the South and East China Seas of about 2200 µmol L$^{-1}$" First, I don't understand why it's important to match the South and East China Seas if they are not specifically using strains isolated from those seas and trying to predict how organisms there will respond. Second, I don't see how bicarbonate concentration is equated with alkalinity, as CO$_3^{2-}$ also contributes to alkalinity, and for alkalinity every unit of CO$_3^{2-}$ counts twice. I think carbonate usually can contribute about a fifth or a fourth or so of total alkalinity (see Zeebe and Wolfgladrow 2001 or other references).

Response: This sentence '*The synthetic seawater medium Aquil was prepared according to Sunda et al. (2005), added by 2200 µmol L$^{-1}$ bicarbonate (as opposed to 2380 µmol L$^{-1}$ in the original recipe), in order to reflect the alkalinity in the South and East China Seas of about 2200 µmol L$^{-1}$ (Chou et al., 2005; Qu et al., 2017).*' was replaced by 'The Aquil medium was prepared according to Sunda et al. (2005) with the addition of 2200 µmol L$^{-1}$ bicarbonate, resulting in initial concentrations of 2200 µmol L$^{-1}$ total alkalinity (TA). This reflects 2200 µmol L$^{-1}$ alkalinity in the South and East China Seas (Chou et al., 2005; Qu et al., 2017).' These changes are in **Lines 162–169** on page 8.

We think this is a logical question, and 2380 µmol L$^{-1}$ bicarbonate can also be added into seawater (Sunda et al. 2005).

In general, $HCO_3^-$ in the natural seawater accounts for more than 90% of the dissolved inorganic carbon (DIC), $CO_3^{2-}$ for about 9%, and CO$_2$ for less than 1% (Zeebe and Wolf-Gladrow 2001). Alkalinity (TA) is calculated as

TA=[$HCO_3^-$]+2[$CO_3^{2-}$]+[$HPO_4^{2-}$]+[OH$^-$]+2[$PO_4^{3-}$]-[H$^+$].......          equation 2

$HCO_3^- \rightleftharpoons H^+ + CO_3^{2-}$          equation 3

$HCO_3^- + H_2O \rightleftharpoons OH^- + H_2CO_3$          equation 4

According to equations 2 and 3, when 1 mole $HCO_3^-$ (1 mol TA) dissociates to 1 mol $CO_3^{2-}$ (2 mol TA) and 1 mol H$^+$ (−1 mol TA), alkalinity did not change. According to equations 2 and 4, when 1 mole $HCO_3^-$ (1 mol TA) reacts with 1 mol H$_2$O to produce 1 mol OH$^-$ (1 mol TA) and 1 mol H$_2$CO$_3$, alkalinity did not change. Thus, bicarbonate concentration is equated with alkalinity.

Sunda, W. G., Price, N. M., and Morel, F. M. M.: Trace metal ion buffers and their use in culture studies, in: Algal culturing techniques, edited by: Andersen R. A., Elsevier Academic Press, London, 53–59, 2005

Zeebe, R. E., Wolf-Gladrow, D. A. : $CO_2$ in seawater: equilibrium, kinetics, isotopes. Amsterdam, Elsevier, 2001.

Lines 161-165: I mentioned above the problems with these lines

Response: See response to **Lines 162–169 (above).**

Line 172: How often were nutrients measured?

Response: Nutrients concentrations were measured before and at the end of experiments.

Lines 182-184: Was pH measured immediately or after storage? pH should be measured immediately, as I understand (within a couple hours is best).

Response: pH was measured within 10 min after the pH sample was taken.

'The $pH_T$ was immediately measured at 20 $^oC$ with a pH meter' (**Line 218**, page 10).

Lines 210-215: I already mentioned the major problems I have with the methodology as described here. Perhaps they can fix that.

Response: Recently, we measured cell concentration using a Cell Lab Quanta SC flow cytometer (Beckman Coulter) and a Z2 Coulter Particle Count and Size Analyzer. Cell concentration was 14,550 cells $mL^{-1}$ when it was measured by a flow cytometer (Fig. R1 and R2 in the response letter) and was 15, 210 cells $mL^{-1}$ when it was measured by the Z2 Coulter Particle Count and Size Analyzer (Fig. R1; 2; 3 in the response letter). Variation in measured cell concentration between two methods was 4.3%. Thus, we don't think that the cell concentration measured by using a Z2 Coulter Particle Count and Size Analyzer cause error.

Cell concentration was also measured by a Cell Lab Quanta SC flow cytometer (Beckman Coulter), and variation in measured cell concentration between two methods was about 4.3% (**Lines 250–253**, page 12)

Line 225: Do they mean "difference" instead of "variance"

Response: '*variance*' was replaced by 'difference' (**Line 265**, page 12).

Lines 251-255: It would be invaluable to know when nutrients were depleted. Do they have data on this?

Response: DIN and DIP concentrations were measured before and at the end of incubations, and these data were shown in Table S2. As mentioned above, DIN and DIP concentration did not limit growth and POC production rates in this study.

Lines 256-264: This whole paragraph is basically redundant with Table 1. Also, it seems "(mean values for the beginning and end of incubations)" means that the beginning and ending values have been averaged together, while in Table S2 the beginning and ending values are given separately. Table S2 is far more useful, especially for assessing the changes in carbonate parameters during the experiment. I would place that table (S2) in the main text, using it to replace the current Table 1. Then in this text the focus should be more on how consistent were carbonate parameters over time and across treatments within the LC and within the HC treatments. Furthermore, when I calculate the averages using the values given in the text immediately before, I get different values (405 for LC and 918 for HC). What is happening? Were some replicates not used?

Response: We agree with this referee that Table 1 was replaced by Table S2 in the main text.

These contents '*The carbonate system parameters (mean values for the beginning and end of incubations) are shown in Table 1. For low $CO_2$ (LC) condition, the $pCO_2$ levels of the media were about 435 μatm at HNHP, 410 μatm at LN and 370 μatm at LP conditions, and the $pH_T$ values (reported on the total scale) were about 8.10 at HNHP, 8.11 at LN and 8.16 at LP. For high $CO_2$ (HC) condition, the $pCO_2$ levels of the media were about 970 μatm at HNHP, 935 μatm at LN and 850 μatm at LP, and the $pH_T$ values were about 7.80 at HNHP, 7.80 at LN, and 7.85 at LP conditions.*' were replaced by 'The carbonate system parameters of the seawater at the beginning and end of the incubation are shown in Table 1. Within the low $CO_2$ (LC) treatment, $pCO_2$ levels of the seawater declined by 16% at HNHP, 19% at LN and 8% at LP, and pH values increased by 0.07 at HNHP, 0.06 at LN and 0.02 at LP (Tukey HSD, all $p < 0.05$). Within the high $CO_2$ (HC) treatment, $pCO_2$ levels of the seawater declined by 23% at HNHP, 21% at LN and 32% at LP, and pH values increased by 0.1 at HNHP, 0.09 at LN and 0.15 at LP (Tukey HSD, all $p < 0.05$). Average $pCO_2$ levels were 410 μatm for all LC conditions, and were 920 μatm for all HC conditions.' These changes are in **Lines 315–328** on page 15.

We checked the carbonate chemistry parameters and found that at LC treatments, the $pCO_2$ levels of the seawater were 439 μatm at HNHP (435 μatm was written in the manuscript (MS)), 409 μatm at LN (410 μatm in the MS) and 371 μatm at LP conditions (370 μatm in the MS); at HC treatments, the $pCO_2$ levels of the media were about 973 μatm at HNHP (970 μatm was written in the MS), 936 μatm at LN (935 μatm in the MS) and 852 μatm under LP conditions (850 μatm in the MS). Average $pCO_2$ levels were 408 μatm (410 μatm in the MS) at all LC conditions, and were 920 μatm (925 μatm in the MS) at all HC conditions. Thus, variation between the written, rounded off data and the original data causes slight differences.

Lines 368-373: This is not a very good description. It seems to exaggerate small differences between HC and LC.

Response: We have deleted the description of rETR in the main text in **Lines 498–523** on pages 23 and 24.

Line 467: I can't find any reference to Fig. 5 in the Results. Why does it appear suddenly in the Discussion? Further, I have problems with this Michaelis-Menten fit: It does not make any sense to fit growth rate in a batch culture (measured from initial and final concentrations) to the initial phosphate concentration. This seems to ignore understanding of phytoplankton macronutrient physiology since Droop. But, as there isn't a clear description of this experiment, I am not sure. Finally, the ability to calculate a half-saturation from the data in the graph would be very limited because there is no value in an intermediate range of growth (growth rate is either 0 or saturated or nearly saturated). For this entire paragraph, the study was not designed to address the details of phosphate metabolism, which has already been fairly extensively studied in this species, and their discussion of the previous work is unclear

Response: In **Fig. 1:** We found that growth rates of *E. hulxeyi* were similar between LP and HNHP treatments at LC and high light conditions. This may be due to high affinity for DIP of *E. huxleyi* (Dyhrman and Palenik, 2003). To test this hypothesis, we performed one experiment that examined the response of growth rate of *E. huxleyi* to DIP concentrations at LC and 200 μmol photons m$^{-2}$ s$^{-1}$.

This text '*Under light saturation condition, relationship of growth rates of E. huxleyi with phosphate concentrations indicated a very high affinity for dissolved inorganic phosphate (DIP) with 0.04 μmol L$^{-1}$ half-saturation constant for DIP (Fig. 5).*' was replaced by 'Furthermore, growth rate of *E. huxleyi* is nearly saturated at 0.25 μmol L$^{-1}$ DIP and is saturated at 0.5 μmol L$^{-1}$ DIP and above. This demonstrated that *E. huxleyi* possesses a high affinity for DIP (Fig. 5) which allowed *E. huxleyi* to take up $PO_4^{3-}$ efficiently.' These changes are in **Lines 646–655** on page 30.

Dyhrman, S. T., and Palenik, B.: Characterization of ectoenzyme activity and phosphate-regulated proteins in the coccolithophorid *Emiliania huxleyi*, J Plank. Res., 25, 1215–1225, https://doi.org/10.1093/plankt/fbg086, 2003.

Line 529: Do the authors consider the ballast effect to be completely irrelevant? Also, I have an issue with considering E. huxleyi as representative of the biogeochemically most important coccolithophores. It is the most numerically abundant, most widespread, and most easy to culture in the laboratory. However, E. huxleyi is definitely not the coccolithophore responsible for most sinking inorganic carbon and it may not be an appropriate model for the responses of other principal groups of coccolithophores, as it (and it's close relatives in the Gephyrocapsa genus) is different from most coccolithophores. For example, E. huxleyi does not require Si for calcification while most do.

Response: We agree with this referee that *E. huxleyi* is definitely not the coccolithophore responsible for most sinking inorganic carbon. However, *E. huxleyi* is the most abundant and most widespread coccolithophore species, and it is true that changes in PIC:POC ratios have the potential to affect sinking rate of *E. huxleyi* (Hoffmann et al., 2015)

'In addition, larger PIC:POC ratios have the potential to accelerate sinking rate of *E. huxleyi* cells, facilitating the export of carbon into deeper waters (Hoffmann et al., 2015). ' These changes are in **Lines 752–754** on pages 34 and 35.

In general, I have a hard time following the Discussion. It lacks clarity and focus, and seems to stray into inadequate review of important but peripheral themes. It's difficult for me to provide more detailed comments as I am not convinced that they know what state of nutrient limitation (or not) the cells were in when harvested.

Response: We thank this referee to spent time to review our manuscript and provide useful comments. We refocus on growth, POC and PIC production rates, and the fitted *a* and maximum values of growth, POC and PIC production rates. As mentioned above, low DIN and DIP did not limit growth and POC production in this study.

**Interactive effects of seawater carbonate chemistry, light intensity and nutrient**

**availability on physiology and calcification of the coccolithophore *Emiliania***

***huxleyi***

**Yong Zhang[1], Feixue Fu[2], David A. Hutchins[2], and Kunshan Gao[1]**

[1]State Key Laboratory of Marine Environmental Science, Xiamen University, Xiamen,

China

[2]Department of Biological Sciences, University of Southern California, Los Angeles,

California

Running head: *carbonate chemistry,* Emiliania huxleyi, *light, nutrients*

Correspondence to: Kunshan Gao (ksgao@xmu.edu.cn)

Keywords: calcification; $CO_2$; coccolithophore; growth; light; nutrient; photosynthesis

**Abstract.** Rising atmospheric carbonate dioxide ($CO_2$) levels lead to increasing $CO_2$ concentration and declining pH in seawater, as well as ocean warming. This enhances stratification and shoals the upper mixed layer (UML), hindering the transport of nutrients from deeper waters and affecting the light intensity to which phytoplankton are exposed exposing phytoplankton to increased light intensities. In the present this study, we investigated combined impacts of $CO_2$ levels (410 µatm (LC) and 9250 µatm (HC)), light intensities (80–480 µmol photons $m^{-2}$ $s^{-1}$) and nutrient concentrations [101 µmol $L^{-1}$ dissolved inorganic nitrogen (DIN) and 10.5 µmol $L^{-1}$ dissolved inorganic phosphate (DIP) (HNHP); 8.8 µmol $L^{-1}$ DIN and 10.5 µmol $L^{-1}$ DIP (LN); 101 µmol $L^{-1}$ DIN and 0.4 µmol $L^{-1}$ DIP (LP)] on growth, photosynthesis and calcification of the coccolithophore *Emiliania huxleyi*. HC and LN synergistically decreased growth rates of *E. huxleyi* at all light intensities. LN and HC synergistically reduced growth and POC production rates. High light intensities compensated for inhibition of LP on growth rates at LC, but exacerbated inhibition of LP at HC. At high light intensity, LP did not limit growth rate at LC but led to increased high-light inhibition of growth rate at HC. These results indicate that the ability of *E. huxleyi* to compete for nitrate and phosphate may be reduced in future oceans with high $CO_2$ and high light intensities. These results showed that effects of nutrient concentrations on physiological rates of *E. huxleyi* were modulated by $CO_2$ level and light intensity. Low nutrient concentrations increased the maximum value and the light-use efficiencies of calcification rate. particulate inorganic carbon quotas and the sensitivity of maximum electron transport rates to light intensity. Light-use

Our results suggest that interactive effects of multiple environmental factors on coccolithophores need to be considered when predicting their contributions to the biological carbon pump and feedbacks to climate change.

**1 Introduction**

Rising atmospheric $CO_2$ level leads to increasing seawater $CO_2$ concentration and decreasing pH, which is known as ocean acidification (OA) (Caldeira and Wickett, 2003). On the other hand, rising atmospheric $CO_2$ also leads to global and ocean warming, which enhances water column stratification and shoals the upper mixed layer (UML) (Wang et al., 2015).

This affects light exposure of phytoplankton dwelling  therein (Steinacher et al. 2010). In addition, enhanced stratification reduces the transport of nutrients from deep oceans to the

UML (Behrenfeld et al., 2006), which reduces the nutrient concentrations in the

UML.

Coccolithophores take up $CO_2$ and/or $HCO_3^-$ from seawater for carboxylation, and use $HCO_3^-$ for calcification which produces coccoliths.

Calcification processes generate $CO_2$ due to production of protons,  and therefore influencing $CO_2$ influx into the oceans (Rost and Riebesell, 2004). Growth rate, particulate organic (POC) and inorganic carbon (PIC) production rates of *Emiliania huxleyi*, the most abundant calcifying coccolithophore species, usually display optimum responses to a broad range of $CO_2$ concentration (Bach et al., 2011).  Growth, POC and PIC production rates could increase, decrease and be unaffected by rising $CO_2$ treatments across a narrow $CO_2$ range, which is dependent on the optimal $CO_2$ levels of these physiological rates and the selected $CO_2$ range (Langer et al., 2009; Richier et al., 2011; Bach et al., 2015; Jin et al., 2017).  Differences in sampling locations, experimental setups,  deviations in the measuring methods and intraspecific differences can generally be responsible for the differential responses of growth, POC and PIC productions to rising $CO_2$ in *E. huxleyi* (Langer et al., 2009; Meyer and Riebesell, 2015).

POC production as well as growth rates usually increase with elevated light intensity, level off at saturated light intensity and decline at inhibited high light intensity in cocolithophores (Zhang et al., 2015; Jin et al., 2017). Reduction in pigment content and effective photochemical quantum yield ($F_v' / F_m'$) are characteristics of photo-acclimation to high light intensity (Geider et al., 1997;

[revised manuscript text omitted]

concentrations were less than 10% (0.5%–9.1%). Langer et al. (2013) found that growth of cells on the fourth to sixth days of batch cultures was in the exponential phase even at 3 μmol L$^{-1}$ NO$_3^-$ or at 0.29 μmol L$^{-1}$ PO$_4^{3-}$ with the same $E.$ $huxleyi$

strain. In this study, all parameters were measured on the fourth to the sixth days, so it is most likely that cells in all treatments were sampled in the exponential growth phase.

**2.2 Nutrient concentrations, total alkalinity and $pH_T$ measurements**

Sampling started at 10:30 a.m. and finished at 12:00 a.m.. 50 mL samples for determination of inorganic nitrogen and phosphate concentrations were syringe-filtered (0.22 μm pore size, Haining) and measured using a scanning spectrophotometer (Du 800, Beckman Coulter) according to Hansen and Koroleff (1999).

Carbonate chemistry parameters were calculated from total alkalinity (TA) and, $pH_T$ (total scale), phosphate, temperature, and salinity using the CO₂ System CO2SYS (Pierrot et al., 2006). In the final days of incubation, 25 mL samples for TA measurements were filtered (0.22 μm pore size, Syringe Filter) by gentle pressure with 200 mbar in the pump (GM-0.5A, JINTENG) and stored at 4 $^o$C for a maximum of 7 days. TA was measured at 20 $^o$C by potentiometric titration (AS-ALK1+, Apollo SciTech) according to Dickson et al. (2003). Samples for $pH_T$ measurements were syringe-filtered (0.22 μm pore size), and the bottles were filled with overflow and closed immediately. The $pH_T$ was immediately measured at 20 $^o$C with a pH meter (Benchtop pH, Orion 8102BN) calibrated with an equimolal equimolar pH buffer (Tris•HCl, Hanna) for sea water media (Dickson, 1993). Carbonic acid constants $K_1$ and $K_2$ were calculated according to taken from Roy et al. (1993).

**2.3 Measurements of photochemical parameters**

The effective photochemical quantum yield ($F_v' / F_m'$) and maximum photochemical quantum yield ($F_v / F_m$) of photosystem II (PSII) were assessed using a XE-PAM

(Walz, Germany) at 1:00 p.m.. 3 ml samples were taken from the incubation bottles, and $F_v' / F_m'$ values were measured immediately at active light intensities similar to the incubation light levels. 3 mL samples were kept  in the dark for 15 min at

$^o$C, and $F_v / F_m$ values were determined at a measuring light intensity of 0.3 μmol photons m$^{-2}$ s$^{-1}$ and a saturation pulse of  5000 μmol photons m$^{-2}$ s$^{-1}$ with 0.8 s.

**2.4 Cell density measurements**

At the end of the incubation, about 25 ml samples were taken from the incubation bottles at about 2:30 p.m.. Cell densities were measured by using a Z2 Coulter

Particle Counter and Size Analyzer (Beckman Coulter). The diameter of detected particles was set to 3 to 7 μm in the instrument, which excludes detached coccoliths because the diameter of coccolith is less than 3 μm (Müller et al., 2012). Cell concentration was also measured by a Cell Lab Quanta SC flow cytometer (Beckman

Coulter), and variation in measured cell concentration between two methods was about 4.3%. Growth rate (μ) was calculated according to the equation: $\mu = (\ln N_1 - \ln$

$N_0) / d$, where $N_0$ is 200 cells mL$^{-1}$ and $N_1$ is the cell concentration in the final days of experiment, and $d$ is the growth time span in days.

**2.5 Particulate organic (POC) and inorganic carbon (PIC) measurements**

GF/F filters, pre-combusted at 450 $^o$C for 8 h, were used to filter the samples of total particulate carbon (TPC) and particulate organic carbon (POC). TPC and POC

samples were stored darkly at –20$^o$C. For POC measurements, samples were fumed with HCl for 12 h to remove inorganic carbon, and samples for TPC measurements were not treated with HCl. All samples were dried at 60 $^o$C for 12 h, and analyzed using a Perkin Elmer Series II CHNS/O Analyzer 2400 instrument (Perkin Elmer

Waltham, MA). Particulate inorganic carbon (PIC) quota was calculated as the variance difference between TPC quota and POC quota. POC and PIC production rates were calculated by multiplying  cellular contents with μ (d$^{-1}$), respectively. Variations in measured carbon content between the four replicates were calculated to be 1–13% in this study.

**2.6 Response of growth rate of *E. huxleyi* to different dissolved inorganic phosphate (DIP) concentrations**

L Aquil media were enriched with 100 μmol L$^{-1}$ DIN, aerated for 24 h at 20 $^{o}$C with air containing 400 μatm $p$CO$_2$, sterilized by filtration (0.22 μm pore size, Polycap 75 AS, Whatman) and then pumped into autoclaved 250 mL PC bottles. 10 μmol L$^{-1}$, 3 μmol L$^{-1}$, 1.5 μmol L$^{-1}$, 0.5 μmol L$^{-1}$, 0.25 μmol L$^{-1}$ DIP (finial concentration) were respectively added into Aquil media with three replicates at each DIP concentration. 200 cells mL$^{-1}$ was inoculated to Aquil media and all samples were cultured at 200 μmol photons m$^{-2}$ s$^{-1}$ for 4 days before starting the experiment. Finial cell concentration was measured by using a Z2 Coulter Particle Count and Size Analyzer (Beckman Coulter).

**2.7 Data analysis**

Responses of growth rates, POC and PIC  production rates, PIC:POC ratio to incubation light intensities were fitted using the model provided by Eilers and Peeters (1988): $y = \dfrac{PAR}{a \times PAR^2 + b \times PAR + c}$, where the parameters $a$, $b$ and $c$ are fitted in a least square manner. The apparent light use efficiency, the slope ($\alpha$), for each light response curve was estimated as $\alpha = 1/c$. The maximum values ($V_{max}$) of growth, POC and PIC production rates were calculated according to $V_{max} = \dfrac{1}{b + 2\sqrt{ac}}$:

A three-way ANOVA was used to determine the main effect of dissolved inorganic  nutrient concentration, $p$CO$_2$, light intensity and their interactions for these variables.  A two-way ANOVA was performed to test the main effect of dissolved inorganic nutrient concentration, $p$CO$_2$ and their interactions on fitted $a$ and $V_{max}$ of growth, POC and PIC production rates. When necessary, a Tukey Post hoc (Tukey HSD) test was used to identify the differences between two CO$_2$ levels,  nutrient concentrations or light intensities. A Shapiro-Wilk's test was conducted to test residual normality and a Levene test was used to test for variance homogeneity of significant data. Statistical analysis was conducted by using R and significant level was set at $p < 0.05$.

**3 Results**

**3.1 Dissolved inorganic nitrogen and phosphate concentrations, and carbonate chemistry paremeters**

At the HNHP condition, dissolved inorganic nitrogen (DIN) and phosphate (DIP) concentrations were $101 \pm 1.1$ μmol L$^{-1}$ and $10.5 \pm 0.2$ μmol L$^{-1}$, respectively, at the beginning of the experiments, and were $92.8 \pm 1.6$ μmol L$^{-1}$ and $9.7 \pm 0.2$ μmol L$^{-1}$ in the final days of the experiment (Table S2). At the LN condition, DIN concentrations were 8.8 ± 0.1 μmol L$^{-1}$ at the beginning of the experiment and were 1.0 ± 0.4 μmol L$^{-1}$ at the end of the experiment. In the LP treatment, DIP concentrations were 0.4 ± 0.1 μmol L$^{-1}$ at the beginning of the experiment, and were below the detection limit (< 0.04 μmol L$^{-1}$) at the end of the experiment.

The carbonate system parameters of the seawater at the beginning and end of the incubation  are shown in Table 1. ~~For low CO$_2$ (LC) condition, the $pCO_2$ levels of the media were about 435 μatm at HNHP, 410 μatm at LN and 370 μatm under LP conditions, and the $pH_T$ values (reported on the total scale) were about 8.10 at HNHP, 8.11 at LN and 8.16 at LP. For high CO$_2$ (HC) condition, the $pCO_2$ levels of the media were about 970 μatm at HNHP, 935 μatm at LN and 850 μatm at LP, and the $pH_T$ values were about 7.80 at HNHP, 7.80 at LN, and 7.85 at LP conditions. .~~ Within the low CO$_2$ (LC) treatment, $pCO_2$ levels of the seawater declined by 16% at HNHP, 19% at LN and 8% at LP, and pH values increased by 0.07 at HNHP, 0.06 at LN and 0.02 at LP (Tukey HSD, all $p < 0.05$). Within the high CO$_2$ (HC) treatment, $pCO_2$ levels of the seawater declined by 23% at HNHP, 21% at LN and 32% at LP, and pH values increased by 0.1 at HNHP, 0.09 at LN and 0.15 at LP (Tukey HSD, all $p < 0.05$). Average $pCO_2$ levels were 410 μatm for all LC conditions, and were 920 μatm for all HC conditions.

**3.2 Growth rate**

Under each nutrient condition, at both LC and HC, growth rates of *E. huxleyi*

increased with elevated light intensity up to 200 µmol photons m$^{-2}$ s$^{-1}$ and significantly declined thereafter ( Tukey  HSD, all df = 2, all $p < 0.001$) (Fig. 1; Table 2). Compared with LC, growth rates at HC were 2%–7%

lower at HNHP (Tukey HSD, $p < 0.05$), 5%–9% lower at LN (Tukey HSD, $p < 0.01$)

and 3%–24% lower at LP (Tukey HSD, $p < 0.01$), respectively (Table 3). Under LP

treatment, HC-induced reduction of growth rate was large at high light intensity (Fig. 1c).

At LC, growth rate at LN was similar with that at HNHP under limited light intensity with 80 µmol photons m$^{-2}$ s$^{-1}$ (Tukey HSD, df = 1, $p = 0.82$), and was significantly lower than at HNHP under optimal and supra-optimal light intensities (Tukey HSD, both df = 1, $p < 0.01$ for 200 µmol photons m$^{-2}$ s$^{-1}$ ; $p = 0.005$

for 480 µmol photons m$^{-2}$ s$^{-1}$ ). At HC, growth rates at LN were significantly lower than those at HNHP under limited, optimal and supra-optimal light intensities (Tukey HSD, all df = 1, $p < 0.01$ for 80, 200, 480 µmol photons m$^{-2}$ s$^{-1}$ ).

At LC and at 80 µmol photons m$^{-2}$ s$^{-1}$, growth rate at LP was lower than at HNHP

(Tukey HSD, df = 1, $p < 0.001$); while at 120–480 µmol photons m$^{-2}$ s$^{-1}$, growth rates were no significant differences between LP and HNHP (Tukey HSD, all df = 1, all $p >$

0.1) (Fig. 1; Table 3). At HC and at 80, 120 and 480 µmol photons m$^{-2}$ s$^{-1}$, growth rates were significantly lower at LP than at HNHP; at 200 and 320 µmol photons m$^{-2}$

s$^{-1}$, growth rates were not significantly different between LP and HNHP (Tukey HSD, both df = 1, both $p > 0.05$).

**3.3 POC quota**

Under HNHP or LP conditions, at LC, POC quotas were not significantly different among 80, 120 and 200 μmol photons m$^{-2}$ s$^{-1}$ and increased with increased light intensity from 200 to 480 μmol photons m$^{-2}$ s$^{-1}$ (Three-way ANOVA; Tukey Post hoc, both df = 1, both $p < 0.01$); while at HC, POC quotas increased with elevated light intensity up to 480 μmol photons m$^{-2}$ s$^{-1}$ (Fig. 2a,c; Tables 2; 3). At LN, at both LC and HC, POC quotas at 320 μmol photons m$^{-2}$ s$^{-1}$ were significantly larger than at other light intensities (Fig. 2b).

At HNHP or at LN, POC quotas did not show significant differences between HC and LC (Fig. 2a,b). At LP, at 80 μmol photons m$^{-2}$ s$^{-1}$, POC quotas were significantly larger at LC than at HC (df = 1, $p = 0.003$), while at 480 μmol photons m$^{-2}$ s$^{-1}$, they were lower (df = 1, $p = 0.001$).

At both LC and HC, POC quotas were not significantly different between LN and HNHP at 80–320 μmol photons m$^{-2}$ s$^{-1}$, while they were lower at LN than at HNHP at 480 μmol photons m$^{-2}$ s$^{-1}$ ($p < 0.01$). At both LC and HC, POC quotas were not significantly different between LP and HNHP at 80–480 μmol photons m$^{-2}$ s$^{-1}$ (all df = 1, all $p > 0.05$).

**3.4 PIC quota**

At HNHP or at LN, under either LC or HC, PIC quotas increased with increasing light intensity until 320 μmol photons m$^{-2}$ s$^{-1}$ (Three-way ANOVA; Tukey Post hoc, all df = 1, all $p < 0.001$) and then leveled off with further increasing light intensity (Fig.

2d,e; Tables 2; 3). At LP under LC conditions, PIC quotas increased significantly when light intensity increased from 80 to 200 μmol photons m$^{-2}$ s$^{-1}$ and significantly declined thereafter (both df = 1, both $p < 0.001$) (Fig. 2f), while at LP and HC, there were no significant differences among the light levels (all $p > 0.05$).

At HNHP or at LN, PIC quotas were larger at LC than at HC (all df = 1, all $p > 0.05$ at 80, 120, 200 treatments; both $p < 0.01$ at 320 and 480 treatments) (Fig. 2d,e). Under LP conditions at 200 and 320 μmol photons m$^{-2}$ s$^{-1}$, PIC quotas were larger at LC than at HC (both df = 1, both $p < 0.05$) (Fig. 2f).

At both LC and HC, PIC quotas were larger at LN than at HNHP (all df = 1, all $p > 0.05$ at 80 treatment; $p < 0.05$ at 120–480 treatments) (Fig. 2d,e). For both LC and HC conditions at 80–200 μmol photons m$^{-2}$ s$^{-1}$, PIC quotas were larger at LP than at HNHP (all df = 1, all $p < 0.05$), while at 320 and 480 μmol photons m$^{-2}$ s$^{-1}$, they were not significantly different between LP and HNHP (Fig. 2f).

**3.5 PIC:POC ratio**

At HNHP under LC, PIC:POC ratio increased with elevated light intensity until 320 μmol photons m$^{-2}$ s$^{-1}$ and significantly declined thereafter (Three way ANOVA, Tukey Post hoc, df = 1, $p < 0.05$) (Fig. 2g; Tables 2; 3), while at HC, they were not significantly different between light treatments (all $p > 0.05$). At LN in both LC and HC treatments, PIC:POC ratio increased when light intensity increased from 80 to 200 μmol photons m$^{-2}$ s$^{-1}$ and were not significantly different between 200, 320 and 480 μmol photons m$^{-2}$ s$^{-1}$ (Fig. 2h). At LP under LC conditions, PIC:POC ratio

**3. 3 $F_v$/$F_m$ and $F_v^{'}/F_m^{'}$**

$F_v$/$F_m$ and $F_v^{'}/F_m^{'}$ showed the same patterns (Fig. 2).

 At all nutrient and $CO_2$ levels, $F_v$/$F_m$ and $F_v^{'}/F_m^{'}$ decreased with elevated light intensity until 480 μmol photons m$^{-2}$ s$^{-1}$ ( Tukey

HSD, all df = 4, all $p$ < 0.01) (Fig. 2a f; Tables 2; 3).

Only at LP and at 480 μmol photons m$^{-2}$ s$^{-1}$ $F_v$/$F_m$ value was significantly larger at LC than at HC ( Tukey HSD, df = 1,  $p$ < 0.01)

(Fig. 2a,c). At LN in the light intensities of 80–480 μmol photons m$^{-2}$ s$^{-1}$,

$F_v$/$F_m$ values were not significantly different between LC and HC (Tukey HSD, all df

= 1, all $p > 0.05$) (Fig. 23b).

At both LC and HC, at 80–480 µmol photons m$^{-2}$ s$^{-1}$ $F_v/F_m$ did not show significant differences between LN and HNHP (Tukey HSD, all df = 1, all $p > 0.05$), and at 480 µmol photons m$^{-2}$ s$^{-1}$, they were lower at LP than at HNHP at both LC and HC (Tukey HSD, both df = 1, both $p < 0.05$) (Fig. 23a,c).

At HNHP  and at 80 to 480 µmol photons m$^{-2}$ s$^{-1}$, $F_v'/F_m'$ values were similar between LC and HC (Tukey HSD, all df = 1, all $p > 0.05$) (Fig. 23d). At LN under 200 µmol photons m$^{-2}$ s$^{-1}$, and at LP under 480 µmol photons m$^{-2}$ s$^{-1}$, $F_v'/F_m'$ values were larger at LC than at HC (Tukey HSD, both df = 1, both $p < 0.01$) (Fig. 23e,f).

At LC under 200 µmol photons m$^{-2}$ s$^{-1}$, $F_v'/F_m'$ values were significantly larger at LN than at HNHP, as well as at LP  than at HNHP (Tukey HSD, both df = 1, both $p < 0.05$) (Fig. 23d,e,f). At HC under 480 µmol photons m$^{-2}$ s$^{-1}$ $F_v'/F_m'$ values were significantly lower at LP than at HNHP (Tukey HSD, df = 1, $p < 0.01$) (Fig. 23d,f).

**3.4 POC production rate**

At HNHP or LP conditions, at both LC and HC, POC production rates increased significantly with increasing light intensity until 480 µmol photons m$^{-2}$ s$^{-1}$ (Tukey HSD, all df = 4, $p < 0.01$) (Fig. 3a,c; Tables 2; 3). At LN, at both LC and HC, POC production rate increased when light intensity increased from 80 to 320 µmol photons m$^{-2}$ s$^{-1}$ (Tukey HSD, both df = 3, $p < 0.01$) and significantly declined thereafter (Fig. 3b).

At HNHP or LN conditions, at all light intensities, POC production rates did not show significant differences between HC and LC treatments (Tukey HSD, all df = 1, all $p > 0.05$) (Fig. 3a,b). At LP, at 80 and 320 μmol photons m$^{-2}$ s$^{-1}$, POC production rates were significantly larger at LC than at HC (Tukey HSD, both df = 1, both $p < 0.01$) (Fig. 3c).

At both LC and HC, at 80–320 μmol photons m$^{-2}$ s$^{-1}$, POC production rates were not significantly different between LN and HNHP, and between LP and HNHP (Tukey HSD, all df = 1, all $p > 0.05$); while at 480 μmol photons m$^{-2}$ s$^{-1}$, they were lower at LN or LP than at HNHP conditions (Tukey HSD, df = 1, $p < 0.05$) (Fig. 3a,b,c).

**3.5 PIC production rate**

At HNHP or LN conditions, at both LC and HC, PIC production rates increased significantly when light intensity increased from 80–320 μmol photons m$^{-2}$ s$^{-1}$ (Tukey HSD, all df = 3, all $p < 0.05$) (Fig. 3d,e; Tables 2; 3), and declined thereafter (Tukey HSD, df = 1, $p < 0.05$ at LC; $p > 0.1$ at HC). At LP condition, at both LC and HC, PIC production rates increased significantly until 200 μmol photons m$^{-2}$ s$^{-1}$ (Tukey HSD, both df = 2, both $p < 0.05$) (Fig. 3f), and declined with further increases in light intensity (Tukey HSD, df = 2, $p < 0.05$ at LC; $p > 0.1$ at HC) (Fig. 3f).

At HNHP or LN conditions, at 320 μmol photons m$^{-2}$ s$^{-1}$, PIC production rates were larger at LC than at HC (Tukey HSD, df = 1, $p < 0.05$) (Fig. 3d,e). At LP, at all light intensities, PIC production rates were no significant differences between LC and HC treatments (Tukey HSD, all df = 1, all $p > 0.05$) (Fig. 3f).

At both LC and HC, at all light intensities, PIC production rates were larger at LN than at HNHP (Tukey HSD, all df = 1, $p > 0.05$ at 80 µmol photons m$^{-2}$ s$^{-1}$; all $p <$ 0.05 at 120–480 µmol photons m$^{-2}$ s$^{-1}$) (Fig. 3d,e). At LC, at 120 and 200 µmol photons m$^{-2}$ s$^{-1}$, PIC production rates were significantly larger at LP than at HNHP (Tukey HSD, both df = 1, both $p < 0.05$). At HC, at all light intensities, PIC production rates were not significantly different between LP and HNHP conditions (Tukey HSD, all df = 1, all $p > 0.05$) (Fig. 3d,f).

**3.6 PIC:POC ratio**

At HNHP and at LC, PIC:POC ratio increased with increasing light intensity until 320 µmol photons m$^{-2}$ s$^{-1}$ (Tukey HSD, df = 3, $p < 0.01$) and slightly declined thereafter (Tukey HSD, df = 1, $p = 0.13$) (Fig. 3g; Tables 2; 3). At HNHP and at HC, they were not significantly different between light treatments (Tukey HSD, df = 4, $p > 0.05$). At LN, at both LC and HC, PIC:POC ratio increased significantly when light intensity increased from 80 to 200 µmol photons m$^{-2}$ s$^{-1}$ (Tukey HSD, both df = 2, $p < 0.01$) and did not show significant differences at 200–480 µmol photons m$^{-2}$ s$^{-1}$ (Tukey HSD, both df = 2, $p > 0.1$) (Fig. 3h). At LP and at LC, PIC:POC ratio increased with increasing light intensity until 200 µmol photons m$^{-2}$ s$^{-1}$, and declined with further increasing light intensity (Tukey HSD, df = 2, $p < 0.05$) (Fig. 3i). At LP and at HC, they were not significantly different between light intensities (Tukey HSD, df = 4, $p >$ 0.05) (Fig. 3i).

At HNHP or at LP, at 80–480 µmol photons m$^{-2}$ s$^{-1}$, PIC:POC ratio were not significantly different between LC and HC treatments (Tukey HSD, all df = 1, all $p >$

0.05) (Fig. 3g,i). At LN, at 320 and 480 µmol photons m$^{-2}$ s$^{-1}$, PIC:POC ratios were larger at LC than at HC (Tukey HSD, both df = 1, both $p < 0.05$) (Fig. 3h).

 At both LC and HC, at 80–480 µmol photons m$^{-2}$ s$^{-1}$, PIC:POC ratios were larger at

LN than at HNHP (Tukey HSD, all df = 1, $p > 0.05$ at 80 µmol photons m$^{-2}$ s$^{-1}$; $p <$

0.05 at 120–480 µmol photons m$^{-2}$ s$^{-1}$) (Fig. 3g,h). At both LC and HC, at 80–200

µmol photons m$^{-2}$ s$^{-1}$ PIC:POC ratios were larger at LP than at HNHP (Tukey HSD, all df = 1, all $p < 0.05$) (Fig 3g,i), while at 320 and 480 µmol photons m$^{-2}$ s$^{-1}$, they were not significantly different between LP and HNHP (Tukey HSD, both df = 1, both

$p > 0.05$) (Fig 3g,i).

**3.7 $ETR_{max}$**

At HNHP and at LC, $ETR_{max}$ increased significantly with increasing light intensities until 200 µmol photons m$^{-2}$ s$^{-1}$ (df = 1, $p < 0.01$), and leveled off with further increasing light intensities (Fig. 3g; Tables 2; 3). At HNHP and at HC, with light intensities increasing from 80 to 120 µmol photons m$^{-2}$ s$^{-1}$, $ETR_{max}$ increased remarkably (df = 1, $p < 0.01$), and declined significantly when light intensities further increased to 480 µmol photons m$^{-2}$ s$^{-1}$ (df = 1, $p < 0.05$). At LN or at LP, under both

LC and HC, $ETR_{max}$ increased with increasing light intensities until 200 µmol photons m$^{-2}$ s$^{-1}$ and declined thereafter (all df = 1, all $p < 0.01$) (Fig. 3h,i).

 At HNHP and only at 480 µmol photons m$^{-2}$ s$^{-1}$, $ETR_{max}$ was lower at HC than at

LC (df =1, $p < 0.01$) (Fig. 3g; Table 3). At LN across the light range of 80–480 µmol photons m$^{-2}$s$^{-1}$, $ETR_{max}$ values were similar between HC and LC (Fig. 3h). At LP

under 320 µmol photons m$^{-2}$s$^{-1}$, $ETR_{max}$ was larger at HC than at LC; while at 480

µmol photons m$^{-2}$s$^{-1}$, they were lower (both df =1, both $p < 0.05$) (Fig. 3i).

At both LC and HC from 80–480 µmol photons m$^{-2}$s$^{-1}$, $ETR_{max}$ values were larger at LN than at HNHP (Tukey Post hoc, all df = 1, $p < 0.01$ for the 120, 200 and 320

treatments at LC; $p > 0.05$ for the 80 and 480 treatments at LC; $p < 0.01$ for the 80,

200, 320 and 480 treatments at HC; $p > 0.05$ for the 120 treatment at HC) (Fig. 3g,h).

At LC under 80 µmol photons m$^{-2}$s$^{-1}$, $ETR_{max}$ was lower at LP than at HNHP (df = 1,

$p > 0.1$); while at 120–480 µmol photons m$^{-2}$s$^{-1}$, $ETR_{max}$ values were larger (Tukey

Post hoc, all df = 1, $p > 0.05$ for 120, 320 and 480 treatments; $p < 0.01$ for 200

treatment) (Fig. 3g,h). At HC under 80 and 120 µmol photons m$^{-2}$s$^{-1}$, $ETR_{max}$ values were lower at LP than at HNHP (Tukey Post hoc, both df = 1, $p > 0.1$ for the 80

treatment; $p < 0.01$ for the 120 treatment), while at 200–480 µmol photons m$^{-2}$s$^{-1}$, they were larger (Tukey Post hoc, all df = 1, $p < 0.01$ for 200 and 320 treatments; $p > $

0.1 for 480 treatment).

**3.87 Apparent light use efficiency ($\alpha$) and maximum value forof growth, POC and PIC production rates**

At each nutrient condition, $\alpha$ values of fitted curves of growth, POC and PIC

production rates were not significantly different between LC and HC, with the exception of $\alpha$ of PIC production rate at LP (Tukey HSD, df = 1, $p < 0.05$) (Fig. 4). At both LC and HC, $\alpha$ values of fitted curves of growth and POC production rates did not show significant differences between HNHP, LN and LP conditions, with the exception of $a$ of POC production rate between HNHP-LC and LP-HC conditions (Tukey HSD, df = 1, $p < 0.05$) (Fig. 4c). At LN under both LC and HC, and at LP under LC, $\alpha$ values of PIC production rates were larger than those of POC production rates, which were larger than those of growth rates (Tukey HSD, all df = 1, all $p < 0.01$) (Fig. 4a,c,e).

~~At HNHP under both LC and HC, $\alpha$ values of fitted curves for POC and PIC production rates were not significantly different, and they were significantly larger than those for growth rates (both df = 1, both $p < 0.01$) (Fig. 4a). At LN under both LC and HC, and at LP under LC, $\alpha$ values for PIC production rates were larger than those for POC production rates, which were larger than those for growth rates (all df = 1, all $p < 0.01$) (Fig. 4b,c). At LP and HC, $\alpha$ values for POC and PIC production rates did not show significant differences and they were larger than that for growth rates (Fig. 4c).~~

At HNHP, LN or LP condition, maximum growth rates were significantly larger at LC than at HC (Tukey HSD, all df = 1, all $p < 0.05$) (Fig. 4b). At both LC and HC, maximum growth rates were larger at HNHP than at LN (Tukey HSD, both df = 1, both $p < 0.05$), and they were similar between HNHP and LP (Tukey HSD, both df = 1, both $p > 0.05$) (Fig. 4b).

At each nutrient condition, maximum POC production rates were slightly larger at LC than at HC (Tukey HSD, all df = 1, all $p > 0.05$) (Fig. 4d). At LC, maximum POC production rate was lower at LN than at HNHP and LP (Tukey HSD, df = 1, $p < 0.05$ between LN and HNHP; $p > 0.05$ between LN and LP). At HC, they did not show significant differences between HNHP, LN and LP conditions (Tukey HSD, df = 2, $p > 0.05$) (Fig. 4d).

At HNHP, LN or LP condition, maximum PIC production rates were significantly larger at LC than at HC (Tukey HSD, all df = 1, all $p < 0.05$) (Fig. 4f). At both LC and HC, maximum PIC production rates were larger at LN than at HNHP or LP (Tukey HSD, df = 2, $p < 0.05$) (Fig. 4f).

**4 Discussion**

In this study, growth rates of *E. huxleyi* were larger than 1 in almost all treatments, and cells divided 1–2 times per day (Fig .1), which indicates non-limiting nutrient conditions during the incubation. Based on measured PON quota and cell concentration in this study (Figs. 1 and S6), PON concentrations at the end of incubations were estimated to be 7.8–9.3 $\mu$mol $L^{-1}$ at different nutrient conditions (Table S2). These data were closely correlated with molar drawdown of dissolved inorganic nitrogen (DIN) during the incubations. Furthermore, residual 1 $\mu$mol $L^{-1}$

DIN in the final day of the incubation showed non-limitation of growth and POC production rates by nitrogen. On the other hand, Rokitta et al. (2016) reported that $F_v/F_m$ of *E. huxleyi* was 50% lower at P-depleted than at P-replete conditions. In this study, $F_v/F_m$ and POC quota were very similar between LP and HNHP treatments (Figs. 2 and S3), which suggest that LP did not limit growth and carbon fixation.

In this study, we investigated synergistic negative effects of low nutrient concentrations and rising $p$CO$_2$ on growth rates, especially at limiting low and inhibiting high light intensities. Notably, high light intensities compensated for inhibition of LP on growth rates at LC. LN reduced POC quota and its sensitivity to light intensity. Both LN and LP increased PIC quotas, PIC:POC ratio, and *ETR* efficiency.

**4.1 Low nutrient dissolved inorganic nitrogen concentrations and high $p$CO$_2$ level synergistically reduced growth rate.**

Langer et al. (2013) detected that cell numbers on the fourth to sixth days during cultures were in the exponential growth phase even at 3 μmol L$^{-1}$ NO$_3^-$ or at 0.29 μmol L$^{-1}$ PO$_4^{3-}$ with the same *E. huxleyi* strain. In addition, other *E. huxleyi* strains were in the exponential phase of growth on the fourth to the seventh days in the cultures with 2.5–8 μmol L$^{-1}$ NO$_3^-$ or at 0.4–0.55 μmol L$^{-1}$ PO$_4^{3-}$ (Perrin et al., 2016; Rokitta et al., 2016). All parameters were measured on the fourth to the sixth days, and it is most likely that cells at all treatments were sampled in the exponential growth phase in this study.

Less energy availability limited growth rates of *E. huxleyi* at lower light intensities, while reduction in growth rates at high light intensities could be related to photooxidative damage or photoinhibition (Fig. 1), because high light intensity can constantly damage the reaction centers of photosystem II (PSII) of *E. huxleyi* (Fig. 2)

and maximize electron turnover rate through PSII centers (Behrenfeld et al. 1998; Ragni et al., 2008).

Lower growth rates at HC than at LC are due to the fact that at HC the negative effect of high [H$^+$] on growth rate was larger than positive effects of increased CO$_2$

and HCO$_3^-$ concentrations (Bach et al., 2011).

et al., 2013), thus resulting in reduced nitrate assimilation. In addition, LN concentration was shown to down-regulate transcripts of genes related to nitrate reductase (NRase) activity, synthesis of amino acids, RNA polymerases and nitrogen metabolism in *E. huxleyi* (Bruhn et al., 2010; Rouco et al., 2013; Rokitta et al., 2014), which led to lower overall biosynthetic activity and decreased the growth rates (Fig. 1). Synergistic effects of LN and HC on growth rates indicate that these conditions may inhibit cellular metabolic activity simultaneously (Fig. 1) (Sciandra et al., 2003). In fact, intracellular $[H^+]$ have been reported to be higher in HC-grown than in LC-grown *E. huxleyi* cells (Suffrian et al., 2011). To transport extra $H^+$ out of cells, *E. huxleyi* at HC need more transporters and energy, but LN is likely to limit the synthesis of these transporters and energy supply (Fig. S6), therefore, it exacerbated the negative effects of high $[H^+]$ on growth of *E. huxleyi* (Fig. S6) (Bruhn et al., 2010).

**4.2 Effect of low dissolved inorganic phosphate concentration on growth rate was modulated by light intensity and $CO_2$ level.**

*E. huxleyi* possesses an exceptional phosphorus acquisition capacity, which could allow it to dominate in phosphate-limiting environments (Dyhrman and Palenik, 2003). In this study, at low levels of light intensity, uptake of phosphate could be energy limited, thus their growth was more inhibited at LP (Fig. 1c). In this study, low light intensity not only limited cell growth but also was suggested to limit phosphate uptake rates (Nalewajko and Lee, 1983). In this case, compared to the HNHP

condition, growth rates of *E. huxleyi* at LP condition were more likely to be limited by low-light intensity (Fig. 1a,c). High light intensity provided energy for cells to take up P, and cells at LP condition need to consume more energy to up-regulate P uptake (Nalewajko and Lee, 1983) which may lead to decreased high-light inhibition of growth rate at LP than at HNHP condition under LC.  Furthermore, growth rate of *E. huxleyi* was nearly saturated at 0.25 µmol L$^{-1}$ DIP and was saturated at 0.5 µmol L$^{-1}$ DIP and above. This demonstrated that *E. huxleyi* possesses a high affinity for DIP (Fig. 5) which allowed *E. huxleyi* to take up PO$_4^{3-}$ efficiently . Rokitta et al. (2016) showed that even though PO$_4^{3-}$ concentration in the culture media declined to zero (undetectable), cell number sustained to increase for 4 days, which indicate that *E. huxleyi* cells could store phosphorus  for later use. Consequently, high energy consumption mechanism,  efficient uptake and storage capacity for phosphorus  in *E. huxleyi* could account for there being no significant differences in growth rates between LP and HNHP at LC and  high light intensities. In fact, as reported previously, higher growth rates of *E. huxleyi* at LP in comparison to

HP were found during exponential growth phase in batch cultures (Rokitta et al., 2016). In natural seawaters, *E. huxleyi* usually starts to bloom following diatom blooms (Tyrrell and Merico, 2004). which may be related to a high growth rate of *E. huxleyi* at low nutrient concentrations.  **  **

Rising $CO_2$ was found to lead to higher phosphorous requirements for growth, carbon fixation and nitrogen uptake,  in *E. huxleyi* (Matthiessen et al., 2012; Rouce et al., 2013). At HC, higher phosphorous requirements may lead to lower growth rates at LP in comparison to HNHP (Fig. 1a,c).  $CO_2$  $CO_2$  $HCO_3^-$  In addition, at LP, cell volume was 17% larger at HC than at LC under the highest light intensity (Table S1). Large cell volume can directly lead to lower growth rates and reduce nutrient uptake by cells, thereby limiting growth. Another possible reason for low tolerance to high-light intensity in growth rate at LP and HC might be a combined effect of LP and HC on the carbon concentrating mechanism (CCM) of *E. huxleyi*. LP or HC is hypothesized to down-regulate the activity of CCM in the green algae *Chlorella emersonii* and in *E.*

*huxleyi*, respectively (Rost and Riebesell, 2004; Beardall et al. 2005). When grown at HC, LP may minimize the activity of CCM of *E. hulxeyi*, which could lead to less energy cost for maintaining high efficient CCM. The saved energy in the HC- and LP-grown cells might have exacerbated photo-inhibition. In summary, high phosphorous requirement, large cell volume and less energy consumption at LP and HC conditions may lead to increased high-light inhibition of growth rates of *E. huxleyi* (Fig. 1).

**4.3 Low dissolved inorganic nutrient concentration and high $p$CO₂ level synergistically reduce POC production rate.**

~~At LC, *E. huxleyi* mainly uses external HCO₃⁻ as an inorganic carbon source for photosynthesis and calcification, and increasing light intensities are able to increase HCO₃⁻ uptake rates (Kottmeier et al., 2016). This may explain why POC and PIC quotas and production rates increased with increasing light intensity (Figs. 2 and S5). HC down-regulates gene expression related to the HCO₃⁻ transporter (Rokitta et al., 2012) and decreases the HCO₃⁻ uptake rate in *E. huxleyi* (Kottmeier et al. 2016), leading to lower PIC quotas at HC than at LC (Fig. 2).~~ At LC, *E. huxleyi* mainly uses external $HCO_3^-$ as an inorganic carbon source to synthesize POC and PIC, and increasing light intensity increases the $HCO_3^-$ uptake rate (Kottmeier et al., 2016) which results in large POC and PIC production rates at high light intensity (Fig. 3). However, at HC, expression of gene related to the $HCO_3^-$ transporter was down-regulated and the $HCO_3^-$ uptake rate was reduced (Rokitta et al., 2012;

Kottmeier et al. 2016), which lead to lower PIC production rates at HC than at LC. Meanwhile, cells at HC can increase $CO_2$ uptake to compensate for low_$HCO_3^-$ -uptake for photosynthetic C fixation (Kottmeier et al., 2016), which explain the similar POC quotas between HC and LC (Fig.  S3).

LN was found to reduce the enzymatic function and cellular metabolic rates such as reduce synthesis and activity of ribulose-1,5-biphosphate carboxylase/oxygenase (RUBISCO), which decreases POC quota at both LC and HC (Falkowski et al., 1989; Rokitta et al., 2014) (Fig. S3 and S6). Furthermore, in comparison to LC, lower cell division rates at HC further reduce POC production rates at LN.  On the other hand, large cell volume at LP and HC condition was responsible for low cell division rate and low POC production rate in comparison to HNHP (Figs 1, 3 and S3).

**4.4 Low dissolved inorganic nutrient concentrations facilitate calcification rate .**

LC and LP treatment decreased cell division rates, elongated cell cycle, and increased coccolith production of *E. huxleyi* in the darkness (Paasche and Brubak, 1994). In the present work, however, we found slightly faster cell division (growth) and identical calcification rates at LP and high light intensities (Figs. 1c, 2f and S5). LP has been shown to up-regulate the genes involved in calcium binding proteins such as the glutamic acid related to synthesize of coccolith, calcium homeostasis and transcription factor (*cmyb*) (Wahlund et al., 2004; Dyhrman et al., 2006), and facilitates the formation of cytoplasmic membrane bodies (Shemi et al., 2016). These are related to the pathways associated with production of coccoliths (Young and Henriksen, 2003) and may also be responsible for larger PIC quotas at LP. Nimer and Merrett (1993) reported that decreased DIN concentration facilitates calcification rate of *E. huxleyi*. This is consistent with our result. Due to lower photosynthetic carbon fixation rate and larger calcification rate at LN in comparison to HNHP (Fig. 3), we could expect that at LN, a high proportion of intracellular $HCO_3^-$ or $CO_2$ was reallocated to synthesize particulate inorganic carbon. On the other hand, at LP, slightly larger PIC production rate is likely due to larger cell volume in comparison to HNHP (Fig. 3).

Calcification of coccolithophores makes an important contribution to marine carbonate counter pumps in the pelagic ocean (Rost and Riebesell, 2004). Enhanced calcification of *E. huxleyi* at low nutrient concentrations implies that blooms of calcifying *E. huxleyi* diminish the potential of the oceanic $CO_2$ uptake compared to non-calcifying phytoplankton blooms. On the other hand In addition, larger PIC:POC

ratios  have the potential to accelerate  sinking rate of *E. huxleyi* cells, facilitating the export of carbon into deeper waters (Hoffmann et al., 2015).

$_{max}$~~ values were severely limited by low energy input. Supraoptimal light intensities have been found to significantly reduce the abundance of several proteins involved in repair and assembly of PSII, such as repair of photodamaged Psb D1 proteins in the reaction center of PSII of *E. huxleyi* (McKew et al., 2013). These suggest that high light intensity is likely to do great damage to the PSII structure and then reduce the *ETR*. Especially at HC, supraoptimal light intensity and saved energy from down-regulation of CCM activity synergistically decreased *ETR* (Fig. 3).~~

$_{max}$~~ at LN or at LP and at saturating light intensities likely resulted from larger calcification rates (Figs. 2 and 3). On the other hand, growth, photosynthetic carbon fixation and nitrogen uptake need energy originating from electron transport (Zhang et al., 2015). At LP and at limiting levels of light intensity, lower growth, photosynthetic carbon and nitrate assimilation rates coincided with lower *ETR* (Figs. 1-3), implying correlations of these physiological processes.~~

To provide organic carbon fixed by photosynthesis to support growth and other metabolic processes, cells need to maintain larger light-use efficiency (*α*) for POC production rates (Fig. 4).

aided by higher light-use efficiencies (Fig. 4). In addition, besides taking up inorganic carbon sources and Ca²⁺ from the seawater to calcify, cells need extra energy to expel H⁺ generated during calcification from the cells (Jin et al., 2017), these may also account for higher light-use efficiencies for PIC production rates. To calcify, *E. hulxeyi* cells need to take up $HCO_3^-$ and $Ca^{2+}$ from the seawater, which consumes energy. Besides that, they also need to extrude $H^+$ generated during calcification into the cytosol to favour the conversion of $HCO_3^-$ to $CO_3^-$, which also needs some energy (Paasche 2002). Thus, calcification is an energy consuming process. To maintain large calcification rate at low nutrient concentration, cells possess high light-use efficiencies and can then obtain more energy to take up $HCO_3^-$ and $Ca^{2+}$, and extrude $H^+$ into the cytosol.

Using a chemostat culture, Müller et al. (2017) reported that DIN or DIP limitation decreased the POC and PIC production rates (in pg C cell⁻¹ d⁻¹) by 50% and rising $pCO_2$ levels did not affect POC production rates. However, when normalized to cell volume, nutrient limitation did not affect POC and PIC production rates (in pg C cellV⁻¹ d⁻¹), and rising $pCO_2$ levels reduced POC and PIC production rates. In our study, decreased DIN or DIP concentration reduced the normalized POC production rates (in pg C cellV⁻¹ d⁻¹), and increased the normalized PIC production rates at both LC and HC (Fig. S5). Differing results between the study of Müller et al. (2017) and ours may result from different experimental setups. Growth was really limited by N or P, cells were cultured in a continuous photon flux, and cell growth was in the stable phase when POC and PIC samples were taken in the study of Müller et al. (2017).

While we took POC and PIC samples in the exponential growth phase, and LN or LP

did not really limit growth of *E. huxleyi* in our study.

Nutrient availability, $CO_2$ level and light intensity significantly interacted to affect growth rate, POC and PIC  production rates, $F_v / F_m$, and $F_v' / F_m'$ $_{max}$

(Table 2). Obviously, the question how growth, carbon fixation and calcification rates of *E. huxleyi* would respond to ocean global changes needs to be examined under multiple stressors and under natural environmental variations (Feng et al., 2008, 2017).

Although both HC and HL reduced calcification rates of *E. huxleyi*, low nutrient concentrations showed dominant positive effects on PIC quota or calcification rate (Fig. 3d−f), suggesting that calcification of

*E. huxleyi* may increase in the future pelagic oceans. Our study demonstrates that complex effects of multiple environmental drivers on phytoplankton require us to investigate the underlying mechanisms of these interactions, in order to comprehend how ecological and biogeochemical functions of key phytoplankton groups may respond to ocean global changes.

 **Figure Legends**

 **Figure 1.** Growth rate of *Emiliania huxleyi* as a function of light intensities at low

 $p$CO$_2$ (LC) and high $p$CO$_2$ levels (HC) at high dissolved inorganic nitrogen (DIN) and

 phosphate (DIP) concentrations (HNHP)(**a**), low DIN and high DIP concentrations

 (LN) (**b**), or high DIN and low DIP concentrations (LP) (**c**). The  lines in each

 panel were fitted using the model provided by Eilers and Peeters (1988). The values

 represent the mean ± standard deviation for four replicates.

 **Figure 2.** At both LC and HC, maximum photochemical quantum yield ($F_v/F_m$) of *E.*

 *huxleyi* as a function of light intensity at HNHP (**a**), LN (**b**) and LP (**c**) conditions. At

 both LC and HC, light response of effective photochemical quantum yield ($F_v^{'}/F_m^{'}$) of

 *E. huxleyi* at HNHP (**d**), LN (**e**) and LP (**f**) conditions. The values represent the mean

 ± standard deviation for four replicates.

 **Figure 3.** At both LC and HC, POC production rate of *E. huxleyi* as a function

 of light intensit at HNHP (**a**), LN (**b**) and LP (**c**) conditions. At both LC and HC,

 light response of PIC production rate of *E. huxleyi* at HNHP (**d**), LN (**e**) and

 LP (**f**) conditions. At both LC and HC, light response of PIC:POC ratio of *E. huxleyi*

 at HNHP (**g**), LN (**h**) and LP (**i**) conditions. The  lines in each panel were fitted

 using the model provided by Eilers and Peeters (1988). The values represent the mean

 ± standard deviation for four replicates.

**Figure 3.** At both LC and HC, maximum photochemical quantum yields ($F_v/F_m$) of *E. huxleyi* as a function of light intensities at HNHP (**a**), LN (**b**) and LP (**c**) conditions. At both LC and HC, light responses of effective photochemical quantum yields ($F_v'/F_m'$) of *E. huxleyi* at HNHP (**d**), LN (**e**) and LP (**f**) conditions. At both LC and HC, light responses of fitted maximum electron transport rate ($ETR_{max}$) of *E. huxleyi* at HNHP (**g**), LN (**h**) and LP (**i**) conditions. The values represent the mean ± standard deviation for four replicates.

**Figure 4.** At both LC and HC, fitted $a$ (**a**) and maximum (**b**) of growth rate at HNHP, LN and LP conditions. At both LC and HC, fitted $a$ (**c**) and maximum (**d**) of POC prodution rate at HNHP, LN and LP conditions. At both LC and HC, fitted $a$ (**e**) and maximum (**f**) of PIC production rate at HNHP, LN and LP conditions. $\alpha$ was the slope of fitted lines for growth, POC and PIC production rates. Different letters showed statistical differences based on the Tukey post hoc test. The values represent the mean ± standard deviation for four replicates.

**Figure 4.** At both LC and HC, apparent light-use efficiency ($\alpha$) for growth, POC and PIC production rates of *E. huxleyi* at HNHP (**a**), LN (**b**) and LP (**c**) conditions. $\alpha$ was the slope of fitted lines for growth, POC and PIC production rates. μ represents growth rate, POCpro represents POC production rate and PICpro represents PIC production rate. Different letters showed statistically difference. The values represent the mean ± standard deviation for four replicates.

**Figure 5.** Growth rate of *E. huxleyi* as a function of dissolved inorganic phosphate (DIP) concentration. DIN concentration was

μmol L$^{-1}$ in all culture media, and DIP concentrations were set up to

0.25 μmol L$^{-1}$, 0.5 μmol L$^{-1}$, 1.5 μmol L$^{-1}$, 3 μmol L$^{-1}$ and 10 μmol L$^{-1}$ in the culture media. All samples were incubated at 200 μmol photons m$^{-2}$ s$^{-1}$ and at 410 μatm

*p*CO$_2$ for 4 days, and tThe values represent the mean ± standard deviation for three replicates.

**Table 1.** Carbonate chemistry parameters  of the  seawater at the beginning and end of the incubations at different nutrient conditions and $pCO_2$ levels. TA and pH samples were collected and measured before and in the final days of the experiment.

| | $pCO_2$ (µatm) | pH (total scale) | TA (µmol $L^{-1}$) | DIC (µmol $L^{-1}$) | $HCO_3^-$ (µmol $L^{-1}$) | $CO_3^{2-}$ (µmol $L^{-1}$) | $CO_2$ (µmol $L^{-1}$) | $\Omega$ calcite |
|---|---|---|---|---|---|---|---|---|
|  |  |  |  |  |  |  |  |  |
| |  |  |  |  |  |  |  |  |
|  |  |  |  |  |  |  |  |  |
| |  |  |  |  |  |  |  |  |
|  |  |  |  |  |  |  |  |  |
| |  |  |  |  |  |  |  |  |

| | | | $pCO_2$ (µatm) | pH (total scale) | TA (µmol $L^{-1}$) | DIC (µmol $L^{-1}$) | $HCO_3^-$ (µmol $L^{-1}$) | $CO_3^{2-}$ (µmol $L^{-1}$) | $CO_2$ (µmol $L^{-1}$) | $\Omega$ calcite |
|---|---|---|---|---|---|---|---|---|---|---|
| HNHP | LC | Before | 510±17[a] | 8.04±0.01[a] | 2228±17[a] | 2004±20[a] | 1829±21[a] | 159±2[a] | 16±1[a] | 3.8±0.1[a] |
| | | End | 428±57[b] | 8.11±0.05[b] | 2225±24[a] | 1967±22[b] | 1773±34[b] | 180±18[a] | 14±2[b] | 4.3±0.5[a] |
| | HC | Before | 1210±53[a] | 7.71±0.02[a] | 2219±19[a] | 2131±22[a] | 2010±22[a] | 81±2[a] | 39±2[a] | 1.9±0.1[a] |
| | | End | 935±139[b] | 7.81±0.06[b] | 2225±24[a] | 2098±12[b] | 1966±17[b] | 102±14[b] | 30±4[b] | 2.4±0.3[b] |
| LN | LC | Before | 483±23[a] | 8.06±0.02[a] | 2204±10[a] | 1973±10[a] | 1796±13[a] | 162±6[a] | 16±1[a] | 3.9±0.1[a] |
| | | End | 391±39[b] | 8.12±0.03[b] | 2123±38[b] | 1866±45[b] | 1679±48[b] | 175±9[b] | 13±1[b] | 4.2±0.2[b] |
| | HC | Before | 1126±66[a] | 7.73±0.02[a] | 2201±3[a] | 2105±7[a] | 1983±9[a] | 85±4[a] | 36±2[a] | 2.02±0.1[a] |
| | | End | 888±114[b] | 7.82±0.05[b] | 2142±38[b] | 2016±47[b] | 1890±49[b] | 98±8[b] | 29±4[b] | 2.4±0.2[b] |
| LP | LC | Before | 397±16[a] | 8.14±0.02[a] | 2248±30[a] | 1982±22[a] | 1777±17[a] | 192±8[a] | 13±1[a] | 4.6±0.2[a] |
| | | End | 365±24[b] | 8.16±0.02[a] | 2219±20[b] | 1942±22[b] | 1731±25[b] | 199±8[a] | 12±1[b] | 4.8±0.2[a] |
| | HC | Before | 1140±110[a] | 7.73±0.04[a] | 2215±41[a] | 2128±46[a] | 2005±46[a] | 86±7[a] | 37±4[a] | 2.1±0.2[a] |
| | | End | 780±43[b] | 7.88±0.02[b] | 2228±14[a] | 2084±11[b] | 1941±12[b] | 117±6[b] | 25±1[b] | 2.8±0.1[b] |

HNHP, 101 μmol $L^{-1}$ dissolved inorganic nitrogen (DIN) and 10.5 μmol $L^{-1}$ dissolved inorganic phosphate (DIP); LN, 8.8 μmol $L^{-1}$ DIN; LP, 0.4 μmol $L^{-1}$ DIP. Different letters indicate statistical difference between  the beginning and end of the incubations within low or high $p$CO$_2$ level (Tukey Post hoc, $p < 0.01$). The values are expressed as mean  ± SD calculated from all light intensities

.

**Table 2.** Results of three-way ANOVAs of the impacts of dissolved inorganic  nutrient concentration, $p$CO$_2$, light intensity and their interaction on growth rate,  $F'_v/F'_m$, POC and PIC production rates, and PIC:POC ratio.

| | Factor | F value | p value | Factor | F value | p value |
|---|---|---|---|---|---|---|
| Growth rate (d$^{-1}$) | N | 215.9 | <0.001 | P | 1015.5 | <0.001 |
| | C | 547.8 | <0.001 | C | 213.3 | <0.001 |
| | L | 1330.4 | <0.001 | L | 1863.8 | <0.001 |
| | N×C | 9.1 | =0.004 | P×C | 147.6 | <0.001 |
| | N×L | 11.8 | <0.001 | P×L | 274.4 | <0.001 |
| | C×L | 18.3 | <0.001 | C×L | 11.1 | <0.001 |
| | N×C×L | 4.1 | =0.006 | P×C×L | 19.7 | <0.001 |
| POC quota (pg C cell$^{-1}$) | N | 27.1 | <0.001 | P | 13.7 | <0.001 |
| | C | 0.6 | =0.435 | C | 0.1 | =0.731 |
| | L | 34.7 | <0.001 | L | 103.2 | <0.001 |
| | N×C | 13.2 | <0.001 | P×C | 14.5 | <0.001 |
| | N×L | 17.9 | <0.001 | P×L | 0.4 | =0.780 |
| | C×L | 1.0 | =0.432 | C×L | 21.6 | <0.001 |
| | N×C×L | 1.9 | =0.125 | P×C×L | 7.3 | <0.001 |
| PIC quota (pg C cell$^{-1}$) | N | 544.0 | <0.001 | P | 619.1 | <0.001 |
| | C | 70.5 | <0.001 | C | 105.8 | <0.001 |
| | L | 71.2 | <0.001 | L | 55.3 | <0.001 |
| | N×C | 2.8 | =0.098 | P×C | 6.3 | =0.015 |
| | N×L | 7.0 | <0.001 | P×L | 9.7 | <0.001 |
| | C×L | 11.4 | <0.001 | C×L | 2.2 | =0.078 |
| | N×C×L | 0.6 | =0.639 | P×C×L | 7.0 | <0.001 |
| PIC:POC ratio | N | 934.6 | <0.001 | P | 395.0 | <0.001 |
| | C | 81.8 | <0.001 | C | 9.1 | =0.004 |
| | L | 30.9 | <0.001 | L | 47.6 | <0.001 |
| | N×C | 6.6 | =0.013 | P×C | 13.4 | <0.001 |
| | N×L | 9.8 | <0.001 | P×L | 14.4 | <0.001 |
| | C×L | 6.8 | <0.001 | C×L | 1.5 | =0.202 |
| | N×C×L | 0.7 | =0.567 | P×C×L | 4.7 | =0.002 |
| $F_v/F_m$ | N | 335.8 | <0.001 | P | 171.2 | <0.001 |
| | C | 1.5 | =0.229 | C | 189.6 | <0.001 |
| | L | 246.7 | <0.001 | L | 153.9 | <0.001 |
| | N×C | 16.1 | <0.001 | P×C | 34.8 | <0.001 |
| | N×L | 4.8 | =0.002 | P×L | 13.8 | <0.001 |
| | C×L | 12.6 | <0.001 | C×L | 10.7 | <0.001 |
| | N×C×L | 4.6 | =0.003 | P×C×L | 2.6 | =0.048 |

| | | | | | | |
|---|---|---|---|---|---|---|
| $F_v'/F_m'$ | N | 10.1 | =0.002 | P | 675.4 | <0.001 |
| | C | 33.6 | <0.001 | C | 134.0 | <0.001 |
| | L | 670.5 | <0.001 | L | 1007.7 | <0.001 |
| | N×C | 11.7 | =0.001 | P×C | 195.5 | <0.001 |
| | N×L | 3.4 | =0.014 | P×L | 22.8 | <0.001 |
| | C×L | 14.6 | <0.001 | C×L | 8.2 | <0.001 |
| | N×C×L | 12.6 | <0.001 | P×C×L | 3.5 | =0.012 |
| $ETR_{max}$ $(mol\ e^-\ g^{-1}\ Chl\ a\ h^{-1})$ | N | 811.2 | <0.001 | P | 335.2 | <0.001 |
| | C | 67.9 | <0.001 | C | 71.3 | <0.001 |
| | L | 176.6 | <0.001 | L | 625.4 | <0.001 |
| | N×C | 11.2 | =0.001 | P×C | 20.2 | <0.001 |
| | N×L | 15.3 | <0.001 | P×L | 151.0 | <0.001 |
| | C×L | 4.8 | =0.002 | C×L | 35.1 | <0.001 |
| | N×C×L | 12.7 | <0.001 | P×C×L | 9.4 | <0.001 |

| | Factor | $F$ value | $p$ value |
|---|---|---|---|
| Growth rate (d$^{-1}$) | Nut | 264.7 | ≤0.01 |
| | C | 875.6 | ≤0.01 |
| | L | 2035.8 | ≤0.01 |
| | Nut×C | 53.6 | ≤0.01 |
| | Nut×L | 84.2 | ≤0.01 |
| | C×L | 9.3 | ≤0.01 |
| | Nut×C×L | 26.8 | ≤0.01 |
| $F_v/F_m$ | Nut | 68.6 | ≤0.01 |
| | C | 184.7 | ≤0.01 |
| | L | 225.8 | ≤0.01 |
| | Nut×C | 10.3 | ≤0.01 |
| | Nut×L | 8.1 | ≤0.01 |
| | C×L | 15 | ≤0.01 |
| | Nut×C×L | 5.2 | ≤0.01 |
| $F_v'/F_m'$ | Nut | 63.9 | ≤0.01 |
| | C | 181.8 | ≤0.01 |
| | L | 1161.8 | ≤0.01 |
| | Nut×C | 51.9 | ≤0.01 |
| | Nut×L | 15.3 | ≤0.01 |
| | C×L | 9.9 | ≤0.01 |
| | Nut×C×L | 8.1 | ≤0.01 |
| POC production rate (pg C cell$^{-1}$ d$^{-1}$) | Nut | 11.8 | ≤0.01 |
| | C | 128.9 | ≤0.01 |
| | L | 293.7 | ≤0.01 |
| | Nut×C | 4.9 | =0.01 |
| | Nut×L | 19.0 | ≤0.01 |
| | C×L | 8.47 | ≤0.01 |

| | | | |
|---|---|---|---|
| | Nut×C×L | 1.94 | =0.06 |
| PIC production rate (pg C cell$^{-1}$ d$^{-1}$) | Nut | 624.4 | ≤0.01 |
| | C | 142.0 | ≤0.01 |
| | L | 147.2 | ≤0.01 |
| | Nut×C | 1.9 | =0.16 |
| | Nut×L | 17.3 | ≤0.01 |
| | C×L | 8.1 | ≤0.01 |
| | Nut×C×L | 4.6 | ≤0.01 |
| PIC:POC ratio | Nut | 326.7 | ≤0.01 |
| | C | 57.7 | ≤0.01 |
| | L | 41.8 | ≤0.01 |
| | Nut×C | 8.3 | ≤0.01 |
| | Nut×L | 12.5 | ≤0.01 |
| | C×L | 4.0 | ≤0.01 |
| | Nut×C×L | 3.3 | ≤0.01 |

N, dissolved inorganic nitrogen (DIN, μmol L$^{-1}$); P, dissolved inorganic phosphate (DIP, μmol L$^{-1}$) Nut, dissolved inorganic nutrient concentrations (μmol L$^{-1}$); C, $p$CO$_2$ (μatm); L, light intensity (μmol photons m$^{-2}$ s$^{-1}$); POC and POC production rates quota, particulate organic and inorganic carbon contentproduction rates; PIC quota, particulate inorganic carbon content; $F_v/F_m$, maximum photochemical quantum yield; $F_v^{'}/F_m^{'}$, effective photochemical quantum yield; ETR$_{max}$, maximum electron transport rate.

**Table 3.** Experimental treatments, growth rate, $F_v/F_m$, $F_v'/F_m'$, particulate organic (POC) and inorganic carbon (PIC) production rates, and PIC:POC ratio  in dilute bath cultures.

| Initial N/P | $p$CO$_2$ | I | Growth rate | POC quota | PIC quota | PIC:POC | $F_v/F_m$ | $F_v'/F_m'$ | ETR$_{max}$ |
|---|---|---|---|---|---|---|---|---|---|
| 101/ 10.5 | 435 | 80 | 1.11(0.02) | 8.8(0.5) | 1.6(0.4) | 0.19(0.05) | 0.59(0.01) | 0.58(0.03) | 1.25(0.07) |
| | | 120 | 1.21(0.03) | 9.1(0.3) | 2.3(0.7) | 0.25(0.08) | 0.55(0.00) | 0.54(0.01) | 1.52(0.12) |
| | | 200 | 1.37(0.02) | 8.5(0.6) | 2.8(0.7) | 0.33(0.08) | 0.55(0.01) | 0.48(0.01) | 1.65(0.02) |
| | | 320 | 1.29(0.03) | 9.7(1.0) | 5.0(1.3) | 0.52(0.16) | 0.47(0.03) | 0.37(0.03) | 1.58(0.09) |
| | | 480 | 1.17(0.03) | 12.3(0.7) | 3.5(0.4) | 0.28(0.04) | 0.45(0.06) | 0.31(0.02) | 1.63(0.06) |
| | 970 | 80 | 1.06(0.01) | 7.7(0.4) | 0.9(0.1) | 0.12(0.02) | 0.58(0.01) | 0.57(0.02) | 1.16(0.01) |
| | | 120 | 1.19(0.03) | 8.9(0.2) | 2.2(0.4) | 0.25(0.04) | 0.54(0.01) | 0.52(0.01) | 1.69(0.16) |
| | | 200 | 1.32(0.01) | 8.2(0.7) | 2.3(0.4) | 0.28(0.06) | 0.53(0.01) | 0.47(0.01) | 1.61(0.01) |
| | | 320 | 1.21(0.02) | 9.9(0.8) | 2.9(0.7) | 0.30(0.09) | 0.49(0.03) | 0.37(0.02) | 1.60(0.09) |
| | | 480 | 1.16(0.01) | 11.7(1.2) | 1.7(0.4) | 0.14(0.02) | 0.33(0.03) | 0.28(0.02) | 1.24(0.1) |
| 8.8/ 10.5 | 410 | 80 | 1.08(0.01) | 7.3(0.4) | 2.9(0.6) | 0.39(0.09) | 0.59(0.01) | 0.58(0.01) | 1.44(0.04) |
| | | 120 | 1.21(0.01) | 8.4(0.4) | 4.7(0.9) | 0.57(0.12) | 0.57(0.00) | 0.55(0.01) | 2.03(0.11) |
| | | 200 | 1.31(0.01) | 8.1(0.3) | 5.9(0.8) | 0.74(0.08) | 0.59(0.01) | 0.53(0.01) | 2.50(0.15) |
| | | 320 | 1.29(0.01) | 9.9(0.4) | 8.7(0.7) | 0.87(0.07) | 0.45(0.04) | 0.37(0.04) | 2.10(0.07) |
| | | 480 | 1.12(0.02) | 7.9(0.8) | 6.8(0.8) | 0.87(0.17) | 0.41(0.03) | 0.35(0.04) | 1.69(0.14) |
| | 936 | 80 | 1.00(0.01) | 7.8(0.3) | 2.4(0.7) | 0.31(0.11) | 0.59(0.01) | 0.57(0.01) | 1.66(0.04) |
| | | 120 | 1.11(0.01) | 8.9(0.5) | 4.3(0.3) | 0.48(0.04) | 0.55(0.01) | 0.54(0.02) | 1.86(0.06) |
| | | 200 | 1.25(0.01) | 8.3(0.5) | 5.6(0.8) | 0.68(0.09) | 0.54(0.01) | 0.44(0.01) | 2.35(0.16) |
| | | 320 | 1.21(0.01) | 9.7(0.2) | 5.4(0.4) | 0.56(0.05) | 0.50(0.01) | 0.41(0.03) | 2.00(0.08) |
| | | 480 | 1.06(0.06) | 7.2(1.1) | 4.2(0.6) | 0.54(0.06) | 0.37(0.02) | 0.33(0.04) | 1.76(0.15) |
| 101/ 0.4 | 372 | 80 | 1.00(0.02) | 8.7(0.3) | 3.2(0.5) | 0.36(0.06) | 0.59(0.01) | 0.55(0.01) | 1.01(0.05) |
| | | 120 | 1.24(0.01) | 8.3(0.2) | 4.2(0.4) | 0.51(0.05) | 0.59(0.01) | 0.55(0.01) | 1.58(0.04) |
| | | 200 | 1.39(0.01) | 8.1(0.3) | 5.3(0.5) | 0.66(0.09) | 0.56(0.01) | 054(0.02) | 2.10(0.06) |
| | | 320 | 1.31(0.02) | 9.6(0.5) | 4.1(0.6) | 0.43(0.08) | 0.47(0.02) | 0.38(0.01) | 1.85(0.06) |
| | | 480 | 1.18(0.05) | 10.8(0.6) | 2.7(0.5) | 0.25(0.03) | 0.38(0.08) | 0.29(0.04) | 1.61(0.18) |
| | 852 | 80 | 0.97(0.02) | 6.9(0.5) | 2.6(0.4) | 0.38(0.04) | 0.58(0.01) | 0.54(0.02) | 0.91(0.03) |
| | | 120 | 1.08(0.01) | 9.0(0.1) | 3.7(0.7) | 0.41(0.07) | 0.55(0.01) | 0.49(0.01) | 1.29(0.02) |
| | | 200 | 1.27(0.01) | 8.1(0.1) | 4.0(0.3) | 0.49(0.04) | 0.55(0.01) | 0.51(0.02) | 2.16(0.07) |
| | | 320 | 1.22(0.01) | 8.6(0.1) | 3.1(0.4) | 0.36(0.05) | 0.47(0.03) | 0.37(0.03) | 2.18(0.09) |
| | | 480 | 0.90(0.01) | 12.8(0.6) | 3.5(0.6) | 0.28(0.06) | 0.25(0.03) | 0.17(0.01) | 1.21(0.09) |

| Initial N/P | $p$CO$_2$ | L | Growth rate | $F_v/F_m$ | $F_v'/F_m'$ | POC/cell/d | PIC/cell/d | PIC:POC |
|---|---|---|---|---|---|---|---|---|
| 101/ 10.5 | 439 | 80 | 1.11(0.02) | 0.59(0.01) | 0.58(0.03) | 9.70(0.45) | 1.81(0.43) | 0.19(0.05) |
| | | 120 | 1.21(0.03) | 0.55(0.00) | 0.54(0.01) | 11.03(0.28) | 2.80(0.88) | 0.25(0.08) |
| | | 200 | 1.37(0.02) | 0.55(0.01) | 0.48(0.01) | 11.67(0.71) | 3.82(0.97) | 0.33(0.08) |
| | | 320 | 1.29(0.03) | 0.47(0.03) | 0.37(0.03) | 12.59(1.35) | 6.44(1.67) | 0.52(0.16) |
| | | 480 | 1.17(0.03) | 0.45(0.06) | 0.31(0.02) | 14.54(0.89) | 4.06(0.47) | 0.28(0.04) |
| | 973 | 80 | 1.06(0.01) | 0.58(0.01) | 0.57(0.02) | 8.25(0.30) | 0.99(0.14) | 0.12(0.02) |
| | | 120 | 1.19(0.03) | 0.54(0.01) | 0.52(0.01) | 10.50(0.19) | 2.65(0.39) | 0.25(0.04) |
| | | 200 | 1.32(0.01) | 0.53(0.01) | 0.47(0.01) | 10.74(1.06) | 3.02(0.61) | 0.28(0.06) |
| | | 320 | 1.21(0.02) | 0.49(0.03) | 0.37(0.02) | 12.04(0.91) | 3.55(0.92) | 0.30(0.09) |
| | | 480 | 1.16(0.01) | 0.33(0.03) | 0.28(0.02) | 13.50(1.32) | 2.02(0.50) | 0.14(0.02) |
| 8.8/ 10.5 | 409 | 80 | 1.08(0.01) | 0.59(0.01) | 0.58(0.01) | 7.93(0.39) | 3.08(0.61) | 0.39(0.09) |
| | | 120 | 1.21(0.01) | 0.57(0.00) | 0.55(0.01) | 10.26(0.40) | 5.78(1.10) | 0.57(0.12) |
| | | 200 | 1.31(0.01) | 0.59(0.01) | 0.53(0.01) | 10.60(0.30) | 7.81(1.00) | 0.74(0.08) |
| | | 320 | 1.29(0.01) | 0.45(0.04) | 0.37(0.04) | 12.76(0.47) | 11.17(1.10) | 0.87(0.07) |
| | | 480 | 1.12(0.02) | 0.41(0.03) | 0.35(0.04) | 8.84(0.91) | 7.60(0.85) | 0.87(0.17) |
| | 936 | 80 | 1.00(0.01) | 0.59(0.01) | 0.57(0.01) | 7.85(0.37) | 2.39(0.74) | 0.31(0.11) |
| | | 120 | 1.11(0.01) | 0.55(0.01) | 0.54(0.02) | 9.89(0.53) | 4.76(0.35) | 0.48(0.04) |
| | | 200 | 1.25(0.01) | 0.54(0.01) | 0.44(0.01) | 10.37(0.60) | 7.02(0.94) | 0.68(0.09) |
| | | 320 | 1.21(0.01) | 0.50(0.01) | 0.41(0.03) | 11.73(0.20) | 6.53(0.53) | 0.56(0.05) |
| | | 480 | 1.06(0.06) | 0.37(0.02) | 0.33(0.04) | 8.44(0.57) | 5.63(2.17) | 0.54(0.06) |
| 101/ 0.4 | 371 | 80 | 1.00(0.02) | 0.59(0.01) | 0.55(0.01) | 8.74(0.33) | 3.15(0.46) | 0.36(0.06) |
| | | 120 | 1.24(0.01) | 0.59(0.01) | 0.55(0.01) | 10.23(0.23) | 5.22(0.45) | 0.51(0.05) |
| | | 200 | 1.39(0.01) | 0.56(0.01) | 054(0.02) | 11.22(0.41) | 7.35(0.97) | 0.66(0.09) |
| | | 320 | 1.31(0.02) | 0.47(0.02) | 0.38(0.01) | 12.67(0.78) | 5.42(0.71) | 0.43(0.08) |
| | | 480 | 1.18(0.05) | 0.38(0.08) | 0.29(0.04) | 12.84(0.84) | 3.26(0.58) | 0.25(0.03) |
| | 852 | 80 | 0.97(0.02) | 0.58(0.01) | 0.54(0.02) | 6.66(0.42) | 2.51(0.33) | 0.38(0.04) |
| | | 120 | 1.08(0.01) | 0.55(0.01) | 0.49(0.01) | 9.72(0.22) | 3.96(0.74) | 0.41(0.07) |
| | | 200 | 1.27(0.01) | 0.55(0.01) | 0.51(0.02) | 10.33(0.19) | 5.09(0.34) | 0.49(0.04) |
| | | 320 | 1.22(0.01) | 0.47(0.03) | 0.37(0.03) | 10.57(0.19) | 3.76(0.49) | 0.36(0.05) |
| | | 480 | 0.90(0.01) | 0.25(0.03) | 0.17(0.01) | 11.57(0.49) | 3.19(0.56) | 0.28(0.06) |

Initial N/P, the ratio of dissolved inorganic nitrogen to phosphate at the beginning of experiment; L, light intensity (μmol photons m$^{-2}$ s$^{-1}$). See Table 2 for detailed

More detailed information is given as in Table 2. Data in the brackets are the standard deviations for four replicates.

**Table 4.** Results of two-way ANOVAs of the effects of dissolved inorganic nutrient concentration and $p\mathrm{CO_2}$ on fitted $a$ and maximum value ($V_{max}$) of growth, POC and

PIC production rates. More detailed information is given as in Table 2.

| | | Factor | $F$ value | $p$ value |
|---|---|---|---|---|
| $a$ | Growth rate | Nut | 18.08 | ≤0.001 |
| | | $CO_2$ | 0.186 | 0.6711 |
| | | Nut×$CO_2$ | 0.398 | 0.6776 |
| | POC production rate | Nut | 7.21 | 0.005 |
| | | $CO_2$ | 7.78 | 0.0121 |
| | | Nut×$CO_2$ | 2.50 | 0.11 |
| | PIC production rate | Nut | 21.73 | ≤0.001 |
| | | $CO_2$ | 2.32 | 0.145 |
| | | Nut×$CO_2$ | 2.56 | 0.105 |
| $V_{max}$ | Growth rate | Nut | 24.9 | ≤0.001 |
| | | $CO_2$ | 572.7 | ≤0.001 |
| | | Nut×$CO_2$ | 14.8 | ≤0.001 |
| | POC production rate | Nut | 7.301 | 0.0048 |
| | | $CO_2$ | 15.95 | 0.0009 |
| | | Nut×$CO_2$ | 1.91 | 0.177 |
| | PIC production rate | Nut | 56.06 | ≤0.001 |
| | | $CO_2$ | 86.84 | ≤0.001 |
| | | Nut×$CO_2$ | 0.168 | 0.85 |

[Figure]

Figure 1

[Figure]

Figure 2

[Figure]

Figure 3

[Figure]

Figure 2

[Figure]

[Figure]

Figure 4

[Figure]

Figure 4

[Figure]

[Figure]

Figure 5